# HoT: Highlighted Chain of Thought for Referencing Supporting Facts from Inputs

**Tin Nguyen**[*]
*Auburn University*

*ttn0011@auburn.edu*

**Logan Bolton**[*]
*Auburn University*

*ldb0046@auburn.edu*

**Mohammad Reza Taesiri**
*University of Alberta*

*mtaesiri@gmail.com*

**Trung Bui**

*bhtrung@gmail.com*

**Anh Totti Nguyen**[†]
*Auburn University*

*anh.ng8@gmail.com*

## Abstract

An Achilles heel of Large Language Models (LLMs) is their tendency to hallucinate non-factual statements. A response mixed of factual and non-factual statements poses a challenge for humans to verify and accurately base their decisions on. To combat this problem, we propose Highlighted Chain-of-Thought Prompting (HoT), a technique for prompting LLMs to generate responses with XML tags that ground facts to those provided in the question. That is, given an input question, LLMs would first re-format the question to add XML tags highlighting key facts, and then, generate a response with highlights over the facts referenced from the input. Compared to vanilla chain of thought prompting (CoT), HoT reduces the rate of hallucination and separately improves LLM accuracy consistently on over **22 tasks** from arithmetic, reading comprehension, to logical reasoning. When asking humans to verify LLM responses, highlights help time-limited participants to more accurately and efficiently recognize when LLMs are correct. Yet, surprisingly, when LLMs are wrong, HoTs tend to fool users into believing that an answer is correct. Code is available at: Github.

## 1 Introduction

Chains of Thoughts (CoT) enable LLMs to generate step-by-step solutions to questions, improving both (1) accuracy on many tasks (Wei et al., 2022) that benefit from problem decomposition and (2) transparency in how a model arrives at a final answer. However, a major weakness of LLMs is their tendency to hallucinate non-factual statements (Zhang et al., 2023), making it difficult for humans to verify whether an LLM answer is correct.

Based on this observations, we aim to mitigate it by proposing Highlighted Chain-of-Thought (HoT), a prompting technique that instructs LLMs to generate a CoT answer but with in-line XML tags, grounding in-response facts to in-question facts (Fig. 1). We hypothesize that HoT may improve LLM accuracy on downstream tasks and also user verification accuracy.

Existing methods attempt to combat hallucination and improve verifiability by forcing LLMs to cite websites (Perplexity; SearchGPT, 2024), documents (Bai et al., 2024) or paragraphs (Cohen-Wang et al., 2024) to

---

[*]Equal contribution
[†]Corresponding author

**Question:** Let a,b,c,d be positive real numbers such that a+b+c+d=10. Find the maximum value of ab²c³d⁴ if a=10, b=c=d=0

| Chain of Thought | Highlighted Chain of Thought |
|---|---|
| **Answer:** To find the maximum value ab²c³d⁴ given the constraint a+b+c+d=10, we can use the method of Lagrange multipliers or analyze the function directly. ...

Let'set c=d=x and b=y. Then the constraint becomes: a+y+x+x=10 implies a+y+2x=10.
…Thus the maximum value is
ab²c³d⁴ = 1(2)²(3)³(4)⁴=27658 ✗ | **Reformatted Question:** Let a,b,c,d be positive real numbers such that a+b+c+d=10. Find the maximum value of ab²c³d⁴ if a=10, b=c=d=0
‐ ‐ ‐ ‐ ‐ ‐ ‐ ‐ ‐ ‐ ‐ ‐ ‐ ‐ ‐ ‐ ‐
**Answer:** Given that a=10 and b=c=d=0, the expression we want to maximize is ab²c³d⁴. Substituting the given values, we get: 10*(0)²*(0)³*(0)⁴=10*0*0*0=0. Since the expression is a product of terms involving zero raised to non-negative powers, the entire expression evaluates to 0. There for the maximum value of ab²c³d⁴ under the given condition is 0. ✓ |

Figure 1: CoT and HoT (ours) responses for a MATH500 question in ReasoningTrap benchmarks (Jang et al., 2025), both generated by Gemini-1.5-Flash ✦⚡. **Left**: CoT misses the key constraint $b = c = d = 0$, giving an incorrect answer. **Right**: HoT (a re-formatted question and answer) applies the key constraint $b = c = d = 0$ to the expression $ab^2c^3d^4$, yielding the correct answer of 0. The full reasoning traces of both methods are provided in Tab. 38.

support statements in the response. Yet, there is no work that enables LLMs to generate regular CoTs but with references back to the in-question facts.

In HoT, first, an LLM re-formats the input question to wrap XML tags around key facts. Second, it generates its response but with XML tags around the facts that come from the input, enabling colored highlights (Fig. 1). Such highlights enable users to trace which statements in the response correspond to which facts in the input, which we hypothesize to make human verification faster and more accurate.

We test HoT on **5** LLMs including Gemini-1.5-Flash (✦⚡), Gemini-1.5-Pro (✦) (Reid et al., 2024), Llama-3.1-70B (∞₇₀B), Llama-3.1-405B (∞₄₀₅B) (Dubey et al., 2024), and GPT-4o (⑤) (OpenAI, 2024) across **22** tasks from arithmetic, logical reasoning, reading comprehension, long-context to puzzle, conditioned maths questions. Our main findings are:

1. HoT substantially reduces hallucination rate compared to CoT (from 21.22% to 14.92%) across multiple datasets, as measured by SelfCheckGPT (Manakul et al., 2023) (Sec. 5.1).
2. Compared to few-shot CoT, HoT consistently improves LLM accuracy across **5 LLMs** and **22 datasets**. On average, across arithmetic, question-answering, logical reasoning, long-context, and puzzle tasks, HoT gains +2.61, +2.58, +2.53, +2.03, +11.38 percentage points (pp), respectively (Sec. 5.2).
3. HoT outperforms other advanced prompting techniques (Least-to-Most (LtM) (Zhou et al., 2022), Tree-of-Thought (ToT) (Yao et al., 2023), Self-Refine (Madaan et al., 2023), and Chain-of-Verification (CoVE) (Dhuliawala et al., 2023)) on challenging datasets including r-GSM, Seven Objects, and Date. Furthermore, when combined with Self-Consistency (Wang et al., 2022), ComplexCoT (Fu et al., 2022) methods, HoT achieves superior performance than each of these methods alone and HoT alone (Sec. 5.4).
4. *Both* components of HoT, repeating the question and highlighting facts via XML, independently improve LLM accuracy (Sec. 5.5).
5. The colored highlighting in HoT responses improves the users' speed in verifying the accuracy of LLM answers by ∼25% (62.38s → 47.26s) and also their accuracy (Sec. 5.6).
6. Fine-tuning small LLMs on HoT responses not only enhances accuracy but also strengthens their ability to attend more effectively to input context compared to base and CoT fine-tuned models (Sec. 5.7).

## 2   Related Work

**Generating references to documents**   Recent works have trained LLMs to answer questions by generating responses that include citations to the documents from which they extract supporting information (Cohen-Wang et al., 2024; Bai et al., 2024; Gao et al., 2023; Press et al., 2024; Taylor et al., 2022; Bohnet et al., 2022). Another approach is to generate citations post-hoc, i.e., by having one LLM generate the answer first and another LLM search for citations that support facts in the answers (Ramu et al., 2024; Sancheti et al., 2024; Dasigi et al., 2021). Commercial LLM-powered search engines recently also rolled out their citation feature (Anthropic, 2025; SearchGPT, 2024; BingSearch; Perplexity), which references web pages and online documents that support statements in the responses.

Unlike above works, we focus on generating references to phrase-level and sentence-level facts (Figs. 1 and 5) in the question instead of references to context documents or paragraphs.

**Prompting techniques**   Interestingly, inputting the exact question *twice* to an LLM improves its answer accuracy slightly (Xu et al., 2024). Similarly, given a question, asking an LLM to first repeat the question and then answer it also improves accuracy compared to vanilla CoT (Mekala et al., 2024). Our work is similar in that we also ask LLMs to re-generate the question; however, the difference is that our re-generated question contains XML tags around key facts. Furthermore, unlike the above two works, HoT-prompted LLMs include tags in the *answer*, which further improve LLM accuracy as shown in our ablation study (Sec. 5.5).

**Retrieval-Augmented Generation (RAG)**   Some systems first retrieve relevant documents given an input question and then feed the documents along with the original question to an LLM for generating an answer (Asai et al., 2023; Jin et al., 2024). While they generate references to the retrieved documents (Liu, 2022), HoT generates references to fact phrases in the input question. In RAG systems, feeding HTML tags in web pages instead of plain text improves the accuracy of the retrieved knowledge (Tan et al., 2024). Functionally similar to HTML tags, our XML tags around key facts here make LLMs answer more accurately.

**Span prediction**   is a core NLP task that requires a model to read a question and a paragraph and predict the (start, end) index of the answer embedded in the paragraph (Rajpurkar et al., 2016; Dasigi et al., 2019). Similarly to span prediction, we ask LLMs to identify key phrases in the input question. However, HoT instructs LLMs to perform an extra step of generating the answer with references to the selected spans (i.e., highlighted in-question facts).

**Impact of highlights on human cognition**   In cognitive science, studies have found that selective emphasis techniques, such as text highlighting, can improve comprehension and learning for humans (Fowler & Barker, 1974; Ramírez et al., 2019). In contrast, inappropriate highlighting could harm human accuracy in reading comprehension (Gier et al., 2009; Ramírez et al., 2019). To our knowledge, we are the first to study how key-fact highlights may help users verify the accuracy of an LLM's answer.

## 3   Method

Our goal is to prompt LLMs to produce a response consisting of two parts: (1) a version of the original question with XML tags (`<fact1>`, `</fact1>`, etc.) wrapped around key facts in the input query and (2) an answer that explicitly links statements in the answer to the highlighted facts in the question (Fig. 1). **Our hypothesis** is that asking the LLM to decide which facts are worth wrapping in XML tags encourages the model to better attend to these important facts, thereby reducing hallucinations (Sec. 5.1) and improving the accuracy of the final answer (Sec. 5.2). Additionally, these XML tags can be transformed through regex and CSS to become highlighted in the GUI when the LLM answer is presented to users for verification. We experiment with different XML tag names and find that `<fact{i}></fact{i}>` where $i = 1,2,3$ results in the best LLM accuracy (Sec. H).

### 3.1   Highlighted Chain-of-Thought (HoT)

To prompt LLMs to generate HoTs (Fig. 3), we design the following prompt structure (Fig. 2) and use it for all datasets. First, the 8-shot demonstration examples (which are CoT demonstrations but with XML tags)

would show LLMs how to insert tags and answer questions. Second, the HoT instruction would be a short, explicit request that asks LLMs to insert tags into questions and answer it.

```
{Question 1} {Reformatted Q1} {Answer 1}
{Question 2} {Reformatted Q2} {Answer 2}
 ...
{Question 8} {Reformatted Q8} {Answer 8}
{Question}
{HoT Instruction}
```

Figure 2: 8-shot HoT examples are provided in addition to the explicit directions (`HoT Instruction`) (see Sec. J) to help LLMs understand the expected format. See Tab. 34 for one entire example prompt.

For each dataset, we make separate 8-shot examples (Brown et al., 2020) to demonstrate tag insertion, as instructing LLMs to do this zero-shot is non-trivial, especially since different datasets may require highlighting different types of linguistic structures.

## 3.2 Annotating HoT in-context examples

Generating demonstrations (i.e., few-shot HoT in-context examples) for each dataset from scratch can be time-consuming for humans. Therefore, we propose a 3-step, LLM-assisted approach for humans to quickly create high-quality HoT demonstrations for a given dataset, and apply the same procedure for all datasets. We begin by manually creating 15 HoT question-answer pairs across multiple domains. These HoT examples will then be used to generate extra demonstrations tailored to each dataset.

**Step 1: Humans to insert highlights into HoT meta demonstrations** We take, in total, 15 different CoT question-answer pairs from multiple datasets in a variety of domains including arithmetic, logical reasoning and question-answering tasks. Then, we manually **add XML tags to these questions** following the leave-one-out principle (Li et al., 2016) for identifying important tokens. That is, a fact (to be wrapped around, e.g., `<fact1></fact1>`) is a key phrase in the question that when removed would render the question unanswerable. Any details that are not directly relevant to answering the final question should not be tagged.

After tagging all key phrases in the question, we examine the answer to find semantically matching phrases referring to the same entity (e.g., "$ab^2c^3d^4$" vs. "$10*(0)^2*(0)^3*(0)^4$" in Fig. 1) and surround them with XML tags. That is, every tag in the answer must correspond to an existing tag in the question. These 15 human-annotated examples (see Github repo) are then used as **meta demonstrations** for LLMs to generate few-shot demonstrations for a specific dataset (Step 2).

**Step 2: LLMs to generate CoT responses for 8 questions in a given dataset** To fairly compare HoT and CoT under the common 8-shot CoT setting, we need to create 8 HoT demonstrations for each dataset. To do that, we (a) ask GPT-4o to generate standard CoT answers for 8 random questions in each dataset; and (b) convert these CoT examples into HoT examples in Step 3, leveraging the 15 meta demonstrations (generated in Step 1).

**Step 3: LLMs to insert XML tags into CoT responses for a given dataset** We prompt GPT-4o with the 15 demonstrations from Step 1 as few-shot examples, along with a **question** from Step 2, to generate a reformatted question containing XML tags (see prompts in Sec. N), producing 8 tagged questions. Finally, using the same 15 demonstrations and each tagged question, we instruct GPT-4o to generate a tagged answer, resulting in 8 tagged question-answer pairs per dataset.

## 4 Evaluation over 22 benchmarks and 5 LLMs

We evaluate our method across **22 tasks** spanning five domains: arithmetic, question answering, logical reasoning, reading comprehension, and hard, long-context benchmarks. See Sec. P for more dataset details (e.g., number of instances). We test ✦⚡, ✦, ∞70B, ∞405B, and ⑨ using their default temperatures. The configuration details of each model are listed in Sec. I.

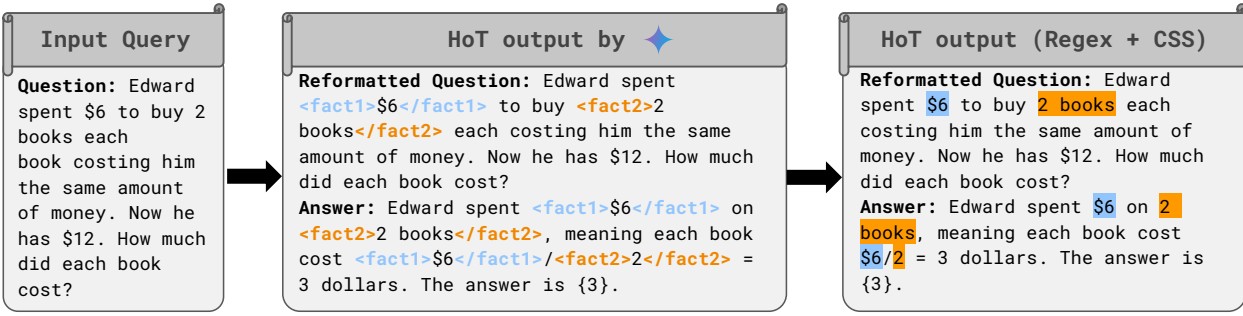

Figure 3: LLMs generate HoT responses by wrapping XML tags around the information that the model determines is the most important. Regex and CSS are then used to visualize the highlights for user readability (see the code to convert XML tags to highlights in Sec. K).

**4 Arithmetic**  We test on arithmetic tasks taken from (Wei et al., 2022): SVAMP (Patel et al., 2021), and AQUA (Ling et al., 2017). We also examine the performance of our method on R-GSM (Chen et al., 2024), which focuses on changing the question premise order to challenge LLM reading comprehension in an arithmetic setting. Finally, we evaluate on GSM-Symbolic dataset (Mirzadeh et al., 2024), which changes the original GSM8K questions through a symbolic template (i.e. changing the numerical values).

**5 Logical reasoning**  We choose five tasks from BigBench Hard (Suzgun et al., 2023): Logical Deduction Five Objects, Logical Deduction Seven Objects, Reasoning about Colored Objects, Causal Judgement, and Navigate. For brevity, we refer to these datasets as Five Objects, Seven Objects, Colored Objects, Judgement, and Navigate.

**3 Question answering**  We choose StrategyQA (Geva et al., 2021), SpartQA (Mirzaee et al., 2021), and Date (Suzgun et al., 2023) to evaluate our method on question-answering tasks.

**2 Reading comprehension**  We use the Break and Census subsets of the DROP reading comprehension benchmark (Dua et al., 2019).

**5 Hard, long-context**  To validate HoT on more challenging and long-context tasks, we evaluate HoT on GPQA Diamond (Rein et al., 2024) and four BigBench Extra Hard (BBEH) (Kazemi et al., 2025) tasks: Time Arithmetic, Spatial Reasoning, Shuffle Objects, and Causal Judgement.

**3 ReasoningTrap datasets**  LLMs often default to familiar reasoning templates, a tendency referred to as *reasoning rigidity* (Jang et al., 2025). To evaluate whether HoT can overcome this limitation, we test on two complementary datasets. The first, **PuzzleTrivial**, consists of logic puzzles that are subtly modified from their original forms, requiring models to move beyond rote solution patterns. The second, **2 Conditioned Math**, is derived from the AIME and MATH500 benchmarks, where problems are augmented with additional constraints. This design yields mathematically challenging tasks that specifically test a model's ability to adapt its reasoning under nonstandard conditions.

## 5 Results

### 5.1 HoT prompting makes LLMs hallucinate less and answer questions more consistently

To better quantify and understand how the grounding effect in HoT improves accuracy, we use the Self-CheckGPT framework (Manakul et al., 2023) to measure whether HoT mitigates hallucinations. This method outputs a hallucination likelihood score that is correlated with the prevalence of hallucinations in LLM-generated responses. We evaluate hallucination across five representative datasets from distinct domains to cover different types of reasoning: r-GSM and GSM-Symbolic (arithmetic), SpartQA (question answering), Break (reading comprehension), and Seven Objects (logical reasoning).

**Experiment**  The core idea supporting SelfCheckGPT (Sec. S) is that when an LLM truly knows a fact, multiple stochastic generations from the same prompt should yield consistent statements. In contrast,

hallucinated facts are often inconsistent across multiple independent responses. Given an LLM response $R$, SelfCheckGPT generates $N$ additional samples $\{S_1, S_2, \ldots, S_N\}$ from the same prompt and uses a *judge* LLM (here, GPT-4o) to evaluate the consistency of each sentence $r_i$ in $R$ with respect to the $N$ samples.

We run `Gemini-1.5-Flash` ✦⚡ using HoT prompting on a set of five diverse benchmarks: r-GSM, GSM-Symbolic, SpartQA, Break, and Seven Objects, and compute the answers' hallucination and consistency rates. We repeat for CoT prompting and compare the two methods.

Table 1: HoT prompting makes `Gemini-1.5-Flash` ✦⚡ **hallucinate consistently less** over a diverse set of tasks. Table shows the SelfCheckGPT hallucination scores. (Lower is better.)

| Prompt | R-GSM | GSM-Symbolic | SpartQA | Break | Seven Objects | Avg |
|---|---|---|---|---|---|---|
| CoT | 12.75 | 44.60 | 17.33 | 8.57 | 22.87 | 21.22 |
| HoT | **7.01** | **36.18** | **16.24** | **4.88** | **10.31** | **14.92** |

Table 2: The rates (%) of *unanimous* responses from `Gemini-1.5-Flash` ✦⚡ across 5 independent runs show that **HoT prompting makes LLMs more consistent in the final answers**.

| Prompt | r-GSM | GSM-Symbolic | SpartQA | Break | Seven Objects |
|---|---|---|---|---|---|
| CoT | 89.55 | 89.55 | 93.75 | 84.75 | 51.20 |
| HoT | **95.00** | **93.00** | **97.00** | **87.25** | **62.40** |

**Results** HoT consistently reduces hallucination scores for ✦⚡ across all 5 datasets (from $21.22 \rightarrow 14.92$ in Tab. 1). This improvement occurs because CoT more often misses key facts in the input question, leading to higher hallucination scores (e.g., Tabs. 46 and 47).

The relationship between hallucination reduction and accuracy increase is not necessarily linear. For example, on Seven Objects, ✦⚡'s SelfCheckGPT hallucination score nearly halves under HoT, yet accuracy increases by only $+4.40$ points (see Tabs. 1 and 6). This is because a model's reasoning could be highly consistent (as measured by SelfCheckGPT), but its final answer does not match the groundtruth for a task (measured by task accuracy). That is, hallucination reduction and accuracy measure different aspects of model behavior and are not perfectly correlated.

Given that HoT produces more unanimous responses than CoT (e.g., $+11.2$ in Seven Objects in Tab. 2), HoT achieves lower hallucination scores regardless of correctness, while CoT's varied responses appear more hallucinatory despite slightly lower accuracy.

## 5.2 HoT prompting consistently improves LLM accuracy over CoT across 5 models & 22 datasets

Since HoT makes `Gemini-1.5-Flash` provide much more consistent answers and hallucinate less over independent runs (Sec. 5.1), we test whether HoT can help LLMs improve **accuracy** over regular question-answering tasks.

We hypothesize that HoT prompting could encourage LLMs to identify and then leverage facts in the question in their chain of thoughts, thereby improving accuracy and reducing hallucinations.

**Experiment** We compare 8-shot CoT and 8-shot HoT on five LLMs, including the proprietary models `GPT-4o` (🆂), `Gemini-1.5-Flash` (✦⚡), `Gemini-1.5-Pro` (✦) and open-source models `Llama-3.1-70B` (∞70B), `Llama-3.1-405B` (∞405B) across 22 tasks. Both the CoT and HoT few-shot demonstration examples use the same questions and answers. The key difference is that HoT examples contain XML tags, while CoT examples do not.

**Results** HoT consistently improves over CoT across most datasets and models. On average, HoT improves LLM accuracy on arithmetic, question answering, logical reasoning, and long-context tasks by $+2.61$ (Tab. 4), $+2.58$ (Tab. 5), $+2.53$ (Tab. 6), and $+2.03$ pp (Tab. 7), respectively. Especially, HoT significantly increases LLM accuracy on three hard datasets: Puzzle and Conditioned Math (AIME, MATH500) by $+11.38$ (Tab. 3).

On recent adversarial, more challenging versions of GSM8K (i.e., r-GSM and GSM-Symbolic), HoT shows substantial improvements of $+3.09$ and $+0.87$ pp (Tab. 4), respectively.

HoT also outperforms CoT on benchmarks involving long and complex reasoning. Specifically, HoT achieves a $+5.55$ over CoT on GPQA Diamond ($38.38\%$ vs. $32.83\%$) and a $+4.50$ gain on BBEH Causal Judgement

(53.00% vs. 48.50%) (Tab. 7). In contrast, on BBEH Spatial Reasoning and BBEH Shuffle Objects, where questions are extremely long, HoT does not show a significant advantage over CoT.

Compared to CoT, HoT reduces contradictory errors by +10.00 on StrategyQA, +26.70 on SpartQA, and +3.30 on AQUA (see Tab. 36). Furthermore, in AQUA, CoT answers in ✦⚡ fail to even produce the answer in 7.7% of cases, while HoT has no such failures.

**Qualitative insights:** The substantial gains on Puzzle and Conditioned Math (+11.38 in Tab. 3) and most other hard datasets can be explained from an observed phenomenon that **HoT encourages LLMs to make use of the important facts already given in the input** more (e.g., Fig. 1, Tab. 37). In contrast, with vanilla CoT prompting, LLMs may overlook important facts, e.g. not using $b = c = d = 0$ in the calculation (Fig. 1), leading to incorrect answers. This ability to pay attention to facts is important for adversarial datasets (e.g. PuzzleTrivial, AIME, MATH500 (Conditioned Math), r-GSM).

Table 3: HoT outperforms CoT across 3 **ReasoningTrap** tasks, with significant gains in MATH500 (Conditioned Math) (+18.00 ✦⚡, an example shown in Fig. 1) and PuzzleTrivial (+12.50 ✦⚡, an example shown in Tab. 37).

| | Prompt | PuzzleTrivial | AIME (Conditioned Math) | MATH500 (Conditioned Math) | Task Mean ($\Delta$) |
|---|---|---|---|---|---|
| ✦⚡ | CoT | 66.25 | 44.12 | 36.00 | 48.79 |
| | HoT ($\Delta$) | **73.75** (+7.50) | **52.94** (+8.82) | **54.00** (+18.00) | **60.23** (+11.44) |
| ✦ | CoT | 65.00 | 41.12 | 44.00 | 40.50 |
| | HoT ($\Delta$) | **71.25** (+6.25) | **50.00** (+8.88) | **56.00** (+12.00) | **59.08** (+18.58) |
| ∞ 70B | CoT | 38.75 | 20.58 | 28.00 | 29.11 |
| | HoT ($\Delta$) | **45.00** (+6.25) | **26.47** (+5.89) | **38.00** (+10.00) | **36.49** (+7.38) |
| ∞ 405B | CoT | 50.00 | 17.65 | 24.00 | 30.55 |
| | HoT ($\Delta$) | **62.50** (+12.50) | **23.50** (+5.85) | **30.00** (+6.00) | **38.67** (+8.12) |
| Model Mean $\Delta$ | | +8.13 | +7.36 | +11.50 | +11.38 |

Table 4: Over all 5 LLMs, HoT consistently improves accuracy over CoT across four **arithmetic** tasks. Notably, HoT achieves the largest performance gains in AQUA (+14.64 for ✦⚡) and R-GSM (+12.73 for ✦).

| Model | Method | SVAMP | AQUA | R-GSM | GSM-Symbolic | Task Mean ($\Delta$) |
|---|---|---|---|---|---|---|
| ✦⚡ | CoT | 92.80 | 81.42 | 85.91 | 81.67 | 85.45 |
| | HoT ($\Delta$) | **94.80**(+2.00) | **96.06**(+14.64) | **86.36**(+0.45) | **83.67**(+2.00) | **90.22**(+4.77) |
| ✦ | CoT | 94.70 | 82.68 | 78.18 | 85.67 | 85.31 |
| | HoT ($\Delta$) | **95.80**(+1.10) | **91.73**(+9.05) | **90.91**(+12.73) | **86.33**(+0.66) | **91.19**(+5.89) |
| ∞ 70B | CoT | 93.00 | 90.94 | 89.09 | 82.33 | 88.84 |
| | HoT ($\Delta$) | **95.10**(+2.10) | 87.01(-3.93) | 89.09 | 82.33 | 88.38(-0.46) |
| ∞ 405B | CoT | 93.70 | 87.40 | 90.91 | **90.33** | 90.59 |
| | HoT ($\Delta$) | **95.50**(+1.80) | **88.98**(+1.58) | **91.82**(+0.91) | 90.00(-0.33) | **91.58**(+0.99) |
| 🟢 | CoT | 94.60 | 79.13 | 89.09 | 87.33 | 87.54 |
| | HoT ($\Delta$) | **95.20**(+0.60) | **82.68**(+3.55) | **90.45**(+1.36) | **89.33**(+2.00) | **89.42**(+1.88) |
| Model Mean $\Delta$ | | +1.52 | +4.98 | +3.09 | +0.87 | +2.61 |

## 5.3 Span-Level LLM Judging Reveals Lower Hallucination Rates with HoT

SelfCheckGPT (Manakul et al., 2023) primarily measures response consistency (see Sec. 5.1) and does not directly localize which parts of an output are hallucinated. To obtain span-level hallucination annotations, we instead use an LLM-as-a-judge setup (see Sec. T), which can flag specific hallucinated spans.

Aligning with (Cossio, 2025; Li et al., 2024), we use four broad hallucination types: contradiction, missing context, calculation error, and logical error.

**1. Contradiction** : Answer directly contradicts a specific fact in the question.
**2. Missing Context** : Answer ignores crucial information from the question that affects the reasoning.

Table 5: Over all 5 LLMs, HoT consistently improves accuracy over CoT across three **QA** tasks (StrategyQA, SpartQA, Date Understanding) and two **reading comprehension** tasks (Break & Census). The largest gains are observed in StrategyQA (+15.07 for 🦙70B) and SpartQA (+11.88 for ✦⚡).

| Model | Method | StrategyQA | SpartQA | Date | Break | Census | Task Mean (Δ) |
|---|---|---|---|---|---|---|---|
| ✦⚡ | CoT | 76.55 | 47.28 | 85.24 | 83.61 | 90.00 | 76.54 |
| | HoT (Δ) | **79.74** (+3.19) | **59.16** (+11.88) | **85.79** (+0.55) | **86.25** (+2.64) | 90.00 | **80.19** (+3.65) |
| ✦ | CoT | 81.75 | 61.88 | 93.31 | 86.39 | 91.75 | 83.02 |
| | HoT (Δ) | **83.45** (+1.70) | **64.85** (+2.97) | **95.82** (+2.51) | **87.36** (+0.97) | **92.50** (+0.75) | **84.80** (+1.78) |
| 🦙70B | CoT | 69.30 | 66.09 | 91.36 | 88.75 | 94.25 | 81.95 |
| | HoT (Δ) | **84.37** (+15.07) | **67.08** (+0.99) | **91.92** (+0.56) | **88.89** (+0.14) | 94.25 | **85.30** (+3.35) |
| 🦙405B | CoT | 85.33 | 69.80 | 95.54 | 90.28 | 93.50 | 86.89 |
| | HoT (Δ) | **88.43** (+3.10) | **72.28** (+2.48) | **97.49** (+1.95) | 90.28 | **94.50** (+1.00) | **88.60** (+1.71) |
| 🟢 | CoT | 83.89 | 55.00 | 96.66 | 86.75 | 86.25 | 81.81 |
| | HoT (Δ) | **85.37** (+1.48) | **59.75** (+4.75) | **97.21** (+0.55) | **87.50** (+0.75) | **90.75** (+4.50) | **84.12** (+2.41) |
| Model Mean Δ | | +4.91 | +4.61 | +1.22 | +0.90 | +1.25 | +2.58 |

Table 6: HoT outperforms CoT across five **logical reasoning** tasks from BigBench Hard with notable gains in Judgement (+15.5 for 🟢) and Five Object (+6.00 for ✦).

| Model | Method | Five Objects | Seven Objects | Colored Objects | Judgement | Navigate | Task Mean (Δ) |
|---|---|---|---|---|---|---|---|
| ✦⚡ | CoT | 78.80 | 74.80 | 94.00 | 71.66 | 92.80 | 82.41 |
| | HoT (Δ) | **82.00** (+3.20) | **79.20** (+4.40) | **95.20** (+1.20) | 71.66 | 92.80 | **84.17** (+1.76) |
| ✦ | CoT | 92.80 | 86.00 | 96.40 | 74.87 | 92.00 | 88.41 |
| | HoT (Δ) | **98.80** (+6.00) | **88.80** (+2.80) | **97.20** (+0.80) | **75.40** (+0.53) | **96.40** (+4.40) | **91.32** (+2.91) |
| 🦙70B | CoT | 92.80 | 79.60 | 92.00 | 67.91 | 87.60 | 83.98 |
| | HoT (Δ) | **94.00** (+1.20) | **83.60** (+4.00) | **93.60** (+1.60) | **71.12** (+3.21) | **90.40** (+2.80) | **86.54** (+2.56) |
| 🦙405B | CoT | 95.60 | 89.60 | 96.80 | 67.91 | 95.20 | 89.02 |
| | HoT (Δ) | **97.20** (+1.60) | **90.00** (+0.40) | **97.20** (+0.40) | **74.33** (+6.42) | **97.20** (+2.00) | **91.19** (+2.16) |
| 🟢 | CoT | **93.60** | 85.20 | 98.40 | 73.80 | 96.40 | 89.48 |
| | HoT (Δ) | 92.80 (-0.80) | **86.40** (+1.20) | **98.80** (+0.40) | **89.30** (+15.5) | 96.40 | **92.74** (+3.26) |
| Model Mean Δ | | +2.24 | +2.56 | +0.88 | +5.13 | +1.84 | +2.53 |

**3. Calculation Error** : The arithmetic computation itself is mathematically incorrect.

**4. Logical Error** : The reasoning step is logically flawed or misinterprets the problem.

**Experiment**  We apply LLM-as-Judge using Gemini-2.5-Flash to evaluate `Gemini-1.5-Flash` HoT responses across five diverse benchmarks: r-GSM, GSM-Symbolic, SpartQA, Break, and Seven Objects. Hallucination rate is computed as the proportion of questions containing at least one hallucinated span, and the same evaluation is repeated for CoT responses. We report both hallucination rate and task accuracy to assess whether HoT reduces hallucinations while preserving or improving performance compared to CoT (see Tabs. 8 and 9).

Table 7: HoT outperforms CoT across five hard, **long-context** tasks, with notable gains in Causal Judgement (+4.50 ✦⚡) and GPQA Diamond tasks (+5.55 ✦⚡).

| | Prompt | BBEH Time Arithmetic | BBEH Shuffle Objects | BBEH Spatial Reasoning | BBEH Causal Judgement | GPQA Diamond | Task Mean (Δ) |
|---|---|---|---|---|---|---|---|
| ✦⚡ | CoT | 31.00 | 3.00 | 9.50 | 48.50 | 32.83 | 24.97 |
| | HoT (Δ) | **32.00** (+1.00) | **7.00** (+4.00) | **10.00** (+0.50) | **53.00** (+4.50) | **38.38** (+5.55) | **28.08** (+3.11) |
| ✦ | CoT | 46.00 | 17.00 | 10.00 | 53.50 | 41.92 | 33.68 |
| | HoT (Δ) | **48.50** (+2.50) | **19.00** (+2.00) | **10.50** (+0.50) | **58.00** (+4.50) | **43.43** (+1.51) | **35.89** (+2.21) |
| 🦙70B | CoT | 26.50 | 14.00 | 10.00 | 47.00 | 21.21 | 23.74 |
| | HoT (Δ) | **29.00** (+2.50) | **15.00** (+1.00) | 10.00 | **50.00** (+3.00) | **25.25** (+4.04) | **25.85** (+2.11) |
| 🦙405B | CoT | 41.00 | 17.00 | 10.00 | 56.00 | 23.23 | 29.45 |
| | HoT (Δ) | **41.50** (+0.50) | 17.00 | **11.00** (+1.00) | 56.00 | **25.25** (+2.02) | **30.15** (+0.70) |
| Model Mean Δ | | +1.63 | +1.75 | +0.5 | +3.00 | +3.28 | +2.03 |

Table 8: HoT reduces the hallucination rate measured as the fraction of responses containing hallucinations.

| Hallucination Rate ↓ | R-GSM | GSM-Symbolic | SpartQA | Break | Seven Objects |
|---|---|---|---|---|---|
| CoT | 12.00 | 14.50 | 80.00 | 14.50 | 21.50 |
| HoT | 6.00 | 14.50 | 74.00 | 14.00 | 12.50 |
| $\Delta$(CoT − HoT) | 6.00 | 0.00 | 6.00 | 0.50 | 9.00 |

Table 9: Accuracy comparison between CoT and HoT across different benchmarks.

| Accuracy ↑ | R-GSM | GSM-Symbolic | SpartQA | Break | Seven Objects |
|---|---|---|---|---|---|
| CoT | 78.18 | 85.67 | 61.88 | 86.39 | 86.00 |
| HoT | 90.91 | 86.33 | 64.85 | 87.36 | 88.80 |
| $\Delta$(HoT − CoT) | 12.73 | 0.66 | 2.97 | 0.97 | 2.80 |

**Results** We observe that while HoT reduces the hallucination rate in Seven Objects by **9.00** points but only improves the accuracy by **2.80** points. One possible explanation is that the Seven Objects dataset requires complex multi-step reasoning to reach the final answer. Hallucinations often appear in intermediate reasoning steps, yet the model can still produce the correct final answer. Consequently, fixing these hallucinations yields only minimal accuracy improvements.

For reading comprehension datasets (Break, SpartQA), hallucinations primarily arise from misusing or ignoring passage information, so we evaluate them using Contradiction and Missing Context. In contrast, hallucinations in arithmetic and logical reasoning datasets (r-GSM, GSM-Symbolic, Seven Objects) mainly result from faulty computation or invalid inference, and are therefore labeled as Calculation Error and Logical Error. This dataset-specific labeling aligns evaluation with each dataset's dominant failure modes.

## 5.4  HoT outperforms CoT and other advanced prompting methods

Given that HoT provides consistent gains over CoT prompting, sometimes substantial on hard and adversarial benchmarks (Sec. 5.2), it is natural to ask: Q1: How would HoT fare against other advanced prompting techniques? Q2: Can HoT be used in tandem with other techniques to even further improve LLM accuracy?

We first examine whether HoT outperforms the Repeated Question prompting (RQ) (Mekala et al., 2024), then extend our comparison by integrating both HoT and CoT with Self-Consistency (SC) and the ComplexCoT strategy. That is, SC takes multiple LLM responses to the same prompt and selects the most frequent answer among these outputs, whereas ComplexCoT selects the most complex answer.

Additionally, we evaluate HoT against multi-step promptings including Least-to-Most (LtM), Tree-of-Thought (ToT), Self-Refine, and Chain-of-Verification (CoVE) (see details at Sec. U).

**Experiment** We evaluate HoT, CoT, RQ, CoT + SC and HoT + SC, ComplexCoT, and ComplexHoT (HoT + ComplexCoT) across five independent runs on 17 benchmarks (7 Arithmetic, 5 Logical Reasoning, 3 Question Answering, and 2 Reading Comprehension, details of each dataset are in Sec. P) and two models (✦⚡, ✦) (Tabs. 4 to 6) and report the mean and standard deviation (std) to ensure reliability. Mean accuracies across all benchmarks are in Tab. 10.

We also compare HoT against four other state-of-the-art prompting techniques on 3 datasets (r-GSM, Seven Objects, and Date) across 5 runs and two models (✦⚡, ✦). The reported mean accuracies are in Tab. 11.

**Result** Over 17 datasets, HoT outperforms both RQ and CoT (Tab. 10). Interestingly, HoT even surpasses CoT + SC by +1.02 (87.24 vs. 86.22; Tab. 10).

When combining HoT with SC, the resultant HoT + SC method surpasses HoT alone and even CoT+SC. Similarly, ComplexHoT outperforms HoT alone and ComplexCoT alone consistently (Tab. 10). That is, HoT provides a **distinct benefit** to LLMs as their accuracy continues to improve as HoT prompting is used in combination with other approaches that reward consistency (SC) and longer thinking (ComplexCoT).

Table 10: Over 5 runs across 17 benchmarks, HoT consistently outperforms both CoT and Repeating Questions (RQ), and even CoT + Self-Consistency (SC), and ComplexCoT. ComplexHoT and HoT + SC also outperforms their counterparts (ComplexCoT and CoT + SC) showing that HoT can complement these methods.

| Model | CoT | RQ | HoT | CoT + SC | HoT + SC | ComplexCoT | ComplexHoT |
|---|---|---|---|---|---|---|---|
| ✦⚡ | 83.21±0.82 | 83.51±1.98 | **85.17±1.85** | 83.68 | **87.18** | 83.79 | **86.37** |
| ✦ | 87.61±1.09 | 88.35±1.52 | **89.31±0.92** | 88.76 | **90.39** | 89.11 | **90.71** |
| **Mean ± std** | 85.41±0.96 | 85.93±1.75 | **87.24±1.38** | 86.22 | **88.79** | 86.45 | **88.54** |

On average over 5 runs and 3 datasets, HoT alone is still the most performing method compared to all other advanced prompting methods of CoT, LtM, CoVE, Self-Refine, and ToT (Tab. 11). Under other prompting techniques, we observe LLMs often miss critical facts (e.g., overlooking temporal indicators like "yesterday" in Date), causing incorrect answers. In contrast, LLMs tend to focus better on key facts under HoT prompting.

Table 11: Mean ± standard deviation over 5 runs across r-GSM, Seven Objects, and Date comparing HoT against LtM, ToT, Self-Refine, and CoVE. HoT consistently outperforms these advanced methods.

| Model | CoT | LtM | CoVE | Self-Refine | ToT | HoT |
|---|---|---|---|---|---|---|
| ✦⚡ | 81.41±0.90 | 75.94±1.42 | 74.07±1.34 | 70.32±0.72 | 81.98±1.70 | **84.21±1.45** |
| ✦ | 88.94±2.07 | 84.38±0.62 | 81.54±1.27 | 81.74±1.66 | 87.39±0.92 | **91.87±0.91** |
| **Mean** | 85.18±3.77 | 80.16±4.22 | 77.81±3.73 | 76.03±5.71 | 84.69±2.71 | **88.04±3.83** |

## 5.5 Ablation study: Repeating the question and adding <fact> tags improves accuracy

As HoT consistently outperforms CoT over many tasks (Secs. 5.2 and 5.4), a burning question is: What components contribute to HoT's success?

HoT consists of two steps: (a) regenerating the question but with XML tags around key facts; and (b) adding tags to the answer. Interestingly, concurrent work showed that repeating the question alone can help improve LLM accuracy (Mekala et al., 2024). Therefore, we perform an ablation study to understand the impact of each HoT component: (1) Repeating the question alone (no tags); (2) Repeating and adding tags to the question; and (3) Adding tags to the answer, but not to the question.

**Experiment** We compare the following variations: (a) **CoT**: Few-shot CoT baseline (Wei et al., 2022); (b) **Repeated Question (R-Q)**: Repeat the input question and then generate the regular CoT answer (Mekala et al., 2024).
(c) **Tags in Question (T-Q)**: Repeat the question and wrap key facts in it with XML tags while leaving the answer untagged.
(d) **Tags in Answer (T-A)**: Repeat the question and wrap key facts in the answer with XML tags.
(e) **Tags in Question and Answer (HoT)**: The full HoT recipe, i.e. wrapping XML tags around key facts in the reformatted question and also adding corresponding tags to the answer.

For each variation, we evaluate 4 models (✦⚡, ✦, ∞₇₀ᴮ, ∞₄₀₅ᴮ) on 400 randomly sampled instances across 6 diverse, representative datasets (AQUA, StrategyQA, R-GSM, Seven Objects, Judgement, & Navigate).

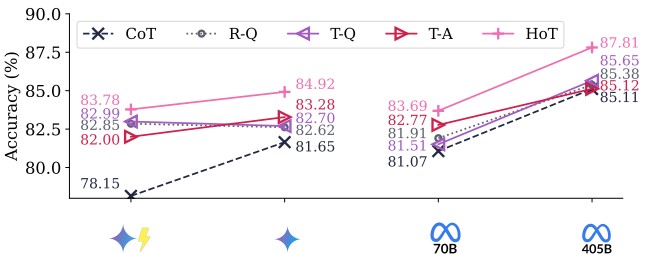

Figure 4: HoT ablation study: Every component—repeating the question (R-Q), adding tags to only question (T-Q), adding tags to only answer (T-A)—independently contributes to the overall accuracy of HoT prompting (+). Each component also outperforms the vanilla CoT (-×-). *y*-axis shows mean accuracy across 6 datasets (the detailed accuracy of each dataset is in Sec. M).

**Results**  First, on average across 6 datasets and 4 LLMs, every variation (R-Q, T-Q, T-A, and HoT) outperforms the baseline CoT (Fig. 6). That is, each HoT component is an improvement over CoT. Second, for some smaller LLMs (✦⚡, ∞₇₀ᵦ), adding each component monotonically increases accuracy while the trend is weaker for larger models (✦).

Interestingly, examining the results of the T-A method shows that, for larger LLMs (i.e. ✦ and ∞₄₀₅ᵦ), while we instruct them via demonstrations to insert tags exclusively to the answer (but not the question), these two models still generate, on average, 3.27 and 2.27 tags, respectively, in each question (Tab. 29). As the result, instructing LLMs to generate **tags exclusively in the answer (T-A) is sufficient for large LLMs to gain accuracy over not having tags at all (CoT, and R-Q)**.

We also find that intentionally scrambling tags in the QA pairs in the few-shot examples significantly reduces HoT's accuracy across datasets (-2.13 pp Sec. F). However, even with mismatched tags, HoT still outperforms CoT by +1.21 pp (Sec. F) perhaps because LLMs do not highlight tags randomly as shown in the demonstrations.

### 5.6  HoT highlights help 🧑 humans improve their speed of verifying LLM answers

High-quality text highlights in classification tasks reduce decision time and effort, while low-quality highlights harm user accuracy (Ramírez et al., 2019). Here, we also aim to assess how our highlights in HoT impact user decision verification accuracy and time. That is, users are asked to decide whether the final answer is accurate given the input question, and the HoT answers. To our knowledge, this would be the first study in the LLM literature that studies how highlights in chains of thought impact users on a downstream task.

**Experiment**  To evaluate the impact of highlights to humans on arithmetic and reading comprehension problems, we measure user accuracy in verifying whether LLM responses (of ∞₈ᵦ, ∞₇₀ᵦ, ∞₄₀₅ᵦ, ✦ and ✦⚡) are accurate for GSM-Symbolic and DROP questions. This verification task is known as **distinction** (Kim et al., 2022) or **verification** task (Taesiri et al., 2022; Nguyen et al., 2021).

We select 30 incorrect and 30 correct HoT responses from each dataset, forming a balanced pool of questions with 120 cases of HoT. We then remove the XML `<fact>` tags from HoT responses, resulting in 120 HoT and 120 CoT responses for user verification.

We recruit 63 users, consisting of undergraduate and graduate students, each verifying 10 LLM responses to questions from GSM-Symbolic and DROP via an online interface (see Sec. A). Users are randomly assigned to see exclusively HoT or CoT responses and then have to predict if a response is `correct` or `incorrect`. To simulate real-world constraints, users are given a maximum of 2 minutes per question, after which, they are required to make a decision.

| Method | Avg Time (secs) | Accuracy (%) LLM is correct | Accuracy (%) LLM is incorrect |
|---|---|---|---|
| HoT | **47.26** | **84.48** ± 20.28 | 54.83 ± 30.13 |
| CoT | 62.38 | 78.82 ± 28.26 | **72.21** ± 21.99 |

Table 12: 🧑 Users spend ∼25% less time when verifying HoT answers (compared to no highlights, i.e. CoT). Highlights make users accept LLM answers more, yielding higher accuracy in `correct` cases but worse in `incorrect` ones.

**Results**  On average, over all 240 cases including both `correct` and `incorrect` answers, **users spend nearly 25% less time** (47.26 vs. 62.38 seconds, i.e. +15.12 seconds faster) when making decisions with highlights (Tab. 12). Interestingly, users perform better on `correct` cases (84.48 vs. 78.82) but worse on `incorrect` cases (54.83 vs. 72.21). That is, **HoT highlights tend to make users believe that an answer is accurate** (Sec. B), making them more likely to accept the answer. This aligns with recent findings (Jaźwińska & Chandrasekar, 2025; Wu et al., 2025) showing that search engines and LLMs often generate misleading citations, causing users to wrongly trust their answers.

**In practice:**  LLMs are often more correct than wrong. Therefore, the ratio of `correct`/`incorrect` cases is far above 50/50. That is, we report user accuracy on these two subsets separately (Tab. 12) as taking an average might lead to *misleading* interpretation. For real-world datasets, **highlights in HoT is still**

**estimated to improve the user verification accuracy** over an entire dataset, i.e., both `correct` and `incorrect` cases (Sec. D).

### 5.7 Learning to highlights make LLMs improve OOD accuracy and pay more attention to key facts

Our previous sections showed that HoT prompting improves the accuracy consistently for large LLMs including Gemini and Llama families. However, prior research also found that smaller LLMs tend to struggle in following instructions (Brown et al., 2020). Similarly, our preliminary results reveal that HoT few-shot prompting does not help small LLMs of a few billion parameters—they fail to highlight facts appropriately.

To smaller LLMs generate highlighted chains of thoughts, we first use HoT few-shot prompting to generate SFT training examples and and repeat the same procedure to generate CoT examples separately. Then, we finetune small open-weight LLMs on these two datasets and compare HoT-finetuned and CoT-finetuned LLMs against the baselines. One might also ask: **Would fine-tuning improve LLM performance on out-of-distribution (OOD) datasets?** Our answer: Yes! We observe that HoT-finetuned models can recall input facts more accurately, resulting in more accurate answers (Tab. 13).

To shed light on why accuracy improves after finetuning on HoT examples, we perform an attention entropy analysis (Zhang et al., 2024) and find that HoT-finetuned models pay more attention to tagged facts than the baseline and CoT-finetuned models.

**Experiment** We finetune LLaMA-3.2-1B (Meta, 2024) and Qwen2.5-1.5B (Yang et al., 2024) on 3,187 CoT responses, and 3,187 HoT responses in total that `Gemini-1.5-Pro` ✦ generates for 17 datasets (i.e., those datasets previously shown in Tabs. 4 to 6). Then, we compare the accuracy of 3 models (original model, model finetuned on CoT, and finetuned on HoT) across 5 OOD benchmarks (Tab. 13). We also measure the attention entropy of three models on PuzzleTrivial questions (Tab. 14) to gain insights into why HoT-training may improve LLM accuracy.

Table 13: HoT-finetuned LLMs consistently outperform both the LLMs finetuned on CoT and the base model across 5 OOD tasks including: 3 ReasoningTrap (PuzzleTrivial, AIME, and MATH500), and 2 long-context, logical tasks (BBEH Causal Judgement, BBEH Time Arithmetic), an example shown in Fig. 5).

| Model | PuzzleTrivial | AIME (Conditioned Math) | MATH500 (Conditioned Math) | BBEH Causal Judgement | BBEH Time Arithmetic | Mean ($\Delta$) |
|---|---|---|---|---|---|---|
| Qwen-2.5-1.5B | 13.75 | 14.70 | 20.00 | 39.50 | 18.50 | 21.30 |
| + SFT on CoT | 28.75 | 11.76 | 18.00 | 15.00 | 10.00 | 16.70 |
| + SFT on HoT | **38.75** | **29.40** | **28.00** | **40.00** | **28.50** | **32.94** |
| Llama-3.2-1B | 11.20 | 2.90 | 10.00 | 21.00 | 30.50 | 15.12 |
| + SFT on CoT | 13.75 | 5.90 | 2.00 | 29.50 | 45.00 | 19.24 |
| + SFT on HoT | **18.75** | **11.80** | 10.00 | **32.00** | **52.50** | **27.02** |

**Results** Llama-3.2-1B finetuned on HoT examples achieves the best accuracy compared to the CoT-finetuned model and the base model (27.02 vs 19.24 and 15.12). Similarly, Qwen-2.5-1.5B finetuned on HoT examples achieves the best accuracy compared to the CoT fine-tuned model and base model (32.94 vs 21.30 and 16.70) (see Tab. 13). After finetuning on HoT, LLMs does not need few-shot prompting anymore. Instead, they generate answers directly with highlights over facts (see Fig. 5).

Interestingly, we find LLMs finetuned on HoT training examples demonstrate a more focus attention maps, i.e., consistently lower attention entropy (e.g., 1.524 for HoT-finetuned Llama-3.2-1B vs. 1.671 for the base Llama-3.2-1B; Tab. 14). Qualitatively, over hard, adversarial datasets such as PuzzleTrivial, HoT-finetuned LLMs accurately recall the key facts that may be uncommon (e.g., *"permanently infertile lions"*; Fig. 5) to answer tricky questions correctly. In contrast, the base LLMs or CoT-finetuned models often overlook such facts (Fig. 5; Left), yielding wrong answers.

| Attention Entropy | Qwen2.5-1.5B | Llama-3.2-1B |
|---|---|---|
| Base | 2.150 | 1.671 |
| + SFT on CoT | 2.141 | 1.847 |
| + SFT on HoT | **1.773** | **1.524** |

Table 14: LLMs finetuned on HoT show lower attention entropy on PuzzleTrivial questions than the CoT-finetuned and base models. Attention entropy is computed as $H(A) = -\sum_i p_i \log(p_i)$, where $A$ denotes the attention distribution over tokens and $p_i$ is the normalized attention weight assigned to token $i$.

Figure 5: **Left:** After finetuned via SFT on CoT examples, Qwen-2.5-1.5B answers incorrectly an adversarial question from PuzzleTrivial as it does *not* factor in the key fact of *"permanently infertile lions"*. **Right:** In contrast, HoT-finetuned counterpart LLM can highlight facts and answer correctly using the fact (*"the lions would never reproduce"*).

## 6 Limitations

**First**, while larger models such as 405B, and are generally able to follow the format reliably, smaller models like 8B and can exhibit inconsistencies, sometimes repeating the few-shot examples (Sec. L). Smaller LLMs might incorrectly repeat the question (in Sec. O) or fail to adhere to the tag format, as demonstrated in the few-shot examples (Sec. L). **Second**, HoT prompting inherently requires LLMs to repeat the question, which costs extra tokens and could be problematic for long-context questions (see Sec. C). **Third**, HoT prompting requires task-specific demonstrations, for which we manually create 8 examples per dataset. However, highlighting is a costly human effort and may be non-trivial to even define for some domains.

## 7 Discussions and Conclusion

We further fine that HoT few-shot prompting causes both `Llama-3.1-70B` and `Llama-3.1-405B` to assign more attention mass to the XML tags tokens (e.g., `<fact1>`) than to other tokens outside the tags (Sec. E). Internally, such attention pattern may improve LLM focus on key facts, thus improving accuracy and reducing hallucination. Additionally, finetuning thinking LLMs DeepSeek-R1 to highlight facts and intermediate results in their *long reasoning chains* (see Secs. Q and R) might improve their accuracy and help users interpret their complex reasoning chains faster.

In this paper, we find the highlights to reduce the verification speed of users substantially; however, do highlights improve the verification accuracy of another LLM judge? Our preliminary results do not find clear evidence supporting this hypothesis (Sec. G), which requires further research.

We present HoT, a novel prompting approach that enables LLMs to directly reference text from the input question in their responses. HoT improves LLM accuracy on arithmetic, question answering, and logical reasoning tasks. Furthermore, highlights in HoT answers also improve user verification speed.

## 8  Broader Impact Statement

HoT highlights supporting information alongside model answers, but it can also introduce risks. When a model presents highlighted evidence, the output may appear more trustworthy—even if the reasoning is flawed or the conclusion is incorrect. Our human study Sec. 5.6) shows that highlighting can increase users' confidence in model responses, including incorrect ones. In such cases, the structured format may create a false sense of reliability. This is especially concerning in high-stakes areas such as medicine, law, or financial decision-making, where small errors can have serious consequences.

We emphasize that highlighting does not guarantee correctness. Users should treat HoT as a tool for inspection, not as proof that the answer is accurate. Future work should explore additional safeguards, such as independent verification, uncertainty estimation, and human review, to reduce the risk of over-trust.

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
