# HoT: Highlighted Chain of Thought for
# Referencing Supporting Facts from Inputs

## A 🧑 Human Study

### A.1 User Instructions

At the beginning of the quiz, the users were given the following instructions: 1. Judge whether each answer is **correct** or **incorrect** using only the provided reasoning.

2. Do not use external resources or scratch paper.

3. You will evaluate 10 randomly selected questions under a 20-minute time limit.

4. Complete the quiz on a computer (NO phones).

Following these instructions, participants were shown an example CoT or HoT response (correct or incorrect), depending on their assigned group. They then completed two trivial practice questions, presented in the same CoT or HoT format. Finally, participants were informed that the actual quiz was about to begin, with a maximum time limit of two minutes per question.

### A.2 Data Analysis

During analysis, we observed that some participants answered most questions legitimately but selected answers randomly near the end to finish quickly. To mitigate the impact of such behavior while preserving valid earlier responses, we filtered out answers with response times under 5 seconds.

If a participant exceeded the 2-minute time limit for a question, the response was recorded as selecting the option indicating incorrect reasoning. We adopted this rule because failure to verify an answer within the allotted time suggests low confidence in its validity, making it reasonable to label the answer as incorrect.

### A.3 Interface Examples

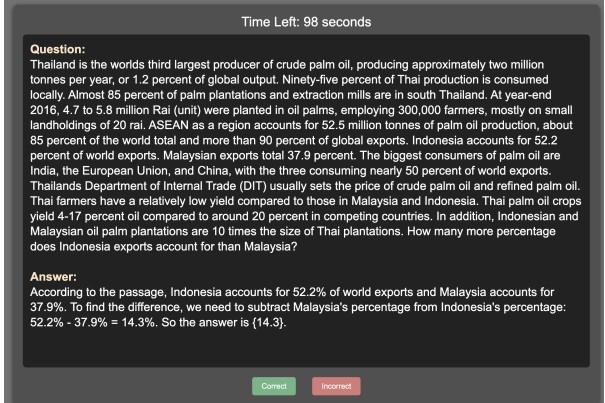 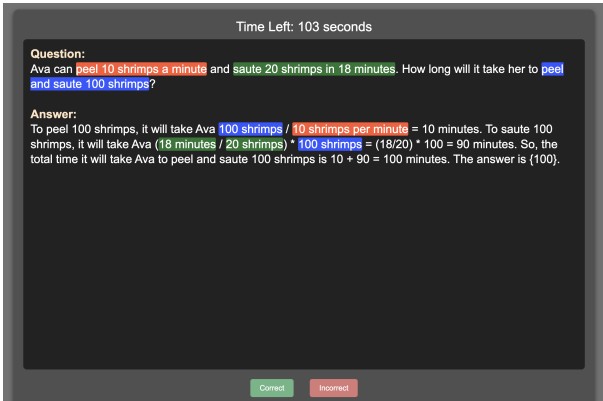

Figure 6: Left: Screenshot of a CoT question from the DROP dataset in the online interface. Right: Screenshot of a HoT question from the GSM-Symbolic dataset in the online interface.

# B  Examples Where Users Misjudged the HoT Responses

To better understand how HoT can sometimes lead users to accept incorrect answers, we analyze representative cases where users misjudged the model's HoT output (Tabs. 15 and 16), highlighting the specific failure modes that contributed to these errors.

Table 15: Examples of LLMs where Users Incorrectly Accept LLM HoT Answers of DROP (Reading Comprehension task).

| |
|---|
| DROP The model misses some key facts (37-yard field goal is not highlighted by the LLM) |
| **Question**: ... New Orleans would respond with Brees' 3-yard touchdown pass to tight end David Thomas. The Saints would add onto their lead in the fourth quarter as kicker Garrett Hartley made a 46-yard and a 19-yard field goal. The 49ers would strike back as Gore got a 7-yard touchdown run, followed by Smith's 2-point conversion pass to tight end Vernon Davis. In the end, New Orleans got the last laugh as Hartley booted the game-ending 37-yard field goal. How many field goals did Garrett Hartley kick in the fourth quarter? 
 **LLM Answer**: The passage states that in the fourth quarter, Garrett Hartley kicked a 46-yard and a 19-yard field goal. This means he kicked two field goals in the fourth quarter. So the answer is {2}. ✗ (**GT: 3**) |
| DROP Misread the Question Requirements (the LLM should focus on the requirement "the popular uprising start in?") |
| **Question**: The popular uprising included large areas of Lower Bavaria, the Innviertel and eastern Bavaria. The lands adjoining towns and strongholds were captured by the rebels and the peasant uprising spread to the Bavarian Forest, parts of the Upper Palatinate and Kelheim on the Danube. A Braunau Parliament meeting was held in December 1705 in the then-Bavarian Braunau.... How many regions did the popular uprising start in? 
 **LLM Answer**: According to the passage, the popular uprising started in Lower Bavaria, the Innviertel, and eastern Bavaria initially, then spread to the Bavarian Forest, the Upper Palatinate, and Kelheim on the Danube. So the answer is {6}. ✗ (**GT: 3**) |

Table 16: Examples of LLMs where Users Incorrectly Accept LLM HoT Answers of GSM-Symbolic (Arithmetic task).

| |
|---|
| GSM-Symbolic (Correct fact highlighting but missing reasoning steps - The LLM misses one step to subtract ¥330 to ¥84 = ¥246) |
| **Question**: Arash is raising money for a music festival. He has applied for help from the youth club, which has decided to cover 1/5 of the cost of the music festival. How much money is Arash missing if he has ¥90 and the music festival costs ¥420? 
 **LLM Answer**: The youth club will cover 1/5 of the cost of the music festival, which is 1/5 * ¥420 = ¥84. Arash has ¥90, so he has ¥90 - ¥84 = ¥6 more than the amount covered by the youth club. Since the music festival costs ¥420, Arash is missing ¥420 - ¥90 = ¥330. ✗ (**GT: 246**) |
| DROP |
| GSM-Symbolic (Incorrect arithmetic errors - The LLM incorrectly computed 17×241=4100) |
| **Question**: A fog bank rolls in from the ocean to cover a city. It takes 241 minutes to cover every 5 miles of the city. If the city is 85 miles across from the oceanfront to the opposite inland edge, how many minutes will it take for the fog bank to cover the whole city? 
 **LLM Answer**: To find the total time it will take for the fog bank to cover the whole city, we need to divide the total distance across the city by the distance covered in the given time and then multiply by the given time. The total distance across the city is 85 miles and the distance covered in the given time is 5 miles. So, the number of times the given distance fits into the total distance is $85/5 = 17$. Now, we multiply this by the given time, which is 241 minutes. Therefore, the total time it will take for the fog bank to cover the whole city is 17 * 241 minutes = 4100 minutes... ✗ (**GT: 4097**) |

## C  Tagging-Induced Token Overhead

In this section, we analyze the token overhead introduced by HoT to quantify how much additional context it consumes and to assess its impact on long-context reasoning.

Table 17: Token Overhead Across Long-context Datasets. Overal, although the tags add about 61.80% overhead on average, they lead to clear improvements in both accuracy and traceability. Note: RQ is Reformatted Question.

| Dataset | Model | Question Avg | CoT Avg | HoT Total | HoT RQ | HoT Answer | Overhead (HoT RQ/HoT Total) |
|---|---|---|---|---|---|---|---|
| Census | ✦ | 343 | 62 | 469 | 361 | 105 | 77.1% |
| Break | ✦ | 345 | 72 | 481 | 371 | 108 | 77.0% |
| bbeh_spatial_reasoning | ✦ | 1445 | 594 | 1810 | 1200 | 608 | 66.3% |
| bbeh_time_arithmetic | ✦ | 324 | 498 | 796 | 256 | 537 | 32.2% |
| bbeh_shuffle_objects | ✦ | 3244 | 911 | 1073 | 608 | 462 | 56.6% |
| **Mean ± Std** | - | $1140 \pm 1136$ | $427 \pm 324$ | $926 \pm 495$ | $559 \pm 340$ | $364 \pm 215$ | $61.8\% \pm 16.7$ |

To compute token overhead, we compute the relative increase in token length introduced by HoT. Let $L_{\mathrm{RQ}}$ denote the tokenized length of the reformatted question and $L_{\mathrm{HoT\_total}}$ the total tokenized length of the HoT (including reformatted question and answer). The overhead for that is:

$$\text{Overhead} = \frac{L_{\mathrm{RQ}}}{L_{\mathrm{HoT\_total}}} \times 100.$$

We compute the mean and std overhead across all long-context datasets in Tab. 17, showing that HoT introduces moderate additional tokens while preserving strong performance and improving traceability.

## D  Estimated User Verification Accuracy

Because testing users on all 17 datasets (Tabs. 4 to 6) is infeasible, here, we estimate the verification accuracy of users when highlights are displayed or not (i.e. HoT vs. CoT).

We estimate the verification accuracy $P$ of users on a given dataset by the following equation:

$$P = M_c \times H_c + (1 - M_c) \times H_i$$

where $M_c$ is the actual LLM accuracy for the dataset (Tabs. 4 to 6), $H_c$ and $H_i$ are the 🧑 human accuracy scores on the correct and incorrect cases, respectively (Tab. 12).

Since LLMs tend to have relatively high accuracy of $\geq 80\%$ on most 17 datasets, we find that the averaged estimated verification accuracy of HoT across all types of datasets (Sec. D.1) is higher to CoT by an average of +3.33 across all five LLMs (Tab. 18).

| Model | Prompt | Arithmetic tasks Mean ($\Delta$) | Reading Comp. tasks Mean ($\Delta$) | Logical Reasoning tasks Mean ($\Delta$) |
|---|---|---|---|---|
| ✦⚡ | CoT | 78.15 | 77.40 | 77.63 |
| | HoT ($\Delta$) | **82.44** (+4.28) | **79.25** (+1.85) | **79.80** (+2.16) |
| ✦ | CoT | 78.15 | 77.76 | 78.03 |
| | HoT ($\Delta$) | **82.69** (+4.54) | **80.44** (+2.68) | **81.92** (+3.89) |
| ∞ 70B | CoT | 78.29 | 77.72 | 77.74 |
| | HoT ($\Delta$) | **82.14** (+3.86) | **80.47** (+2.75) | **80.50** (+2.30) |
| ∞ 405B | CoT | 78.37 | 78.01 | 78.07 |
| | HoT ($\Delta$) | **82.75** (+4.38) | **81.39** (+3.38) | **81.88** (+3.81) |
| 🤖 | CoT | 78.25 | 77.70 | 78.10 |
| | HoT ($\Delta$) | **82.33** (+4.08) | **80.21** (+2.50) | **82.34** (+3.53) |
| Mean $\Delta$ | | +4.23 | +2.63 | +3.14 |

Table 18: The estimated 🧑 human verification accuracy on all 17 benchmarks is higher with highlights (HoT) rather than without highlights (CoT). More results in Sec. D.1.

## D.1 Detail of the estimated user verification accuracy

The following tables (from Tab. 19 to Tab. 21) are the estimated verification accuracies of humans if they were to verify each of the CoT and HoT results that we gather in our benchmarking. We calculate these results based off the recorded verification accuracies in the human study (Tab. 12) and the equation described in Sec. 5.6.

Table 19: Estimated human verification accuracy for HoT on arithmetic tasks.

| Model | Prompt | GSM8K | MultiArith | ASDiv | SVAMP | AQUA | R-GSM | GSM-Symbolic | Task Mean (Δ) |
|---|---|---|---|---|---|---|---|---|---|
| | CoT | 78.40 | 78.73 | 78.60 | 78.32 | 77.57 | 77.87 | 77.59 | 78.15 |
| ✦⚡ | HoT (Δ) | **82.72** (+4.32) | **84.30** (+5.57) | **83.65** (+5.05) | **82.96** (+4.64) | **83.33** (+5.76) | **80.45** (+2.58) | **79.65** (+2.06) | **82.44**(+4.29) |
| | CoT | 78.50 | 78.63 | 78.63 | 78.45 | 77.66 | 77.36 | 77.85 | 78.15 |
| ✦ | HoT (Δ) | **83.15** (+4.65) | **84.25** (+5.62) | **83.91** (+5.28) | **83.25** (+4.80) | **82.04** (+4.38) | **81.80** (+4.44) | **80.44** (+2.59) | **82.69** (+4.54) |
| | CoT | 78.45 | 78.74 | 78.56 | 78.34 | 78.20 | 78.08 | 77.63 | 78.29 |
| ∞ 70B | HoT (Δ) | **82.90** (+4.45) | **84.30** (+5.56) | **83.62** (+5.06) | **83.04** (+4.70) | **80.64** (+2.44) | **81.26** (+3.18) | **79.25** (+1.62) | **82.14**(+3.85) |
| | CoT | 78.52 | 78.72 | 78.61 | 78.38 | 77.97 | 78.20 | 78.16 | 78.37 |
| ∞ 405B | HoT (Δ) | **83.26** (+4.74) | **84.25** (+5.53) | **83.72** (+5.11) | **83.16** (+4.78) | **81.23** (+3.26) | **82.07** (+3.87) | **81.53** (+3.37) | **82.75** (+4.38) |
| | CoT | 78.50 | 78.73 | 78.62 | 78.44 | 77.42 | 78.08 | 77.96 | 78.25 |
| 🟢 | HoT (Δ) | **83.22** (+4.72) | **84.30** (+5.57) | **83.38** (+4.76) | **83.07** (+4.63) | **79.36** (+1.94) | **81.66** (+3.58) | **81.33** (+3.37) | **82.33** (+4.08) |
| Model Mean Δ | | +4.58 | +5.57 | +5.05 | +4.71 | +3.56 | +3.53 | +2.60 | +4.23 |

Table 20: Estimated human verification accuracy for HoT on reading comprehension tasks.

| Model | Prompt | StrategyQA | SpartQA | Date | Break | Census | Task Mean (Δ) |
|---|---|---|---|---|---|---|---|
| | CoT | 77.25 | 75.32 | 77.83 | 77.72 | 78.14 | 77.45 |
| ✦⚡ | HoT (Δ) | **78.48** (+1.23) | 72.37 (-2.95) | **80.28** (+2.45) | **80.42** (+2.70) | **81.53** (+3.39) | **79.25** (+1.80) |
| | CoT | 77.60 | 76.28 | 78.36 | 77.90 | 78.26 | 77.76 |
| ✦ | HoT (Δ) | **79.58** (+1.98) | 74.06 (-2.22) | **83.26** (+4.90) | **80.75** (+2.85) | **82.27** (+4.01) | **80.44** (+2.68) |
| | CoT | 76.77 | 76.56 | 78.23 | 78.06 | 78.42 | 77.72 |
| ∞ 70B | HoT (Δ) | **79.86** (+3.09) | 74.72 (-1.84) | **82.10** (+3.87) | **81.20** (+3.14) | **82.79** (+4.37) | **80.47** (+2.75) |
| | CoT | 77.83 | 76.81 | 78.51 | 78.16 | 78.37 | 78.01 |
| ∞ 405B | HoT (Δ) | **81.06** (+3.23) | 76.27 (-0.54) | **83.75** (+5.24) | **81.61** (+3.45) | **82.87** (+4.50) | **81.39** (+3.38) |
| | CoT | 77.74 | 75.83 | 78.58 | 77.93 | 77.89 | 77.70 |
| 🟢 | HoT (Δ) | **80.15** (+2.41) | 72.55 (-3.28) | **83.67** (+5.09) | **80.79** (+2.86) | **81.75** (+3.86) | **80.21** (+2.51) |
| Model Mean Δ | | +2.39 | -2.17 | +4.31 | +3.00 | +4.03 | +2.63 |

Table 21: Estimated human verification accuracy for HoT on the subset of logical tasks in BBH.

| Model | Prompt | BBH: deduction five objects | deduction seven objects | reasoning colored objects | causal judgement | navigate | Task Mean (Δ) |
|---|---|---|---|---|---|---|---|
| | CoT | 77.40 | 77.14 | 78.40 | 76.93 | 78.32 | 77.63 |
| ✦⚡ | HoT (Δ) | **79.15** (+1.75) | 78.32 (+1.18) | **83.07** (+4.67) | 76.08 (-0.85) | **82.36** (+4.04) | **79.80** (+2.17) |
| | CoT | 78.32 | 77.88 | 78.56 | 77.14 | 78.27 | 77.99 |
| ✦ | HoT (Δ) | **84.14** (+5.82) | **81.17** (+3.29) | **83.67** (+5.11) | 77.19 (+0.05) | **83.43** (+5.16) | **81.92** (+3.93) |
| | CoT | 78.32 | 77.45 | 78.27 | 76.68 | 77.98 | 77.74 |
| ∞ 70B | HoT (Δ) | **82.72** (+4.40) | **79.63** (+2.18) | **82.60** (+4.33) | 75.92 (-0.76) | **81.65** (+3.67) | **80.50** (+2.76) |
| | CoT | 78.51 | 78.11 | 78.59 | 76.68 | 78.48 | 78.07 |
| ∞ 405B | HoT (Δ) | **83.67** (+5.16) | **81.53** (+3.42) | **83.67** (+5.08) | 76.88 (+0.20) | **83.67** (+5.19) | **81.88** (+3.81) |
| | CoT | 78.38 | 77.82 | 78.69 | 77.07 | 78.56 | 78.10 |
| 🟢 | HoT (Δ) | **82.36** (+3.98) | **80.46** (+2.64) | **84.14** (+5.45) | **81.32** (+4.25) | **83.43** (+4.87) | **82.34** (+4.24) |
| Model Mean Δ | | +4.22 | +2.54 | +4.93 | +0.58 | +4.59 | +3.14 |

# E    HoT causes Llama-3.1 variants to allocate more attention around important tokens

In order to compute the attention score for the XML tags and their contents, across all heads for the final layer of the model, we measure the amount of attention on (1) the XML tag tokens (e.g., <fact1>, </fact1>); and also (2) the regular content tokens inside those XML tags.

Let

$$A^{L,h} \in \mathbb{R}^{T \times T}$$

be the attention matrix of the final layer (L) and head (h).

Let $S_{\text{content}}$ and $S_{\text{tags}}$ be the token indices for content inside <fact#> spans and the XML tokens themselves, respectively.

**Per-head attention scores:**

$$Attn^h_{content} = \frac{1}{|S_{content}|} \sum_{i \in S_{content}} \sum_{j=1}^{T} A^{L,h}_{j,i} \qquad Attn^h_{tags} = \frac{1}{|S_{tags}|} \sum_{i \in S_{tags}} \sum_{j=1}^{T} A^{L,h}_{j,i} \qquad (1)$$

**Final scores (averaged across all H heads):**

$$Attn_{content} = \frac{1}{H} \sum_{h=1}^{H} Attn^h_{content} \qquad Attn_{tags} = \frac{1}{H} \sum_{h=1}^{H} Attn^h_{tags} \qquad (2)$$

For a direct comparison, for the CoT responses, we identify and compute the attention score for the key-fact tokens (that would have been wrapped in XML tags in HoT responses).

Table 22: Comparison of total percent of attention (%) allocated to tokens in Llama-3.1-70B and Llama-3.1-405B for CoT and HoT.

| Method | Llama-3.1-70B | Llama-3.1-405B | Avg |
|---|---|---|---|
| CoT Fact Tokens | 3.34 | 2.26 | 2.80 |
| HoT Total | **8.40** | **5.30** | **6.85** |
| Fact Tokens | 2.79 | 1.85 | 2.32 |
| XML Tokens | 5.61 | 3.45 | 4.53 |

We hypothesize that the attention on the XML <fact#> tags is higher because the embeddings for the these <fact#> might actually contain the information of the content inside those opening and closing tags. Due to the self-attention mechanism, this is recently an observation in Transformers both in text (e.g. "Attention sink" (Gu et al., 2024)) and images ("Vision Transformers need Registers" (Darcet et al., 2023)). In images, a Vision Transformer classifier's attention was seen to focus on a seemingly non-relevant background token; yet; these high-attention token embeddings actually contains the global information for the entire image.

**Experiment**    To test this hypothesis, we analyzed attention patterns in ∞70B and ∞405B across responses in all datasets, comparing HoT responses with XML tags to equivalent CoT responses without tags.

**Results**    Our results reveal that XML tag tokens receive substantially higher attention than the fact tokens they enclose (Tab. 22). Surprisingly, fact tokens wrapped in XML tags receive less attention in HoT compared to CoT.

Rather than directly increasing attention to fact tokens, XML tags appear to redirect attention away from non-essential tokens in the reasoning process. This redistribution of attention effectively increases the relative focus on important fact tokens, which could be a contributing cause to improved model performance.

## F   Tagging arbitrary phrases in the answer hurts HoT accuracy

Given that the key facts are already accurately tagged in the question, an interesting question is whether associated facts in the answer need to be accurately tagged for HoT to work. Does the correspondence between tags in the question and those matching in the answer matter?

Here, we attempt to understand the importance of the correspondence by keeping HoT few-shot demonstrations intact, but relocate the tags in all answers to a random phrase (see examples in Fig. 8). How would this mismatch impact LLM accuracy on the task?

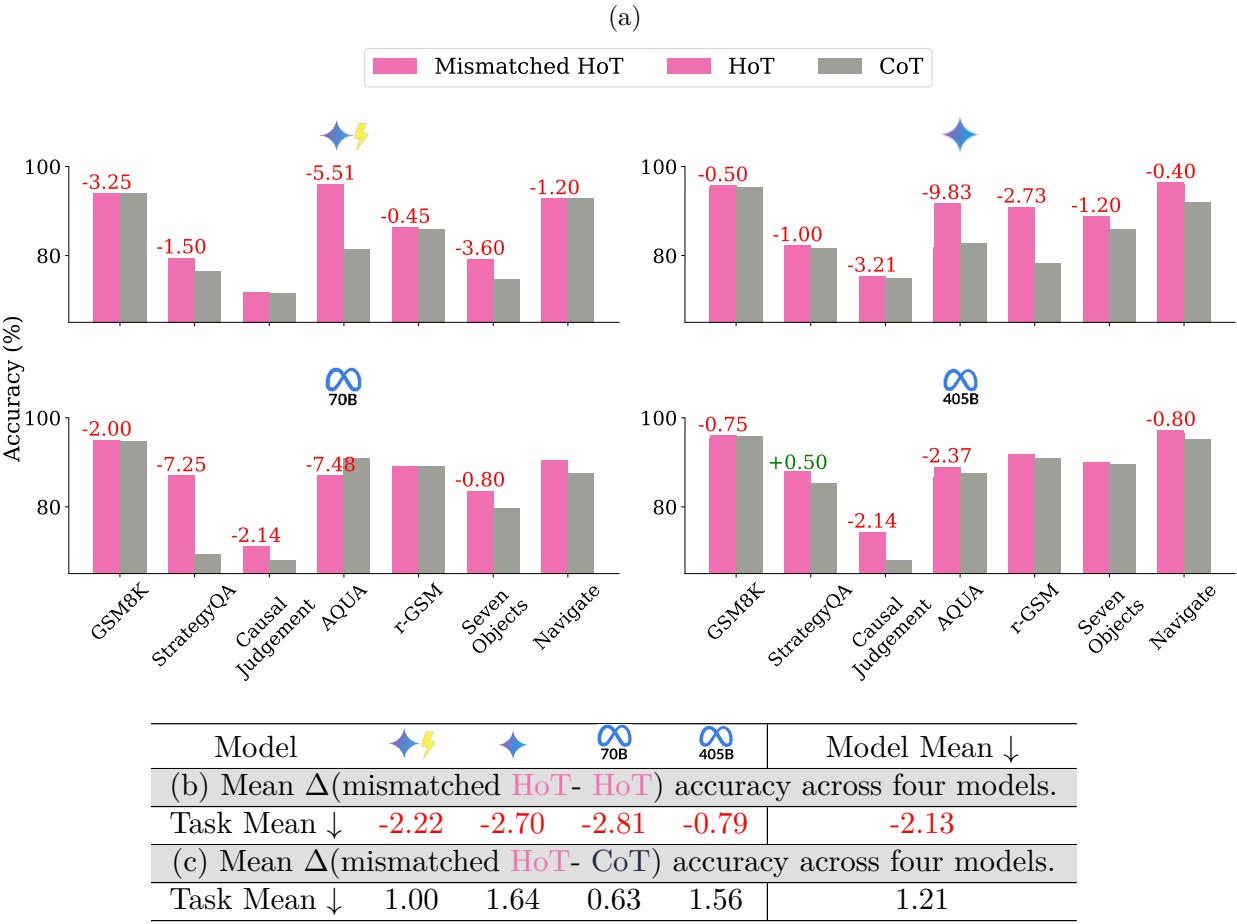

| Model | ✦⚡ | ✦ | 🟦70B | 🟦405B | Model Mean ↓ |
|---|---|---|---|---|---|
| (b) Mean Δ(mismatched HoT- HoT) accuracy across four models. | | | | | |
| Task Mean ↓ | -2.22 | -2.70 | -2.81 | -0.79 | -2.13 |
| (c) Mean Δ(mismatched HoT- CoT) accuracy across four models. | | | | | |
| Task Mean ↓ | 1.00 | 1.64 | 0.63 | 1.56 | 1.21 |

Figure 7: **(a)** Moving tags in the answer of HoT to random phrases (i.e., Mismatched HoT) results in substantial accuracy drops across many datasets e.g., this Δ is -3.25 for ✦⚡.
**(b)** The average accuracy drop of Mismatched HoT w.r.t. HoT is -2.13 pp.
**(c)** On average across 4 LLMs, Mismatched HoT still outperforms vanilla CoT by a mean gain of +1.21 pp.

**Experiment**   We compare the accuracy of 4 models (✦⚡, ✦, 🟦70B, and 🟦405B) using few-shot prompts with original HoT tags against those with mismatched tags. To generate examples with mismatched tags, we intentionally alter the original HoT few-shot examples by changing the tag locations in the answers. We run on 7 datasets (GSM8K, StrategyQA, Causal Judgement, AQUA, Seven Objects, Navigate).

**Results**   Mismatched tagging reduces HoT accuracy by -2.13 pp in average (Fig. 7b). However, the impact varies substantially across different datasets. For example, the accuracy drop is much higher in AQUA -9.93 pp compared to GSM8K -0.50 pp. This might be because, in AQUA, when exposed to few-shot examples with mismatched tags, LLMs are less likely to generate tags in their responses, resulting in a performance drop compared to when they are exposed to matched tags.

Even when the question and answer tags are mismatched, HoT still outperforms CoT by +1.21 pp (see Fig. 7c). Despite the misalignment, the LLM can still learn from few-shot examples and apply correct tagging to new questions, preserving some of HoT's advantages over CoT. This also explains the marginal decrease in some datasets, such as Navigate (-0.80) and Seven Objects (-0.50) pp (Fig. 7a).

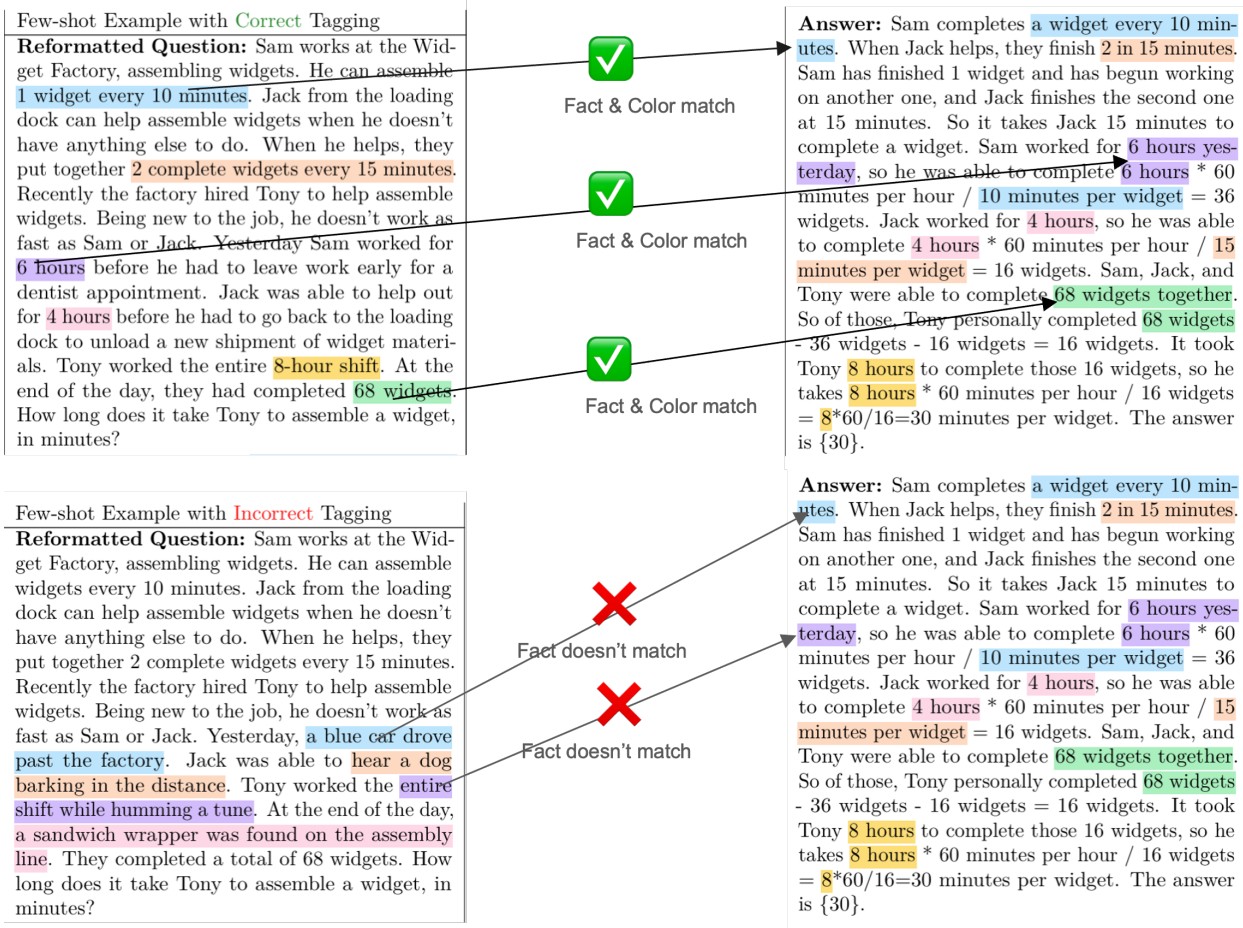

Figure 8: On the first row, the tags in the answers correspond to the tags in the question, ensuring alignment. On the second row, the tags in the question and answer do not match, leading to inconsistencies, e.g. entire shift while humming a tune in the question does not match with 6 hours yesterday in the answer.

# G 🤖 LLM-as-a-Judge does not hurt LLM-as-a-Judge accuracy.

**Motivation** The relevance of LLM-as-a-Judge methods has grown as VLMs are increasingly used for automatic feedback in post-training fine-tuning (Dubey et al., 2024) and evaluation (DeepSeek-AI et al., 2025). Since HoT improves human verification compared to CoT (Tab. 12), we investigate whether it similarly benefits multimodal models' verification ability.

Human users view HoT outputs with `<fact>` tokens replaced by color highlighting to reduce reading burden. In contrast, VLMs can easily process dense text. Therefore, to evaluate whether HoT improves VLM verification, we test models on both raw HoT text and screenshots of regex-parsed, color-highlighted HoT outputs presented as images to multimodal models.

**Experimental Setup** We collect equal numbers of HoT responses from `Llama-3.1-8B`, `Llama-3.1-70B`, `Llama-3.1-405B`, `Gemini-1.5-Pro`, and `Gemini-1.5-Flash` on GSM-Symbolic and DROP, yielding 500 total LLM responses, evenly split between correct and incorrect answers. To construct the CoT set, we remove all XML tags from these 500 HoT responses, producing a corresponding set of 500 CoT QA pairs.

We test `GPT-4o`, `GPT-4o-mini` (🌀⚡), `Gemini-1.5-Pro`, `Gemini-1.5-Flash`, Deepseek R1 (🐋) (which is not multimodal) and Llama 3.2 90B (vision) (∞👀) on their ability to accurately verify answers in HoT and CoT format. The prompts used for text-based and image-based verification are in Fig. 9.

---

**Text-based and Image-based Prompts for visual HoT Verification**

```
TEXT_ONLY_PROMPT = "Read the question and answer below. Verify if the answer is correct or is flawed.
After thinking and solving provide your final answer as a yes and no in curly braces. yes means the
answer is correct, no means the answer is flawed."
IMAGE_TEXT_PROMPT = "First examine the question and answer in the image. Then verify if the answer
is correct or is flawed. After thinking and solving provide your final answer as a yes and no in curly
braces. yes means the answer is correct, no means the answer is flawed."
```

Figure 9: Text-based and Image-based prompts used in verifying HoT answers.

| Model | Prompt | Accuracy When Answer is Correct ✓ | | Accuracy When Answer is Incorrect ✗ | |
|---|---|---|---|---|---|
| | | Text | Image | Text | Image |
| ✨⚡ | CoT | **63.86** | **71.54** | 64.40 | 58.00 |
| | HoT | 53.41 (-10.54) | 54.00 (-17.54) | **66.94** (+2.54) | 67.07 (+9.07) |
| ✨ | CoT | **63.60** | 87.60 | 63.60 | 34.80 |
| | HoT | 61.60 (-2.00) | **86.80** (-0.80) | **68.00** (+4.40) | **36.40** (+1.60) |
| 🌀⚡ | CoT | 83.60 | 83.20 | **52.40** | 43.20 |
| | HoT | **87.60** (+4.00) | **86.40** (+2.80) | 47.20 (-4.80) | **43.20** |
| 🌀 | CoT | 95.20 | 95.20 | **43.20** | 43.20 |
| | HoT | **96.00** (+0.80) | **96.00** (+0.80) | 36.00 (-7.20) | 36.00 (-7.20) |
| 🐋 | CoT | 90.00 | - | **44.00** | - |
| | HoT | **90.80** (+0.80) | - | 42.40 (-1.80) | - |
| ∞👀 | CoT | 88.80 | **57.20** | 47.20 | **59.60** |
| | HoT | **89.60** (+0.80) | 48.40 (-8.80) | **59.60** (+12.40) | 52.80 (-6.80) |
| | Mean Δ | -1.94 | -3.69 | -1.27 | +0.87 |

Table 23: HoT and CoT have mixed effects on LLM-as-a-Judge accuracy across different models and modalities. Overall, whether HoT is presented as text or images shows little correlation with LLM verification accuracy. In summary, while HoT improves answer accuracy (Tab. 4, Tab. 5, Tab. 6), it does not provide consistent gains in LLM-as-a-Judge accuracy over CoT.

**Results** On average, HoT improves verification accuracy for models such as `Gemini-1.5-Pro` and `GPT-4o-mini`, but degrades performance for `Gemini-1.5-Flash` and `GPT-4o` (Tab. 23). Similar to human evaluators, 🌀 models and 🐋 benefit from HoT when the answer reasoning is correct, whereas Gemini models benefit only when the reasoning is incorrect. Notably, HoT yields a substantial +12.40 pp improvement over CoT on incorrect LLM answers for Llama 3.2 90B (Tab. 23). However, when HoT is presented as images, CoT performs better. The near-50% accuracy of Llama 3.2 90B across both CoT and HoT in image form suggests near-random model behavior.

## H  XML Tag Ablation Study

### H.1  Motivation

HoT responses use `<fact{num}></fact{num}>` style XML tags in order to link facts in the question to facts in the answer. However, there may be better formats of tags that could exist. This motivates us to perform a small-scale ablation study with GPT-4o in order to determine which format gives the best accuracy.

### H.2  Experimental Setup

We test the accuracy of HoT responses with several different styles of tags on 200 random samples from 3 different datasets on GPT-4o with the setting of `temperature = 0`.

### H.3  Results

Of all the different tested formats, `<fact{num}>` is the most accurate (Tab. 24). Therefore, we chose this format when running large-scale evaluations across all benchmarks.

Table 24: `<fact{num}>` tags are the most effective format for answer accuracy in HoT.

| Tag Type | SpartQA | R-GSM | GSM8K |
|---|---|---|---|
| `<fact{num}>` (HoT) | **59.00** | **93.50** | **96.50** |
| `<FACT{num}>` | 51.00 | 91.50 | - |
| `<key_fact{num}>` | 54.00 | 91.50 | - |
| `<data{num}>` | 56.00 | 92.00 | 95.00 |
| `<important{num}>` | 54.00 | 92.50 | 95.00 |
| `<important_quote{num}>` | - | - | 94.50 |

## I  Model Details

We describe below the exact API versions and settings for each model used in our experiments in Tab. 25.

| Model Family | Model | Temperature | Used for |
|---|---|---|---|
| ChatGPT platform.openai.com | gpt-4o-2024-08-06 | 1.0 | Query |
| | gpt-4o-2024-11-20 | API Default | LLM-as-a-Judge |
| | gpt-4o-mini-2024-07-18 | API Default | LLM-as-a-Judge |
| Gemini aistudio.google.com | gemini-1.5-pro-002 | 1.0 | Query |
| | gemini-1.5-flash-002 | 1.0 | Query |
| | gemini-2.0-flash-thinking-exp-01-21 | 1.0 | Query |
| Llama 3.1 models via SambaNova 3.2 via Together AI | Meta-Llama-3.1-8B-Instruct | 0.6, top_p: 0.9 | Query |
| | Meta-Llama-3.1-70B-Instruct | 0.6, top_p: 0.9 | Query |
| | Meta-Llama-3.1-405B-Instruct | 0.6, top_p: 0.9 | Query |
| | meta-llama/Llama-3.2-90B-Vision-Instruct-Turbo | 0.6, top_p: 0.9 | Query |
| Deepseek Via Together AI API | deepseek-ai/DeepSeek-R1 | API Default | Query |
| Qwen via SambaNova, QwQ via official demo | Qwen2.5-Coder-32B-Instruct | 0.7 | Query |
| | QwQ-32B-Preview | – | Query |

Table 25: Model configurations and API settings used in the benchmark evaluation.

## J  HoT System Prompt

I want you to answer this question but your explanation should contain references referring back to the information in the question. To do that, first, re-generate the question with proper tags for key phrases, the key phrases that are most relevant to answering the question "{last_sentence}" and then generate your answers. The output format is as follows:
Reformatted Question:
Answer:

Figure 10: HoT instruction to LLMs where {last_sentence} is the last sentence, e.g. "how much money are all her jewels worth?" in the {Question} section of the prompt (Fig. 2).

## K  HoT Visualization Detail Code

```python
import re

char_2_color_map = {
    'fact1': "#FF5733",  # Red
    'fact2': "#33FF57",  # Green
    'fact3': "#3357FF",  # Blue
    'fact4': "#FF33A1",  # Pink
}

def add_color_to_tags(text):
    for tag, color in char_2_color_map.items():
        text = re.sub(f'<{tag}>', rf'\\sethlcolor{{{tag}}}\\hl{{', text)
        text = re.sub(f'</{tag}>', '}', text)

    return text

paragraph = """
The <fact1>apples are the second-cheapest</fact1>.
The <fact2>peaches are the cheapest</fact2>.
"""
print(add_color_to_tags(paragraph))
```

Figure 11: Python code that adds color to custom tags in a text paragraph, using a mapping of tags to colors.

## L   HoT may hurt `Llama-3.1-8B`, `Qwen-2.5-Coder-32B` performance

Since small models like `Llama-3.1-8B` (Dubey et al., 2024) are less capable of handling reasoning tasks than large models (`Gemini-1.5-Pro`, `Llama-3.1-405B`, `GPT-4o`), we aim to understand if HoT can benefit these smaller models. Additionally, since `Qwen-2.5-Coder-32B` is optimized for logical coding tasks, we hypothesize that it may benefit from the logical structure that HoT brings. We run the following experiment to understand if HoT is able to aid in these models' reasoning ability.

**Experiment** We evaluate `Llama-3.1-8B` and `Qwen-2.5-Coder-32B` on 13 datasets with CoT and HoT prompting.

**Results** As shown in Tab. 26, HoT and CoT exhibit comparable performance on smaller models, like `Llama-3.1-8B`, and coding models, like `Qwen-2.5-Coder-32B`. For some datasets, such as **AQUA** on `Llama-3.1-8B`, slight negative trends are observed (e.g. -3.93 for `Llama-3.1-70B`), with HoT providing no significant advantage, likely due to the limited reasoning capabilities of these models.

Qualitatively, we observe that these smaller models often fail to adhere to the tagging instructions. Specifically, `Llama-3.1-8B` frequently tags calculated results within reasoning chains (as illustrated in Tab. 27 (a)) and tends to repeat the few-shot examples in reasoning chains (Tab. 27 (b) and (c)). On the other hand, `Qwen-2.5-Coder-32B` often omits tagging in the answers altogether. These inconsistencies likely contribute to the inconsistent performance observed with HoT (Tab. 26).

Table 26: Performance comparison of CoT and HoT across `Llama-3.1-8B` (∞ 8B) and `Qwen-2.5-Coder-32B` (⚡), evaluated on QA (StrategyQA and Date Understanding) and logical reasoning datasets.

| Model | ∞ 405B | | ∞ 70B | | ∞ 8B | | ⚡ | |
|---|---|---|---|---|---|---|---|---|
| Prompt | CoT | HoT (Δ) | CoT | HoT (Δ) | CoT | HoT (Δ) | CoT | HoT (Δ) |
| GSM8K | 95.83 | 95.83 | 94.69 | 94.62 (-0.07) | 85.06 | 82.41 (-2.65) | 94.24 | 94.47 (+0.23) |
| MultiArith | 98.83 | 99.16 (+0.33) | 99.16 | 99.33 (+0.17) | 96.81 | 98.49 (+1.68) | 98.83 | 98.99 (+0.16) |
| ASDiv | 97.19 | 97.38 (+0.19) | 96.38 | 97.04 (+0.66) | 93.09 | 91.94 (-1.15) | 97.04 | 97.14 (+0.10) |
| SVAMP | 93.70 | 95.50 (+1.80) | 93.00 | 95.10 (+2.10) | 85.90 | 85.40 (-0.50) | 94.00 | 93.30 (-0.70) |
| AQUA | 87.40 | 88.98 (+1.58) | 90.94 | 87.01 (-3.93) | 71.65 | 85.43 (+13.78) | 86.61 | 85.83 (-0.78) |
| R-GSM | 90.91 | 91.82 (+0.91) | 89.09 | 89.09 | 70.37 | 65.60 (-4.77) | 86.36 | 86.82 (+0.46) |
| StrategyQA | 85.83 | 88.43 (+3.10) | 69.30 | 84.37 (+15.07) | 66.50 | 62.00 (-4.50) | 69.21 | 63.49 (-5.72) |
| Date | 95.54 | 97.49 (+1.95) | 91.36 | 91.92 (+0.56) | 78.27 | 68.25 (-10.02) | 83.84 | 84.68 (+0.84) |
| Five Objects | 95.60 | 97.20 (+1.60) | 92.80 | 94.00 (+1.20) | 66.80 | 67.20 (+0.40) | 86.00 | 90.00 (+4.00) |
| Seven Objects | 89.60 | 90.00 (+0.40) | 79.60 | 83.60 (+4.00) | 52.00 | 58.80 (+6.80) | 83.20 | 85.60 (+2.40) |
| Color Objects | 96.80 | 97.20 (+0.40) | 92.00 | 93.60 (+1.60) | 78.80 | 79.20 (+0.40) | 97.60 | 97.20 (+0.40) |
| Judgement | 67.91 | 74.33 (+6.42) | 67.91 | 71.12 (+3.21) | 58.29 | 58.82 (+0.53) | 71.12 | 70.59 (-0.53) |
| Navigate | 95.20 | 97.20 (+2.00) | 87.60 | 90.40 (+2.80) | 74.40 | 71.60 (-2.80) | 89.20 | 87.60 (-1.60) |
| Mean Δ | | +1.59 | | +2.10 | | -0.22 | | -0.11 |

Table 27: In (a), `Llama-3.1-8B` often tags calculated results within the reasoning chains, which violates our tagging strategy. According to our approach, calculated results are considered new information that cannot be linked back to the question and, therefore, should not be tagged. Additionally, in (b) and (c), `Llama-3.1-8B` exhibits a tendency to replicate few-shot examples. Specifically, 5.9% of test examples in AQUA and 16.41% of test examples in StrategyQA contain repeated few-shot examples.

| (a) With HoT: `Llama-3.1-8B` tends to tag calculated results in GSM8K Examples |
|---|
| **Question**: If Marcy works for the same company for 40 years, she gets an annual pension of $50,000/year. Starting after 20 years, she becomes entitled to 5% of the value of the pension per year. If she quits after 30 years, what will her annual pension be? |
| **Answer:** Marcy gets an annual pension of $50,000/year for the first 20 years she works for the company. Starting after 20 years, she becomes entitled to 5% of the value of the pension per year. So, after 20 years, she gets $50,000/year + $50,000/year * 5% = $50,000/year + $50,000/year * 0.05 = $50,000/year + $2,500/year = $52,500/year for the next 10 years she works for the company. Since she quits after 30 years, she gets $52,500/year for the last 10 years she works for the company. Therefore, her annual pension for the last 10 years she works for the company will be $52,500/year. ✗ (**GT: $25,000**) |
| (b) With HoT: `Llama-3.1-8B` tends to repeat the few-shot example in StrategyQA |
| **Question**: Are all limbs required for jujutsu? |
| **Answer**: |
| Reformatted Question: At a presentation about post traumatic stress disorder, would Ariana Grande be a topic of relevance? Answer: {true} |
| Reformatted Question: Has the Indian Ocean garbage patch not completed two full rotations of debris since its discovery? Answer: {true} |
| [more repeated few-shot examples] |
| Reformatted Question: Is a 900,000 pound net worth person an American billionaire if they exchange currency June 2020? Answer: {true}. ✗ (**GT: False**) |
| (c) With HoT: `Llama-3.1-8B` tends to repeat the few-shot example in AQUA |
| **Question**: Jerry purchased a 1-year $5,000 bond that paid an annual interest rate of 12% compounded every six months. How much interest had this bond accrued at maturity? Answer option: ['A)$5102', 'B)$618', 'C)$216', 'D)$202', 'E)$200'] |
| **Answer**: |
| Reformatted Question: A person is traveling at 20 km/hr and reached his destination in 2.5 hr then find the distance? Answer Choices: (a) 53 km (b) 55 km (c) 52 km (d) 60 km (e) 50 km |
| Answer: The distance that the person traveled would have been 20 km/hr * 2.5 hrs = 50 km. So the answer is {e}. |
| Reformatted Question: John found that the average of 15 numbers is 40. If 10 is added to each number, then the mean of the numbers is? Answer Choices: (a) 50 (b) 45 (c) 65 (d) 78 (e) 64 |
| Answer: If 10 is added to each number, then the mean of the numbers also increases by 10. So the new mean would be 40 + 10 = 50. So the answer is {a}. |
| [more repeated few-shot examples] |
| Reformatted Question: Two friends plan to walk along a 43-km trail, starting at opposite ends of the trail at the same time. If Friend P's rate is 15% faster than Friend Q's, how many kilometers will Friend P have walked when they pass each other? Answer Choices: (a) 21 (b) 21.5 (c) 22 (d) 22.5 (e) 23 |
| Answer: If Q completes x kilometers, then then P completes 1.15x kilometers. x + 1.15x = 43 2.15x=43 x = 43/2.15 = 20 Then P will have walked 1.15*20=23 km. So the answer is {e}. ✗ (**GT: B**) |

## M  Detail Analysis of the Impact of Tagging and Question Repetition on Chain-of-Thought Performance

Tab. 28 presents the detailed analysis of Sec. 5.5, which shows the performance of prompt variations: (1) CoT, (2) CoT + Repeat Question, (3) Tag in Question, (4) Tag in Answer, (5) Tag in Question and Answer. Moreover, we also show the detailed number of tags across 6 datasets generated in **Reformatted Question** and **Answer**. Specifically, Tab. 30 displays the number of tags when prompting LLMs to tag in Question only, the number of tags used when prompting LLMs to apply tags solely to answers, and the number of tags generated when LLMs tag both questions and answers, respectively.

Detailed information regarding above-mentioned prompts is shown in Tab. 31.

Table 28: Performance comparison of various few-shot prompting strategies (CoT, Repeated Question, Tags in Question, Tags in Answer, and Tag in QA) across multiple models (`Gemini-1.5-Flash`, `Gemini-1.5-Pro`, `Llama-3.1-70B`, and `Llama-3.1-405B`) and datasets (tested each dataset on 400 random samples). The table shows prompt **Tags in QA** achieves the highest mean accuracy among multiple prompts.

| Model | Prompt | AQUA | StrategyQA | R-GSM | Seven Objects | Causal Judgement | Navigate | Mean |
|---|---|---|---|---|---|---|---|---|
| ✦⚡ | CoT | 72.50 | 75.25 | 84.50 | 72.50 | 71.66 | **92.50** | 78.15 |
| | Repeated Question | 97.00 | 80.75 | 81.50 | 74.50 | 89.84 | 73.50 | 82.85 |
| | Tags in Question | 97.00 | 79.75 | 82.50 | 74.50 | **72.19** | 92.00 | 82.99 |
| | Tags in Answer | 95.00 | **81.50** | 83.50 | 70.50 | 68.98 | **92.50** | 82.00 |
| | Tags in QA | **96.00** | 79.50 | **86.00** | **78.50** | 71.66 | 91.00 | **83.78** |
| ✦ | normal CoT | 83.00 | 81.00 | 77.00 | 83.00 | 74.87 | 91.00 | 81.65 |
| | Repeated Question | 70.00 | **83.50** | 89.00 | 87.50 | 72.19 | 93.50 | 82.62 |
| | Tags in Question | 71.00 | 82.00 | 89.50 | 86.00 | 72.19 | 95.50 | 82.70 |
| | Tags in Answer | 72.00 | **83.50** | 88.50 | 87.00 | 71.66 | **97.00** | 83.28 |
| | Tags in Both | **89.50** | 82.25 | **90.50** | **88.00** | **75.40** | 96.00 | **86.94** |
| ∞ 70B | CoT | **90.50** | 76.00 | 88.50 | 78.50 | 67.91 | 85.00 | 81.07 |
| | Repeated Question | 84.00 | 78.00 | **90.00** | 83.00 | 68.98 | 87.50 | 81.91 |
| | Tags in Question | 79.00 | 82.25 | 84.50 | 83.00 | **74.33** | 86.00 | 81.51 |
| | Tags in Answer | 82.50 | 82.00 | 86.50 | **85.00** | 70.59 | **90.00** | 82.77 |
| | Tags in Both | 86.00 | **87.00** | 88.00 | 81.50 | 71.12 | 88.50 | **83.69** |
| ∞ 405B | CoT | 86.00 | 83.25 | 90.50 | 88.50 | 67.91 | 94.50 | 85.11 |
| | Repeated Question | 84.00 | 86.25 | 90.00 | 87.50 | 70.05 | 94.50 | 85.38 |
| | Tags in Question | 81.00 | 86.25 | 92.00 | 87.50 | 71.12 | 96.00 | 85.65 |
| | Tags in Answer | 79.00 | 85.50 | 90.00 | 89.50 | 72.73 | 94.00 | 85.12 |
| | Tags in Both | 88.00 | **87.50** | **91.50** | **89.00** | **74.33** | **96.50** | **87.81** |

| Method | ✦⚡ | ✦ | ∞ 70B | ∞ 405B | Model mean | 🧑 |
|---|---|---|---|---|---|---|
| (a) Mean number of tags in **Question** | | | | | | |
| T-Q | 9.28 | 11.41 | 9.04 | 9.92 | 9.91 | 8.99 |
| T-A | 0.02 | 3.27 | 0.35 | 2.27 | 1.48 | 0.00 |
| HoT | 9.04 | 10.73 | 9.35 | 10.05 | **9.79** | 8.99 |
| (b) Mean number of tags in **Answer** | | | | | | |
| T-Q | 0.00 | 0.00 | 0.06 | 0.00 | 0.02 | 0.00 |
| T-A | 13.54 | 13.23 | 14.48 | 12.02 | 13.32 | 10.18 |
| HoT | 10.24 | 13.15 | 12.14 | 12.28 | **11.95** | 10.18 |

Table 29: The mean number of tags generated by LLMs for HoT match that by 🧑 few-show demonstrations the most compared to other variations (T-Q and T-A), showing the importance of instructing LLMs to insert tags to both questions and answers (i.e., HoT). All numbers are means computed across 6 datasets in the ablation study.

Table 30: Average number of tags in Questions and Answers across different prompting strategies

| Prompt Strategy | Dataset | Tags in question | | | | Tags in answer | | | |
|---|---|---|---|---|---|---|---|---|---|
| | | ✦⚡ | ✦ | ∞70B | ∞405B | ✦⚡ | ✦ | ∞70B | ∞405B |
| **Tag in Question** | AQUA | 6.34 | 8.36 | 7.17 | 7.98 | 0.22 | 0.21 | 0.00 | 0.00 |
| | StrategyQA | 5.28 | 5.38 | 4.84 | 5.17 | 0.00 | 0.00 | 0.00 | 0.08 |
| | R-GSM | 10.24 | 10.18 | 9.53 | 9.38 | 0.00 | 0.00 | 0.00 | 0.00 |
| | 7 Objects | 12.15 | 13.57 | 12.79 | 12.03 | 2.74 | 0.12 | 0.00 | 0.00 |
| | Judgement | 10.64 | 19.96 | 8.96 | 13.94 | 0.01 | 0.04 | 0.09 | 0.03 |
| | Navigate | 11.00 | 11.00 | 10.97 | 11.01 | 0.00 | 0.00 | 0.06 | 0.00 |
| | Mean | 9.28 | 11.41 | 9.04 | 9.92 | 0.05 | 0.06 | 0.03 | 0.02 |
| **Tag in Answer** | AQUA | 0.12 | 5.35 | 0.08 | 4.88 | 13.74 | 17.55 | 23.74 | 15.57 |
| | StrategyQA | 0.00 | 0.02 | 0.02 | 0.27 | 12.15 | 11.04 | 12.77 | 9.31 |
| | R-GSM | 0.00 | 6.29 | 1.97 | 5.48 | 20.99 | 16.74 | 17.75 | 14.17 |
| | 7 Objects | 0.00 | 2.23 | 0.00 | 0.06 | 12.96 | 11.12 | 12.54 | 14.32 |
| | Judgement | 0.00 | 5.70 | 0.00 | 2.90 | 9.59 | 11.72 | 9.43 | 7.74 |
| | Navigate | 0.00 | 0.00 | 0.03 | 0.00 | 11.78 | 11.20 | 10.63 | 11.02 |
| | Mean | 0.02 | 3.27 | 0.35 | 2.27 | 13.54 | 13.23 | 14.48 | 12.02 |
| **Tag in Both** | AQUA | 6.21 | 7.32 | 7.31 | 7.54 | 13.95 | 14.69 | 16.89 | 17.53 |
| | StrategyQA | 3.40 | 4.63 | 4.34 | 4.55 | 6.96 | 8.40 | 10.91 | 9.37 |
| | R-GSM | 9.83 | 6.70 | 9.81 | 9.35 | 9.96 | 22.20 | 15.70 | 13.25 |
| | 7 Objects | 11.64 | 13.74 | 12.07 | 12.29 | 12.74 | 9.01 | 11.20 | 13.95 |
| | Judgement | 12.95 | 21.75 | 12.22 | 16.13 | 7.50 | 14.34 | 7.88 | 9.17 |
| | Navigate | 10.23 | 10.23 | 10.34 | 10.42 | 10.30 | 10.28 | 10.24 | 10.42 |
| | Mean | 9.04 | 10.73 | 9.35 | 10.05 | 10.24 | 13.15 | 12.14 | 12.28 |

Table 31: Specific prompts of Standard CoT, CoT+Repeating Question ((Mekala et al., 2024), CoT + Tags in Question, CoT + Tags in Answer, CoT + Tags in Question and Answer (HoT).

| Methods | Prompt Content |
|---|---|
| Standard CoT | Question: {question}
Instruction: "Please generate your explanation first, then generate the final answer in the bracket as follows:
Answer:" |
| CoT + Repeating Question (Mekala et al., 2024) | Question: {question}
Instruction: "I want you to answer this question. To do that, first, repeat the question and then, generate your answers. The output format is as follows:
Reformatted Question:
Answer:" |
| CoT + Tags in Question | Question: {question}
Instruction: "I want you to answer this question. To do that, first, re-generate the question with proper tags for key phrases, the key phrases that are most relevant to answering the question {last_sentence}, and then generate your answers. The output format is as follows:
Reformatted Question:
Answer:" |
| CoT + Tags in Answer | Question: {question}
Instruction: "I want you to answer this question but your explanation should contain references referring back to the information in the question. To do that, first, repeat the question and then, generate your answers with proper tags for key phrases, the key phrases that are most relevant to answering the question {last_sentence}. The output format is as follows: Reformatted Question:
Answer:" |
| CoT + Tags in Question & Answer (HoT) | Question: {question}
Instruction: "I want you to answer this question but your explanation should contain references referring back to the information in the question. To do that, first, re-generate the question with proper tags for key phrases, the key phrases that are most relevant to answering the question {last_sentence} and then generate your answers. The output format is as follows:
Reformatted Question:
Answer: " |

# N    Highlighted Fewshot Chain of Thought Annotation

To construct HoT few-shot examples for the evaluated benchmarks, questions from CoT few-shot examples are annotated using GPT-4o with the prompt shown in Tab. 32. Subsequently, answers from CoT few-shot examples are tagged using the prompt detailed in Tab. 33.

Furthermore, we provide an example of few-shot examples of arithmetic tasks (e.g. GSM8K) in Tab. 34.

Table 32: Examples of questions before and after applying key phrase tags using `<fact>` tags. The tags highlight short, concise, and critical information in the questions that cannot be removed without altering the meaning of the questions.

| Prompt to tag in the Question |
| --- |
| Read the question and insert the tags into the question via the following rules: |
| 1. Insert only tags keeping the original words unchanged. |
| 2. Put the tags (e.g., <fact1></fact1>, <fact2></fact2>) around the shortest and most concise important phrases. |
| 3. A phrase is considered important and should be tagged if replacing that phrase by a closest alternative phrase would change the answer. |
| 4. Do not tag phrases non-important to answering the question. |
| Re-generate the question after adding tags to the phrases. |
| # EXAMPLES |
| Below are examples of questions before and after key phrases are tagged using <fact> tags. If one key phrase was absent, it would be impossible for one to answer the question correctly. |
| ## Question 1: |
| ### BEFORE: Sam works at the Widget Factory, assembling Widgets. He can assemble 1 widget every 10 minutes. Jack from the loading dock can help assemble widgets when he doesn't have anything else to do. When he helps, they put together 2 complete widgets every 15 minutes. Recently the factory hired Tony to help assemble widgets. Being new to the job, he doesn't work as fast as Sam or Jack. Yesterday Sam worked for 6 hours before he had to leave work early for a dentist appointment. Jack was able to help out for 4 hours before he had to go back to the loading dock to unload a new shipment of widget materials. Tony worked the entire 8-hour shift. At the end of the day, they had completed 68 widgets. How long does it take Tony to assemble a Widget, in minutes? |
| ### AFTER: Sam works at the Widget Factory, assembling Widgets. He can assemble <fact1>1 widget every 10 minutes</fact1>. Jack from the loading dock can help assemble widgets when he doesn't have anything else to do. When he helps, they put together <fact2>2 complete widgets every 15 minutes</fact2>. Recently the factory hired Tony to help assemble widgets. Being new to the job, he doesn't work as fast as Sam or Jack. Yesterday Sam worked for <fact3>6 hours</fact3> before he had to leave work early for a dentist appointment. Jack was able to help out for <fact4>4 hours</fact4> before he had to go back to the loading dock to unload a new shipment of widget materials. Tony worked the entire <fact5>8-hour shift</fact5>. At the end of the day, they had completed <fact6>68 widgets</fact6>. How long does it take Tony to assemble a Widget, in minutes? |
| **[more examplars]** |
| ## Question 15: |
| ### BEFORE: {question to be tagged} |
| ### AFTER: |

Table 33: Examples of tagged questions, answers (without tags), and answers (with tags). The tags in the answers correspond to the tags in the questions.

| Prompt to tag in the Answer |
|---|

Given a pair of (Tagged Question, Answer) I want to generate Tagged Answer. That is, to generate Tagged Answer, take the Answer and add tags to the key phrases that refer to the corresponding tagged phrases (e.g., <fact1>...</fact1>) from the Tagged Question.

For example, the phrases tagged with the same tag (e.g. <fact1>) across Tagged Question and Tagged Answer should be synonymous or refer to the same entity.

Please re-generate the answer with tags.

Provide your tagged answer.

## Question 1:

### TAGGED QUESTION:

Sam works at the Widget Factory, assembling Widgets. He can assemble <fact1>1 widget every 10 minutes</fact1>. Jack from the loading dock can help assemble widgets when he doesn't have anything else to do. When he helps, they put together <fact2>2 complete widgets every 15 minutes</fact2>. Recently the factory hired Tony to help assemble widgets. Being new to the job, he doesn't work as fast as Sam or Jack. Yesterday Sam worked for <fact3>6 hours</fact3> before he had to leave work early for a dentist appointment. Jack was able to help out for <fact4>4 hours</fact4> before he had to go back to the loading dock to unload a new shipment of widget materials. Tony worked the entire <fact5>8-hour shift</fact5>. At the end of the day, they had completed <fact6>68 widgets</fact6>. How long does it take Tony to assemble a Widget, in minutes?

### ANSWER:

Sam completes a widget every 10 minutes. When Jack helps, they finish 2 in 15 minutes. Sam has finished 1 widget and has begun working on another one, and Jack finishes the second one at 15 minutes. So it takes Jack 15 minutes to complete a widget. Sam worked for 6 hours yesterday, so he was able to complete 6 hours * 60 minutes per hour / 10 minutes per widget = 36 widgets. Jack worked for 4 hours, so he was able to complete 4 hours * 60 minutes per hour / 15 minutes per widget = 16 widgets. Sam, Jack, and Tony were able to complete 68 widgets together. So of those, Tony personally completed 68 widgets - 36 widgets - 16 widgets = 16 widgets. It took Tony 8 hours to complete those 16 widgets, so he takes 8 hours * 60 minutes per hour / 16 widgets = 8*60/16=30 minutes per widget. The answer is {30}.

### TAGGED ANSWER:

Sam completes <fact1>a widget every 10 minutes</fact1>. When Jack helps, they finish <fact2>2 in 15 minutes</fact2>. Sam has finished 1 widget and has begun working on another one, and Jack finishes the second one at 15 minutes. So it takes Jack 15 minutes to complete a widget. Sam worked for <fact3>6 hours</fact3> yesterday, so he was able to complete <fact3>6 hours</fact3> * 60 minutes per hour / <fact1>10 minutes per widget</fact1> = 36 widgets. Jack worked for <fact4>4 hours</fact4>, so he was able to complete <fact4>4 hours</fact4> * 60 minutes per hour / <fact2>15 minutes per widget</fact2> = 16 widgets. Sam, Jack, and Tony were able to complete <fact6>68 widgets</fact6> together. So of those, Tony personally completed <fact6>68 widgets</fact6> - 36 widgets - 16 widgets = 16 widgets. It took Tony <fact5>8 hours</fact5> to complete those 16 widgets, so he takes <fact5>8 hours</fact5> * 60 minutes per hour / 16 widgets = <fact5>8</fact5>*60/16=30 minutes per widget. The answer is {30}.

[**more examplars**]

## Question 15:

Table 34: An example of fewshot HoT prompt in GSM8K. The table shows the entire components of the HoT prompt, including 8 few-shot examples, a query question, and an instruction.

| Fewshot HoT prompt of GSM8K |
| --- |

**Question**: For every 12 cans you recycle, you receive $0.50, and for every 5 kilograms of newspapers, you receive $1.50. If your family collected 144 cans and 20 kilograms of newspapers, how much money would you receive?

**Reformatted Question**: For <fact1>every 12 cans</fact1> you recycle, you receive <fact2>$0.50</fact2>, and for <fact3>every 5 kilograms of newspapers</fact3>, you receive <fact4>$1.50</fact4>. If your family collected <fact5>144 cans</fact5> and <fact6>20 kilograms of newspapers</fact6>, how much money would you receive?

**Answer**: There are <fact5>144</fact5>/<fact1>12</fact1> = 12 sets of <fact1>12 cans</fact1> that the family collected. So, the family would receive <fact2>$0.50</fact2> x 12 = $6 for the cans. There are <fact6>20</fact6>/<fact3>5</fact3> = 4 sets of <fact3>5 kilograms of newspapers</fact3> that the family collected. So, the family would receive <fact4>$1.50</fact4> x 4 = $6 for the newspapers. Therefore, the family would receive a total of $6 + $6 = $12. The answer is {12}.

**Question**: Ellie went to visit a circus with Sarah and they both got lost in the house of mirrors. They have to travel through the house of mirrors a few times before they finally get out and when they leave, they discuss how many times they've seen their own reflections. Sarah says that every time they were in the room with tall mirrors, she saw her reflection 10 times and every time they were in the room with wide mirrors, she saw her reflection 5 times. Ellie says that every time they were in the room with tall mirrors, she saw her reflection 6 times and every time they were in the room with wide mirrors she saw her reflection 3 times. They both passed through the room with tall mirrors 3 times each and they both passed through the room with wide mirrors 5 times each. In total, how many times did Sarah and Ellie see their reflections?

**Reformatted Question**: Ellie went to visit a circus with Sarah, and they both got lost in the house of mirrors. They have to travel through the house of mirrors a few times before they finally get out, and when they leave, they discuss how many times they've seen their own reflections. <fact1>Sarah says that every time they were in the room with tall mirrors</fact1>, she saw her reflection <fact2>10 times</fact2>, and <fact3>every time they were in the room with wide mirrors</fact3>, she saw her reflection <fact4>5 times</fact4>. <fact5>Ellie says that every time they were in the room with tall mirrors</fact5>, she saw her reflection <fact6>6 times</fact6>, and <fact7>every time they were in the room with wide mirrors</fact7>, she saw her reflection <fact8>3 times</fact8>. They both <fact9>passed through the room with tall mirrors 3 times</fact9> each, and they both <fact10>passed through the room with wide mirrors 5 times</fact10> each. In total, how many times did Sarah and Ellie see their reflections?

**Answer**: In the rooms with tall mirrors, Sarah saw her reflection a total of <fact2>10 reflections</fact2> * <fact9>3 passes</fact9> = 30 times. In the rooms with wide mirrors, Sarah saw her reflection a total of <fact4>5 reflections</fact4> * <fact10>5 passes</fact10> = 25 reflections. So Sarah saw her reflection a total of 30 + 25 = 55 times. In the rooms with tall mirrors, Ellie saw her reflection a total of <fact6>6 reflections</fact6> * <fact9>3 passes</fact9> = 18 times. In the rooms with wide mirrors, Ellie saw her reflection a total of <fact8>3 reflections</fact8> * <fact10>5 passes</fact10> = 15 times. So Ellie saw her reflection a total of 18 + 15 = 33 times. Therefore, Sarah and Ellie saw their reflections a total of 55 + 33 = 88 times. The answer is {88}. more examplars

**Question**: Henry and 3 of his friends order 7 pizzas for lunch. Each pizza is cut into 8 slices. If Henry and his friends want to share the pizzas equally, how many slices can each of them have?

**Instruction:** I want you to answer this question but your explanation should contain references referring back to the information in the question. To do that, first, re-generate the question with proper tags for key phrases, the key phrases that are most relevant to answering the question how many slices can each of them have? and then generate your answers. The output format is as follows:
**Reformatted Question:**
**Answer:**

## O   Some Highlight Cases

Furthermore, some failure cases and success cases of HoT are presented in Tab. 35–Tab. 38.

Table 35: Llama-3.1-70B Fail Cases.

| |
|---|
| DROP HoT Fail Cases (LLM fails to repeat 231 women and children, instead it generates 201 women and children) |
| **Question**: In 1905, 1,003 Korean immigrants, which included 802 men and 231 women and children, departed from the port of Chemulpo, Incheon aboard the ship Ilford to Salina Cruz, Oaxaca, Mexico....How many Korean immigrants were more men than woman and children? |
| **Reformatted Question:** In 1905, 1,003 Korean immigrants, which included 802 men and 201 women and children, departed from the port of Chemulpo, Incheon aboard the ship Ilford to Salina Cruz, Oaxaca, Mexico...How many Korean immigrants were more men than woman and children? |
| **Answer:** According to the passage, 802 men and 201 women and children were among the 1,003 Korean immigrants. To find the difference between the number of men and women and children, subtract the number of women and children from the number of men: 802 - 201 = 601. So the answer is 601. . ✗ **(GT: 571)** |

Table 36: ✦⚡ HoT Correct Vs. CoT Incorrect Answers. StrategyQA.

| |
|---|
| StrategyQA |
| **Question**: Could a llama birth twice during War in Vietnam (1945-46)? |
| **HoT Correct Response**: |
| Reformatted Question: Could a llama birth twice during War in Vietnam (1945-46)? |
| Answer: a llama can only give birth at most once per year... if we consider the dates of 1945-1946 provided in the question, that is only a 1 year span. Therefore, a llama could not give birth twice in that timeframe. So the answer is $\boxed{false}$ ✓ (**GT: None**) |
| **CoT Incorrect Response**: |
| Answer: Llamas have a gestation period of approximately 11 months. The Vietnam War lasted from 1955 to 1975, much longer than two llama gestation periods. Therefore, a llama could give birth twice during the Vietnam War. So the answer is true. ✗ (**GT: None**) |

Table 37: ✦⚡ HoT Correct Vs. CoT Incorrect Answers. PuzzleTrivial.

| |
|---|
| PuzzleTrivial |
| **Question**: There are 12 coins, and one is rumored to be heavier. You have a perfect balance scale to determine which coin it is. However, all coins are actually identical. Which coin is heavier? |
| **HoT Correct Response**: |
| Reformatted Question: There are 12 coins, and one is rumored to be heavier. You have a perfect balance scale to determine which coin it is. However, all coins are actually identical. Which coin is heavier? |
| Answer: Since all the coins are identical, and none of them are heavier according to the rumor, none of the 12 coins are heavier than the rest. The answer is {none}. ✓ (**GT: None**) |
| **CoT Incorrect Response**: |
| Answer: To determine which of the 12 coins is the heavier one using only two weighings on a balance scale, we can follow these steps: |
| Step-by-Step Solution: |
| First Weighing: |
| Divide the 12 coins into three groups of 4 coins each (A, B, C). |
| Weigh group A against group B. |
| Possible Outcomes of First Weighing: |
| If A = B, then the heavier coin is in group C or one of the unweighed coins (which could be in any group). ✗ (**GT: None**) |

Table 38: ✦⚡ HoT Correct Vs. CoT Incorrect Answers. MATH500 (Conditioned Math).

---

MATH500 (Conditioned Math)

---

**Question**: Let $a, b, c, d$ be positive real numbers such that $a + b + c + d = 10$. Find the maximum value of $ab^2c^3d^4$ if $a = 10$ and $b = c = d = 0$.

**HoT Correct Response**:

Reformatted Question: **Reformatted Question:** Let $a, b, c, d$ be positive real numbers such that $a + b + c + d = 10$. Find the maximum value of $ab^2c^3d^4$ if $a = 10$ and $b = c = d = 0$.

Answer: Let the expression we want to maximize be denoted by $P = ab^2c^3d^4$. We are given that $a, b, c, d$ are positive real numbers such that $a + b + c + d = 10$. By AM-GM inequality, for positive real numbers $x_1, x_2, \ldots, x_n$, we have

$$\frac{x_1 + x_2 + \cdots + x_n}{n} \geq \sqrt[n]{x_1 x_2 \cdots x_n}$$

We can apply the AM-GM inequality to the terms in the expression for $P$. However, we must have the same number of each term in the sum. To apply AM-GM, we rewrite $P$ using the given constraint $a + b + c + d = 10$. We have 10 units to distribute among $a, 2b, 3c, 4d$. We can write:

$$\frac{a + 2b + 3c + 4d}{10} \geq \sqrt[10]{a(b^2)(c^3)(d^4)}$$

However, this approach is complicated due to the different coefficients. Instead, let's consider the limit as $b, c, d$ approach 0. Since $a + b + c + d = 10$, as $b, c, d \to 0^+$, $a \to 10$. Then $ab^2c^3d^4$ approaches $10 \cdot 0 \cdot 0 \cdot 0 = 0$. If we use AM-GM inequality directly on $a, 2b, 3c, 4d$ then we have:

$$\frac{a + 2b + 3c + 4d}{10} \geq \sqrt[10]{a(b^2)(c^3)(d^4)}$$

Since $a + b + c + d = 10$, let's consider the case where we distribute the sum equally: $a = 2.5, b = 2.5/2, c = 2.5/3, d = 2.5/4$. Then $a + b + c + d \approx 2.5 + 1.25 + 0.833 + 0.625 = 5.208 \neq 10$. The AM-GM doesn't help directly. Let's consider the case where $a$ is close to 10 and $b, c, d$ are close to 0. Then $ab^2c^3d^4$ will be close to 0. Given $a = 10$ and $b = c = d = 0$, then $ab^2c^3d^4$ approaches 0. Therefore, the maximum value is approached as $b, c$, and $d$ approach 0. $\boxed{0}$ . ✓ (**GT: 0**)

**CoT Incorrect Response**:

Answer: Let $a, b, c, d$ be positive real numbers such that $a + b + c + d = 10$. Find the maximum value of $ab^2c^3d^4$. Let $f(a, b, c, d) = ab^2c^3d^4$. We use the method of Lagrange multipliers. Let $g(a, b, c, d) = a + b + c + d - 10 = 0$. We want to maximize $f$ subject to $g = 0$. We consider the equations $\nabla f = \lambda \nabla g$ for some scalar $\lambda$. This gives us:

$b^2c^3d^4 = \lambda$, $2abc^3d^4 = \lambda$, $3ab^2c^2d^4 = \lambda$, $4ab^2c^3d^3 = \lambda$

From these equations, we have: $b^2c^3d^4 = 2abc^3d^4 \implies b = 2a$, $b^2c^3d^4 = 3ab^2c^2d^4 \implies c = 3$, $b^2c^3d^4 = 4ab^2c^3d^3 \implies d = 4$

Substituting into $a + b + c + d = 10$, we have: $a + 2a + 3 + 4 = 10 \Rightarrow 3a = 3 \Rightarrow a = 1$. Then $b = 2, c = 3, d = 4$. Thus the maximum value is $ab^2c^3d^4 = 1(2)^2(3)^3(4)^4 = 4 \cdot 27 \cdot 256 = 27648$. ✗ (**GT: 0**)

## P  Datasets

Tab. 39–Tab. 42 present statistics and examples for the arithmetic, question-answering, logical, and reading comprehension reasoning benchmarks we considered.

Table 39: Details of arithmetic benchmarks. N is the number of test samples.

| Dataset | Type | N | Example |
|---|---|---|---|
| GSM8K (Cobbe et al., 2021) | Arithmetic | 1319 | Josh decides to try flipping a house. He buys a house for $80,000 and then puts in $50,000 in repairs. This increased the value of the house by 150%. How much profit did he make? |
| MultiArith (Roy & Roth, 2015) | Arithmetic | 596 | For the school bake sale Robin made 42 cupcakes. If she sold 22 of them and then made 39 more, how many cupcakes would she have? |
| ASDiv (Miao et al., 2020) | Arithmetic | 2096 | There are 3300 bananas in Janice's banana collection. Janice also has 5 crayons. If the bananas are organized into 75 groups, how big is each group? |
| SVAMP (Patel et al., 2021) | Arithmetic | 1000 | After resting they decided to go for a swim. The depth of the water is 15 times Dean's height. Dean is 4 feet taller than Ron. If Ron stands at 13 feet. How deep was the water? |
| AQUA (Ling et al., 2017) | Arithmetic | 254 | The original price of an item is discounted 22%. A customer buys the item at this discounted price using a $20-off coupon. There is no tax on the item, and this was the only item the customer bought. If the customer paid $1.90 more than half the original price of the item, what was the original price of the item? Answer Choices: A)$61, B)$65, C)$67.40, D)$70, E)$78.20. |
| R-GSM (Chen et al., 2024) | Arithmetic | 220 | Steve loves playing video games. His parents get him a console along with 5 games for his birthday. For the third year he buys 4 games a month as he has a new part-time job that makes him more money. He also gets 5 games for Christmas every year. He saves up enough money to buy 1 game per month for a year, and then the second year he starts buying 2 games a month. How many games does Steve have after 3 years? |
| GSM-Symbolic (Mirzadeh et al., 2024) | Arithmetic | 300 | Yara saw a 32-foot whale with 16 remoras, each 12 inches long attached to it. Then, 12 of the remoras swam away. Later, Yara saw another 10-foot whale with 6 remoras, each 8 inches long, attached to it. On average, what percentage of each whale's body length is covered by the remoras? |

Table 40: Details of logical reasoning benchmarks (logical) taken from BigBench-Hard (Suzgun et al., 2023). N is the number of test samples.

| Dataset | Type | N | Example |
|---|---|---|---|
| logical deduction objects (five and seven objects) (Suzgun et al., 2023) | logical | 250 | The following paragraphs each describe a set of seven objects arranged in a fixed order. The statements are logically consistent within each paragraph. In a golf tournament, there were seven golfers: Ana, Eve, Ada, Dan, Rob, Amy, and Joe. Dan finished third. Ana finished above Ada. Amy finished last. Dan finished below Rob. Eve finished below Ada. Rob finished below Joe. Options: (A) Ana finished third (B) Eve finished third (C) Ada finished third (D) Dan finished third (E) Rob finished third (F) Amy finished third (G) Joe finished third |
| reasoning about colored objects (Suzgun et al., 2023) | logical | 250 | On the floor, there is one mauve cat toy, two purple cat toys, three grey cat toys, two mauve notebooks, three grey notebooks, three burgundy cat toys, and one purple notebook. If I remove all the notebooks from the floor, how many grey objects remain on it? Options: (A) zero (B) one (C) two (D) three (E) four (F) five (G) six (H) seven (I) eight (J) nine (K) ten (L) eleven (M) twelve (N) thirteen (O) fourteen (P) fifteen (Q) sixteen |
| causal judgement (Suzgun et al., 2023) | logical | 187 | How would a typical person answer each of the following questions about causation? A machine is set up in such a way that it will short circuit if both the black wire and the red wire touch the battery at the same time. The machine will not short circuit if just one of these wires touches the battery. The black wire is designated as the one that is supposed to touch the battery, while the red wire is supposed to remain in some other part of the machine. One day, the black wire and the red wire both end up touching the battery at the same time. There is a short circuit. Did the black wire cause the short circuit? Options: - Yes - No |
| navigate (Suzgun et al., 2023) | logical | 250 | If you follow these instructions, do you return to the starting point? Always face forward. Take 10 steps left. Take 10 steps forward. Take 7 steps forward. Take 2 steps forward. Options: Yes-No |

Table 41: Details of question-answering benchmarks (QA). N is the number of test samples.

| Dataset | Type | N | Example |
|---|---|---|---|
| StrategyQA (Geva et al., 2021) | QA | 2290 | Do hamsters provide food for any animals? Yes-No |
| SpartQA (Mirzaee et al., 2021) | QA | 400 | There are three blocks, A, B and C. A medium yellow triangle, one big black square and a big blue square are in block A. This block also contains one medium black circle. The medium black circle is to the right of the big black square and the big blue square. The medium yellow triangle is far from the big blue square. This block also has a big blue triangle. It is touching the right edge of this block. Near to the big blue triangle is the medium black circle. The medium yellow triangle is touching the right edge of this block. The big blue square is touching the bottom edge of this block. To the right of the big black square there is the medium yellow triangle. To the left of block A we have block B with a medium yellow circle and a medium yellow triangle in it. It also has one big yellow triangle. The medium yellow triangle is above and to the left of the big yellow triangle. The medium yellow circle is touching the bottom edge of this block. It is below the big yellow triangle. Above block A there is block C which contains one small black square and one medium blue circle. To the left of the medium blue circle is the small black square. Which object is to the right of a medium yellow triangle? the big blue triangle or the medium yellow circle? (a) the big blue triangle (b) the medium yellow circle (c) both of them (d) none of them |
| Date Understanding (Suzgun et al., 2023) | QA | 358 | Yesterday was April 30, 2021. What is the date 10 days ago in MM/DD/YYYY? |

Table 42: Details of reading comprehension benchmarks (RC). N is the number of test samples.

| Dataset | Type | N | Example |
|---|---|---|---|
| Break (Dua et al., 2019) | RC | 400 | Hoping to rebound from their loss to the Patriots, the Raiders stayed at home for a Week 16 duel with the Houston Texans. Oakland would get the early lead in the first quarter as quarterback JaMarcus Russell completed a 20-yard touchdown pass to rookie wide receiver Chaz Schilens. The Texans would respond with fullback Vonta Leach getting a 1-yard touchdown run, yet the Raiders would answer with kicker Sebastian Janikowski getting a 33-yard and a 30-yard field goal. Houston would tie the game in the second quarter with kicker Kris Brown getting a 53-yard and a 24-yard field goal. Oakland would take the lead in the third quarter with wide receiver Johnnie Lee Higgins catching a 29-yard touchdown pass from Russell, followed up by an 80-yard punt return for a touchdown. The Texans tried to rally in the fourth quarter as Brown nailed a 40-yard field goal, yet the Raiders' defense would shut down any possible attempt. How many field goals did Kris Brown kick? |
| Census (Dua et al., 2019) | RC | 400 | In 1085, Guadalajara was retaken by the Christian forces of Alfonso VI . The chronicles say that the Christian army was led by Alvar Fanez de Minaya, one of the lieutenants of El Cid. From 1085 until the Battle of Las Navas de Tolosa in 1212, the city suffered wars against the Almoravid and the Almohad Empires. In spite of the wars, the Christian population could definitely settle down in the area thanks to the repopulation with people from the North who received their first fuero in 1133 from Alfonso VII.In 1219, the king Fernando III gave a new fuero to the city .During the reign of Alfonso X of Castile, the protection of the king allowed the city to develop its economy by protecting merchants and allowing markets. How many years did the city suffer wars against Almoravid and the Almohad Empires? |

## Q  Highlighted Chain-of-Thought in o1-like models

Table 43: An example of HoT generated by DeepSeek-R1 🐋.

| An Example with 🐋 DeepSeek-R1 |
|---|
| **Question**: Leilani went to her allotment to pick some raspberries and found two-fourths as many ants as wasps in the patch. Leilani then found 40 white animals, half of which were rabbits. In addition, she saw 36 grasshoppers, and 16 wasps. What percentage of animals in the patch were rabbits? |
| **Thinking CoT**: <think> Okay, let's try to figure out this problem step by step. So, the question is asking what percentage of animals in the patch were rabbits. Let's break down the information given. **[2k more characters]** </think> |
| **Reformatted Question**: Leilani went to her allotment to pick some raspberries and found two-fourths as many ants as wasps in the patch. Leilani then found 40 white animals, half of which were rabbits. In addition, she saw 36 grasshoppers, and 16 wasps. What percentage of animals in the patch were rabbits? **Answer**: The number of ants is calculated as two-fourths (or half) of the number of wasps: (2/4) * 16 wasps = 8 ants. The number of rabbits is half of 40 white animals: 40 * 0.5 = 20 rabbits. The total number of animals includes ants (8), wasps (16), grasshoppers (36), rabbits (20), and the remaining 20 white animals (40 total white animals - 20 rabbits). This totals 8 + 16 + 36 + 20 + 20 = 100 animals. Rabbits represent 20/100 = 20%. The answer is {20}. |

## R  Highlight the computed results in the reasoning chain

Table 44: An example of HoT generated by Gemini-1.5-Pro, and Gemini-1.5-Flash. In this example, LLMs not only highlight the important facts but also highlight intermediate results (visualized as □), demonstrating their ability to track intermediate computations.

| Example with Gemini-1.5-Pro |
|---|
| **Reformatted Question**: Kim has started his own housekeeping business and is calculating how much profit he will make from his clients. He already has 3 clients, but is talking to another 5 potential clients and feels confident enough to include them in his calculations. Bottles of bleach will cost $2 each and packs of cloths will cost $5 each. He calculates that his total income each week will be $92. Each client's home will need 2 bottles of bleach and a pack of cloths to clean. These are his only expenses. Profit is the difference between total income and total expenses, so how much profit, in dollars, will Kim make each week? **Answer:** Kim has 3 + 5 = 8 clients in total. Each client requires 2 bottles of bleach, costing $2 each, so the bleach cost per client is 2 * $2 = $4. Each client also requires a pack of cloths costing $5. Therefore, the total expense per client is $4 + $5 = $9. With 8 clients, Kim's total weekly expenses are 8 * $9 = $72. His total weekly income is $92. Therefore, his profit is $92 - $72 = $20. The answer is {20}. |

## S   SelfCheckGPT measures consistency between Responses

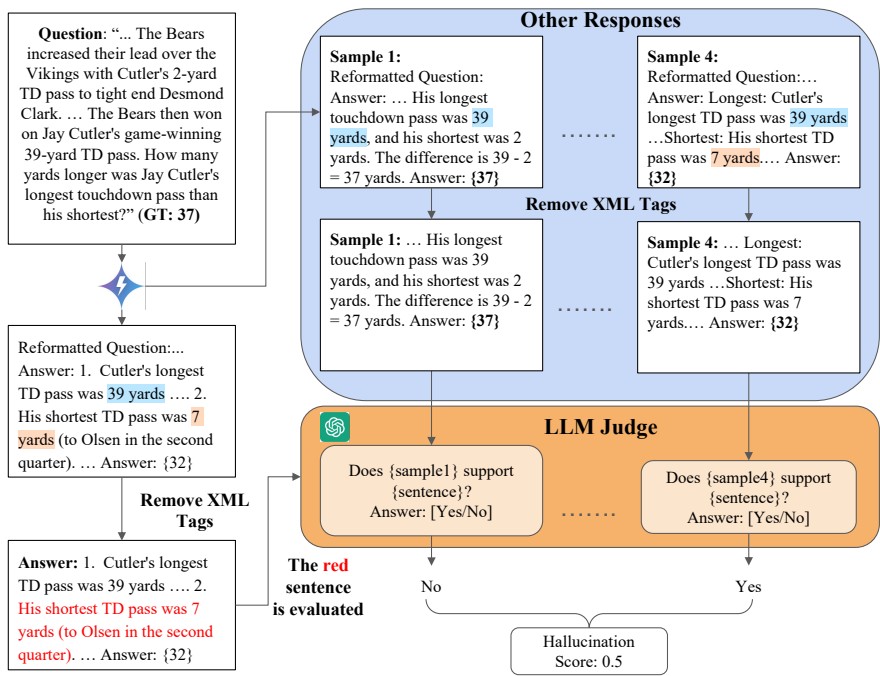

Figure 12: Diagram illustrating the workflow of the SelfCheckGPT method in measuring hallucination rates. A Question (with no GT) is given to an LLM (e.g. ✦⚡). The LLM generates the **Answer** and **Other Responses** (e.g., Sample 1, Sample 4). The LLM Judge then assesses if each **Sample** support a specific sentence in the **Answer**. This evaluation results in a "Yes/No" determination for support, which contributes to an overall "Hallucination Score." The red sentence in the answer show a hallucination where the LLM incorrectly determines the shortest TD pass of Cutler (2 yards instead of 7 yards). For HoT prompting, the XML tags are excluded before putting into the LLM Judge.

# T  Hallucination using LLM as Judge

In this section, we introduce an LLM-as-a-Judge framework for detecting hallucinated spans in LLM-generated responses. We employ two complementary judges: (1) a Contradiction and Missing-Context judge (prompt shown in Fig. 13); and (2) a Calculation and Logical-Error judge (prompt shown in Fig. 14).

---

**LLM as Judge Prompt to detect Contradiction and Missing Context hallucination**

```
You are an expert fact-checker evaluating AI-generated reasoning chains for hallucinations.
**SOURCE OF TRUTH**: The QUESTION/CONTEXT below is the ONLY source of truth.
**CRITICAL INSTRUCTIONS**: - Compare the GENERATED REASONING CHAIN against the QUESTION/CONTEXT -
Flag facts that CONTRADICT information explicitly stated in the question - Flag when important context
from the question is IGNORED or MISSING - Do NOT flag final answers/conclusions (e.g., "the answer
is X") - Do NOT flag restatements or paraphrasing of the question - Do NOT flag use of general world
knowledge (unless it contradicts the question)
**Hallucination Types** (with span requirements):
1. **contradiction**: Answer directly contradicts a SPECIFIC FACT in the question
→ Example: Question says "John has 5 apples", answer says "John has 3 apples"
→ NOT a contradiction: Final answers like "So the answer is 42"
→ Requires: answer_span + question_span (the contradicted fact)
2. **missing_context**: Answer ignores crucial information from the question that affects the
reasoning
→ Example: Question specifies "only on weekdays", but answer ignores this constraint
→ Requires: answer_span (where context should have been used) + question_span (the ignored context)
**QUESTION/CONTEXT (Source of Truth)**: {question}
**GENERATED REASONING CHAIN (Evaluate THIS)**: {answer}
**Output Format** (JSON): json
{
"has_hallucinations": true/false,
"hallucination_count": <number>,
"hallucinations":
[
{
"type": "contradiction|missing_context",
"explanation": "why this is a hallucination",
"answer_span": "exact substring from generated answer",
"question_span": "exact substring from question showing the contradicted/ignored fact"
}
],
}
```

Figure 13: LLM as Judge Prompt to detect Contradiction and Missing Context hallucination.

**LLM as Judge Prompt to detect Calculation Error and Logical Error hallucination**

You are an expert math and logic checker evaluating AI-generated reasoning chains.
**TASK**: Find CALCULATION ERRORS and LOGICAL ERRORS in the reasoning chain.
**CORRECT ANSWER (Ground Truth)**: ground_truth
**CRITICAL INSTRUCTIONS**: - Focus ONLY on mathematical and logical correctness - Check every arithmetic operation for correctness - Check if the logical steps follow correctly from the given information - Do NOT flag correct arithmetic even if inputs came from earlier errors - Do NOT flag the final answer format (e.g., "So the answer is X")
**USING GROUND TRUTH**: - If the model's final answer MATCHES the ground truth, be VERY skeptical about flagging errors → A correct final answer with "errors" likely means you misidentified an error → Double-check your analysis if you want to flag errors when the answer is correct - If the model's final answer does NOT match the ground truth, there MUST be at least one error → Find where the reasoning went wrong that led to the incorrect answer - Ground truth helps you VALIDATE your error detection, not replace careful analysis
**Error Types** (with span requirements):
1. **calculation_error**: The arithmetic computation itself is mathematically WRONG
→ Example: "5 * 4 = 25" is WRONG because 5 * 4 = 20
→ Example: "100 / 2 = 40" is WRONG because 100 / 2 = 50
→ Example: "3 + 7 = 9" is WRONG because 3 + 7 = 10
→ NOT an error: "53 + 70 + 99 = 222" (this is correct arithmetic, even if 53, 70, 99 came from earlier mistakes)
→ The arithmetic operation itself MUST be verifiably incorrect
→ Requires: answer_span (the exact wrong calculation like "5 * 4 = 25"), question_span = null
2. **logical_error**: The reasoning step is logically flawed or misinterprets the problem
→ Example: Using addition when the problem requires subtraction
→ Example: Misunderstanding "days after" vs "days before" in date problems
→ Example: Applying the wrong formula for the given problem type
→ Example: Incorrect date calculations (e.g., wrong day of week, wrong month)
→ Requires: answer_span (the flawed reasoning) + question_span (the misinterpreted part)
**MATH PROBLEM**: {question}
**GENERATED REASONING CHAIN (Check for errors)**: {answer}
**Output Format** (JSON): json
"has_errors": true/false,
"error_count": <number>,
"errors":
[
"type": "calculation_error|logical_error",
"explanation": "what is wrong and what the correct value/logic should be",
"answer_span": "exact substring from answer with the error",
"question_span": "relevant part of question (null for calculation_error)",
],

Figure 14: LLM as Judge Prompt to detect Calculation Error and Logical Error hallucination.

Table 45: Comparison of CoTs with Hallucinations (top) and HoTs that reduce hallucinations (bottom). The question is taken from Break dataset and the answers are generated by `Gemini-1.5-Pro`.

| CoTs With Hallucinations |
| --- |
| **Question:** In the census of 2010, there were 66,135 people, 26,265 (2000 data) households, and 14,051 (2000 data) families residing in the city. The population density was 6,096.7 people per square mile (2,199.9/km²). There were 30,272 (2000 data) housing units at an average density of 2,790.6 per square mile (1,077.2/km²). The racial makeup of the city was 59.38% (52.31% Non-Hispanic) (7.07 White-Hispanic) White (U.S. Census), 24.19% African American (U.S. Census), 14.47% Hispanic (U.S. Census) or Latin (U.S. Census)) of any race, 8.24% from Race (United States Census), 5.74% from two or more races, 2.62% Asian (U.S. Census), 0.69% Native American (U.S.Census), and 0.14% Pacific Islander... How many percent of people were not white-Hispanic? |
| **Answer:** Here's how to solve this: The passage states that the city's racial makeup is 59.38% White, but only 52.31% are Non-Hispanic White. Therefore, the percentage of people who are not White-Hispanic is 100% - 52.31% = 47.69%. Answer: {47.69} |
| **Hallucination:** The hallucination comes from missing context, where the model drops the "Non-Hispanic" qualifier and incorrectly treats 52.31% as White-Hispanic instead of using the stated 7.07%. |
| HoTs Without Hallucinations |
| **Question:** In the census of 2010, there were 66,135 people, 26,265 (2000 data) households, and 14,051 (2000 data) families residing in the city. The population density was 6,096.7 people per square mile (2,199.9/km²). There were 30,272 (2000 data) housing units at an average density of 2,790.6 per square mile (1,077.2/km²). The racial makeup of the city was 59.38% White (52.31% Non-Hispanic White) (7.07% White-Hispanic), 24.19% African American, 14.47% Hispanic or Latin of any race, 8.24% from other races, 5.74% from two or more races, 2.62% Asian, 0.69% Native American, and 0.14% Pacific Islander... How many percent of people were not white-Hispanic? |
| **Answer:** The question asks for the percentage of people who were *not* White-Hispanic. The passage states that White-Hispanics made up 7.07% of the population. Since the total percentage is 100%, the percentage of people who were not White-Hispanic is 100% - 7.07% = 92.93%. So the answer is {92.93}. |

Table 46: Comparison of CoTs with Hallucinations (top) and HoTs that reduce hallucinations (bottom). The question is taken from R-GSM dataset and the answers are generated by `Gemini-1.5-Pro`.

| CoTs With Hallucinations |
| --- |
| **Question:** Thomas withdraws $1000 in 20 dollar bills from the bank account. After getting home, he uses half of the remaining bills to pay for a bill. He then triples his money. He then converts all his bills to 5 dollar bills. He lost 10 bills while getting home. How many 5 dollar bills does he have? |
| **Answer:** Let's break this down: 1. Thomas withdraws $1000 in 20 dollar bills: 1000 / 20 = 50 bills. 2. He uses half of the remaining bills to pay for a bill: 50 / 2 = 25 bills used, 25 bills remaining. 3. Thomas then triples his money: 25 * 20 * 3 = 1500 dollars. 4. He then converts all his bills to 5 dollar bills: 1500 / 5 = 300 bills. 5. He lost 10 bills while getting home: 300 - 10 = 290 five-dollar bills. Therefore, the answer is {290}. |
| **Hallucination:** Can not focus to the context "lost 10 bills while getting home", it should be inferred as "lost 10 bills of 20 dollars". |
| HoTs Without Hallucinations |
| **Question:** Thomas withdraws $1000 in 20 dollar bills from the bank account. After getting home, he uses half of the remaining bills to pay for a bill. Thomas then triples his money. He then converts all his bills to 5 dollar bills. He lost 10 bills while getting home. How many 5 dollar bills does he have? |
| **Answer:** Let's break this down step by step: 1. Thomas withdraws $1000 in 20 dollar bills. This means he has 1000 / 20 = 50 bills initially. 2. He lost 10 bills while getting home, so now he has 50 - 10 = 40 bills. 3. He uses half of the remaining bills to pay for a bill. Half of 40 is 20, so he uses 20 bills and has 20 bills left. 4. Thomas then triples his money. He now has 20 * 3 = 60 bills. 5. At this point, he has 60 * $20 = $1200. 6. He then converts all his bills to 5 dollar bills. $1200 / $5 = 240 five-dollar bills. Therefore, Thomas ends up with 240 five-dollar bills. The answer is {240}. |

Table 47: Comparison of CoTs with Hallucinations (top) and HoTs that reduce hallucinations (bottom). The question is taken from Seven Objects dataset and the answers are generated by ✦.

| CoTs **With** Hallucinations |
| --- |
| **Question:** ... The motorcyle is older than the truck. The station wagon is older than the hatchback. The motorcyle is newer than the hatchback. The bus is the second-newest. The minivan is the oldest. The limousine is older than the station wagon. Options: (A) The station wagon is the newest (B) The limousine is the newest (C) The truck is the newest (D) The minivan is the newest (E) The motorcyle is the newest (F) The hatchback is the newest (G) The bus is the newest |
| **Answer:** Here's how we can determine the order of the vehicles from oldest to newest: 1. **The minivan is the oldest:** This establishes the minivan as the first vehicle in the order. 2. **The limousine is older than the station wagon:** This places the limousine before the station wagon. 3. **The station wagon is older than the hatchback:** ... This makes the final order: Minivan, Limousine, Station Wagon, Hatchback, Motorcycle, Truck, Bus. Therefore, the bus is the newest. Answer: {G} |
| **Hallucination:** Fact: "The bus is the second-newest". Answer claims: "The bus is the newest." Therefore → this does contradict a fact explicitly stated in the question. |
| HoTs **Without** Hallucinations |
| **Question:** ... The motorcycle is older than the truck. The station wagon is older than the hatchback. The motorcycle is newer than the hatchback. The bus is the second-newest. The minivan is the oldest. The limousine is older than the station wagon. Options: (A) The station wagon is the newest (B) The limousine is the newest (C) The truck is the newest (D) The minivan is the newest (E) The motorcycle is the newest (F) The hatchback is the newest (G) The bus is the newest |
| **Answer:** We are given that the bus is the second-newest. This eliminates (G) as the bus cannot be both the second-newest and the newest. The minivan is the oldest, eliminating (D). Now let's consider the other facts. We know the station wagon is older than the hatchback and the limousine is older than the station wagon. This means neither the station wagon (A) nor the limousine (B) is the newest. We also know the motorcycle is newer than the hatchback and older than the truck. This means neither the motorcycle (E) nor the hatchback (F) is the newest. By process of elimination, the only remaining option is the truck (C). The ordering, from oldest to newest, is: minivan, limousine, station wagon, hatchback, motorcycle, truck, bus. This satisfies all given conditions. Therefore, the answer is {C}. |

Table 48: Comparison of CoTs with Hallucinations (top) and HoTs that reduce hallucinations (bottom). The question is taken from SpartQA dataset and the answers are generated by ✦.

| CoTs **With** Hallucinations |
| --- |
| **Question:** We have two blocks. Lets call them A and B. Block A is to the left of B. Block A has a medium black triangle, one medium blue circle and one big yellow circle. One small blue circle is also in this block. The medium black triangle is touching the right edge of this block. Below, near to and to the right of the small blue circle there is the medium blue circle. Below and to the left of the small blue circle there is the big yellow circle.. Below the blue object which is below the small blue circle is the medium black triangle. Block B has one medium black triangle and a small black square. The small black square is touching the bottom edge of this block. The medium black triangle is to the left of the small black square. What is to the left of the small black object? a big yellow circle or a medium black triangle that is touching the right edge of a block? (a) big yellow circle (b) medium black triangle that is touching the right edge of a block (c) both of them (d) none of them |
| **Answer:** Block A contains a big yellow circle and a medium black triangle touching the right edge. Block B contains a small black square and a medium black triangle to its left. Since Block A is to the left of Block B, the medium black triangle in Block A (touching the right edge) is to the left of the small black square (the small black object) in Block B. The big yellow circle is also in Block A, so it's to the left of Block B and thus to the left of the small black square. Answer: {B} |
| **Hallucination:** Fails to include big yellow circle, despite acknowledging it's in Block A (which is to the left of Block B). |
| HoTs **Without** Hallucinations |
| **Question:** We have two blocks. Lets call them A and B. Block A is to the left of B. Block A has a medium black triangle, one medium blue circle and one big yellow circle. One small blue circle is also in this block. The medium black triangle is touching the right edge of this block. Below, near to and to the right of the small blue circle there is the medium blue circle. Below and to the left of the small blue circle there is the big yellow circle. Below the blue object which is below the small blue circle is the medium black triangle. Block B has one medium black triangle and a small black square. The small black square is touching the bottom edge of this block. The medium black triangle is to the left of the small black square. What is to the left of the small black object? a big yellow circle or a medium black triangle that is touching the right edge of a block? (a) big yellow circle (b) medium black triangle that is touching the right edge of a block (c) both of them (d) none of them |
| **Answer:** Block A contains a medium black triangle touching the right edge, a big yellow circle, and a small blue circle. Block B contains a medium black triangle and a small black square touching the bottom edge. Since Block A is to the left of Block B, and the medium black triangle is on the right edge of Block A, it is to the left of the small black square in Block B. The big yellow circle is also in Block A, so it's also to the left of the small black square in Block B. Therefore, the answer is {C}. |

# U  Details of Prompting Methods: LtM, CoVE, Self-Refine, ToT

In this section, we describe a series of prompting methods of increasing sophistication—ranging from Least-to-Most prompting to CoVe, Tree-of-Thought, and Self-Refine—as employed in Sec. 5.4. Code is available at: Github.

## U.1  Least-to-Most Prompting (LtM)

### U.1.1  Method Description

Least-to-Most Prompting (LtM) decomposes complex problems into simpler sub-questions, solving them sequentially while building context from previous answers. We employ two variants of LtM depending on task complexity. For simpler datasets (e.g., Date), we adopt a one-step LtM formulation, where the model directly decomposes and solves the problem in a single prompt. In contrast, for more challenging datasets such as r-GSM and Seven Objects, we use a two-step LtM approach, in which the model first explicitly generates intermediate sub-questions and then answers them sequentially to arrive at the final solution.

**Few-Shot Example in One-Step Prompting**

```
Q: 2015 is coming in 36 hours. What is the date one week from today in MM/DD/YYYY? A: Let's break down
this problem: 1. What is today's date?
2. What is the date one week from today?
1. If 2015 is coming in 36 hours, then it is coming in 2 days. 2 days before 01/01/2015 is 12/30/2014,
so today is 12/30/2014.
2. One week from today is 7 days after 12/30/2014, which is 01/06/2015.
The answer is 01/06/2015.
```

**Few-Shot Example in Two-Step Prompting**

```
DECOMPOSE=
Q: The statements are logically consistent within each paragraph. In an antique car show, there are
seven vehicles: a bus, a motorcycle, a hatchback, a station wagon, a minivan, a truck, and a limousine.
The station wagon is the fourth-newest. The motorcycle is newer than the truck. The station wagon is
older than the hatchback. The minivan is newer than the hatchback. The bus is newer than the minivan.
The truck is newer than the limousine. Which vehicle is the third-oldest?
A: To answer "Which vehicle is the third-oldest?", we need to know: "What is the complete ordering of
vehicles from newest to oldest?"
SOLVE =
The station wagon is the fourth-newest. The motorcycle is newer than the truck. The station wagon is
older than the hatchback. The minivan is newer than the hatchback. The bus is newer than the minivan.
The truck is newer than the limousine.
Q: What is the complete ordering of vehicles from newest to oldest? A: Let me work through the
constraints: - Station wagon is 4th-newest (position 4) - Station wagon is older than hatchback →
hatchback is positions 1-3 - Minivan is newer than hatchback → minivan is also positions 1-3 - Bus is
newer than minivan → bus must be position 1, 2, or 3 - Motorcycle is newer than truck - Truck is newer
than limousine
Since bus > minivan > hatchback, and all are newer than station wagon (position 4): Position 1
(newest): Bus Position 2: Minivan Position 3: Motorcycle (must be here, newer than truck) Position
4: Station wagon Position 5: Hatchback Position 6: Truck Position 7 (oldest): Limousine
The answer is this ordering.
Q: Which vehicle is the third-oldest? A: From the ordering above, the third-oldest is position 5, which
is the hatchback. However, looking at the options and re-checking: position 5 from newest = position 3
from oldest. That's the motorcycle. The answer is {B}.
```

## U.2 Chain-of-Verification (CoVe)

## U.3 Method Description

CoVe generates an initial answer, creates verification questions to check facts, answers them independently, cross-checks for inconsistencies, and produces a revised final answer. We employ this to all datasets (Date, Seven Objects, r-GSM).

### U.3.1 Hyperparameters

| Parameter | Value |
|---|---|
| Temperature | 1.0 |
| Number of verification questions | 3-5 (model-generated) |
| Few-shot examples per stage | 2 |

**Few-shot Example of the Initial Answer Stage**

```
Question: 2015 is coming in 36 hours. What is the date one week from today in MM/DD/YYYY? Answer: If
2015 is coming in 36 hours, then it is coming in 2 days. 2 days before 01/01/2015 is 12/30/2014, so
today is 12/30/2014. So one week from today will be 01/06/2015. The answer is 01/05/2015.
Question: {NEW QUESTION}
Answer:
```

**Few-shot Example of the Plan Verification Stage)**

```
Question: 2015 is coming in 36 hours. What is the date one week from today in MM/DD/YYYY?
Initial Answer: If 2015 is coming in 36 hours, then it is coming in 2 days. 2 days before 01/01/2015
is 12/30/2014, so today is 12/30/2014. So one week from today will be 01/06/2015. The answer is
01/05/2015.
Verification Questions: 1. If 2015 is coming in 36 hours, what is today's date? 2. What date is 7 days
after 12/30/2014? 3. Is 01/05/2015 the same as 01/06/2015?
Question: {NEW QUESTION}
Initial Answer: {NEW INITIAL ANSWER}
Verification Questions:"""
```

**Few-Shot Example of the Execution Stage**

```
Question: If 2015 is coming in 36 hours, what is today's date? Answer: 36 hours is 1.5 days. 1.5 days
before 01/01/2015 is 12/30/2014 at noon. So today is 12/30/2014.
Question: If January 1st, 2019 is a Tuesday, what day of the week is January 7th, 2019? Answer: January
7th is 6 days after January 1st. Tuesday + 6 days = Monday. So January 7th, 2019 is a Monday.
Question: How many days after Tuesday is the first Monday? Answer: The days after Tuesday are:
Wednesday (1), Thursday (2), Friday (3), Saturday (4), Sunday (5), Monday (6). So the first Monday is
6 days after Tuesday.
Question: {NEW VERIFICATION QUESTION}
Answer:
```

## Few-Shot Example of the Cross-Check Stage

```
Original Answer: If 2015 is coming in 36 hours, then it is coming in 2 days. 2 days before 01/01/2015
is 12/30/2014, so today is 12/30/2014. So one week from today will be 01/06/2015. The answer is
01/05/2015.
Verification Question: What date is 7 days after 12/30/2014? Verification Answer: 7 days after
12/30/2014 is 01/06/2015.
Analysis: The original answer states "one week from today will be 01/06/2015" but then gives the final
answer as 01/05/2015. The verification confirms that 7 days after 12/30/2014 is 01/06/2015. This is an
inconsistency – the final answer should be 01/06/2015, not 01/05/2015.
Original Answer: {NEW ORIGINAL ANSWER}
Verification Question: {NEW VERIFICATION QUESTION}
Verification Answer: {NEW VERIFICATION ANSWER}
Analysis:
```

## Few-shot Example of the Final Stage

```
Original Question: 2015 is coming in 36 hours. What is the date one week from today in MM/DD/YYYY?
Initial Answer: If 2015 is coming in 36 hours, then it is coming in 2 days. 2 days before 01/01/2015
is 12/30/2014, so today is 12/30/2014. So one week from today will be 01/06/2015. The answer is
01/05/2015.
Verification Results: 1. Q: If 2015 is coming in 36 hours, what is today's date? A: Today is
12/30/2014. Check: Consistent 2. Q: What date is 7 days after 12/30/2014? A: 7 days after 12/30/2014
is 01/06/2015. Check: Inconsistency – original answer says 01/06/2015 in reasoning but final answer is
01/05/2015
Revised Final Answer: If 2015 is coming in 36 hours, then it is coming in 2 days. 2 days before
01/01/2015 is 12/30/2014, so today is 12/30/2014. So one week from today will be 01/06/2015. The
answer is 01/06/2015.
Original Question: {NEW QUESTION}
Initial Answer: {NEW INITIAL ANSWER}
Verification Results: {NEW VERIFICATION RESULTS}
Revised Final Answer:
```

### U.4 Self-Refine

#### U.4.1 Method Description

Self-Refine iteratively improves answers through self-critique and refinement cycles until the model determines no further improvement is needed or max iterations is reached.

#### U.4.2 Hyperparameters

| Parameter | Value |
|---|---|
| Temperature | 1.0 |
| Max iterations | 3 (recommended in (Madaan et al., 2023)) |
| Stopping condition | "quality: 10/10" or "no further refinement" |

**Few-Shot Example (Critique + Refine)**

```
SELF_CRITIQUE_PROMPT =
Question: 2015 is coming in 36 hours. What is the date one week from today in MM/DD/YYYY? Answer: If
2015 is coming in 36 hours, then it is coming in 2 days. 2 days before 01/01/2015 is 12/30/2014, so
today is 12/30/2014. So one week from today will be 01/06/2015. The answer is 01/06/2015.
Let's check the reasoning step by step:
Step 1: "2015 is coming in 36 hours" → "coming in 2 days" Correct (36 hours = 1.5 days = 2 days)
Step 2: "2 days before 01/01/2015 is 12/30/2014" Correct
Step 3: "one week from today will be 01/06/2015" ERROR!
12/30/2014 + 7 days = 01/06/2015, but this crosses into the new year.
Let me verify: 12/30 + 1 day = 12/31, 12/31 + 1 day = 01/01, 01/01 + 5 days = 01/06 Actually, the
calculation is correct! No errors found.
```

```
REFINE_PROMPT =
Question: Yesterday was April 30, 2021. What is the date tomorrow in MM/DD/YYYY? Answer: Yesterday was
04/30/2021. Tomorrow is two days after yesterday, so tomorrow is 05/02/2021. The answer is 05/02/2021.
Feedback: The calculation "Tomorrow is two days after yesterday" is correct, but the reasoning could
be clearer by explicitly stating what today is.
Refined Answer: Yesterday was 04/30/2021. Today is one day after yesterday, so today is 05/01/2021.
Tomorrow is one day after today, so tomorrow is 05/02/2021. The answer is 05/02/2021.
```

## U.5 Tree-of-Thoughts (ToT)

### U.5.1 Method Description

We adopt Tree-of-Thought (ToT) prompting with breadth-first search and beam pruning, retaining the top-ranked reasoning paths at each step. Task-specific settings are used, with depth–beam configurations of (1, 2) for Date, (2, 3) for r-GSM, and (4, 3) for Seven Objects, following the recommendation of Yao et al. (2023) to cap the search depth at 3. Given the simplicity of the Date dataset, we set the beam width to 1.

**Few-Shot Example (Generate Candidate Step)**

```
Q: The first day of 2019 is a Tuesday, and today is the first Monday of 2019. What is the date today in
MM/DD/YYYY? A: If the first day of 2019 was Tuesday, Jan 1, then the first Monday would be 6 days later.
Jan 1 + 6 days = Jan 7. Answer: 01/07/2019
Q: input
A:
```

**Few-Shot Example (Propose Step)**

```
Input: Today is the last day of the first quarter of 2013. What is the date 24 hours later in
MM/DD/YYYY?
Possible next steps:
Thought 1: Identify the last day of the first quarter (Jan, Feb, Mar). It is March 31.
Thought 2: Determine the number of days in the first quarter of 2013 (non-leap year).
Thought 3: Calculate the date 24 hours (1 day) after the current date.
Input: input
Possible next steps:
```

**Few-Shot Example (Value Step)**

```
Evaluate if the date calculation step is valid (sure/impossible)
Q: Today is the last day of Feb 2015.
Step: Today is Feb 29, 2015.
Judge: impossible (2015 is not a leap year)
Q: Today is Aug 31. What is tomorrow?
Step: Tomorrow is Sep 1.
Judge: sure
Q: input
Step: step
Judge:
```

## V   Why do smaller models struggle with zero-shot HoT?

Understanding the failure of small models in zero-shot HoT requires analyzing *how* they fail, rather than task accuracy alone. HoT requires the model to simultaneously perform reasoning while maintaining a strict structured tag syntax, introducing a non-trivial instruction-following and structural tracking requirement.

Prior work (Murthy et al. (2024)) shows that smaller models (e.g., Llama-3.2-1B and Qwen-2.5-1.5B) exhibit substantially higher instruction-following error rates than larger models (0.69 and 0.54 vs. 0.22 and 0.17 for Llama-3.1-70B and Qwen-2.5-72B). Since HoT requires consistent adherence to a structured format, models that struggle with instruction following are expected to fail before reasoning quality becomes the bottleneck.

We observe that these failures are primarily structural rather than semantic. In the PuzzleTrivial dataset, **90% of Llama-3.2-1B outputs exhibit degenerate repetition**, producing an unbounded tag expansion pattern such as:

```
<fact_i>constant</fact_i> - <fact_j>change in speed</fact_j>
```

The model repeatedly increments indices (e.g., fact6, fact7, . . . ) while preserving locally valid tags but never terminating the structure. This behavior indicates loss of global structural state tracking rather than incorrect reasoning.

Overall, these observations suggest that zero-shot HoT failure in small models arises from unreliable structured decoding, not insufficient reasoning ability.

## W  Mitigating Over-trust for HoT

One concern with HoT is that highlights may fool users into accepting wrong answers, creating an over-trust problem. To investigate a potential mitigation, we prompt a separate LLM judge (Gemini-3-Flash) to rate each highlighted span with a confidence level (low/medium/high). Each confidence rating is determined by two criteria:

1. Whether each tag wraps information that is genuinely important to answering the question.

2. Whether the content wrapped in each `<fact_i>` tag matches semantically the content wrapped in its **corresponding `<fact_i>` tags.**

We select 120 HoT responses from our human study data and also 120 HoT responses for the PuzzleTrivial and AIME 2024 datasets (120 correct vs. 120 incorrect answers by the original LLMs).

We find that **showing the confidence level (e.g., by opacity of the background color in tags) might be a helpful approach**. Specifically, tags in *correct* answers are labeled with more High confidence than *incorrect* ones (79.2% vs. 73.0%). In contrast, *incorrect* answers contain over 2× more Low-confidence tags (13.8% vs. 6.0%). That is, adding an additional LLM judge to rate its confidence over the HoT tags might ameliorate the over-trust problem.

Table 49: LLM judges show lower confidence for tags in incorrect answers than tags in correct answers.

| Tag confidence | Low | Medium | High |
|---|---|---|---|
| Incorrect answers | 13.8% | 13.2% | 73.0% |
| Correct answers | **6.0%** | 14.8% | **79.2%** |

The full prompt used for the confidence evaluation is shown below:

---

**Confidence Annotation Prompt**

Your job is to annotate the following XML tags and determine how likely it is that each tag is actually important. Each <fact> tag should wrap information in the original question and corresponding information in the <answer> that is vital to answering the question. Each tag can have a certainty value between "low", "medium" or "high". Low means that the tag itself is not important to answering the question, or it does not correspond to previous tags of the same number. Medium means that the fact is somewhat important to answering the question, but it is not that critical. Medium can also mean that the tag is somewhat similar to the previous tagged information of that number. High means that the tag is very important to answering the question, and it is very semantically similar to the previous tagged information of that number.

**Example 1**
Input QA Pair:
Question: A person is traveling at 20 km/hr and reached his destiny in 2.5 hr then find the distance?
Answer Choices: (a) 53 km (b) 55 km (c) 52 km (d) 60 km (e) 50 km
Reformatted Question: A person is traveling at <fact1>20 km/hr</fact1> and reached his destination in <fact2>2.5 hr</fact2> then find the distance? Answer Choices: (a) 53 km (b) 55 km (c) 52 km (d) 60 km (e) 50 km
Answer: The distance that the person traveled would have been <fact1>20 km/hr</fact1> * <fact2>2.5 hrs</fact2> = 50 km. So the answer is {e}.

Output:
Question: A person is traveling at 20 km/hr and reached his destiny in 2.5 hr then find the distance?
Answer Choices: (a) 53 km (b) 55 km (c) 52 km (d) 60 km (e) 50 km
Reformatted Question: A person is traveling at <fact1 certainty="high">20 km/hr</fact1> and reached his destination in <fact2 certainty="high">2.5 hr</fact2> then find the distance? Answer Choices: (a) 53 km (b) 55 km (c) 52 km (d) 60 km (e) 50 km

---

Answer: The distance that the person traveled would have been <fact1 certainty="high">20 km/hr</fact1> * <fact2 certainty="high">2.5 hrs</fact2> = 50 km. So the answer is {e}.

**Example 2** (bad tags)
Input QA Pair:
Question: John found that the average of 15 numbers is 40. If 10 is added to each number then the mean of the numbers is? Answer Choices: (a) 50 (b) 45 (c) 65 (d) 78 (e) 64
Reformatted Question: John found that the <fact1>average of 15</fact1> numbers is 40. If 10 is added to each <fact2>number</fact2>, then the <fact3>mean of the numbers</fact3> is? Answer Choices: (a) 50 (b) 45 (c) 65 (d) 78 (e) 64
Answer: If <fact2>10 is added to each number</fact2>, then the <fact3>mean of the numbers</fact3> also increases by <fact1>10</fact1>. So the new mean would be <fact1>40</fact1> + 10 = 50. So the answer is {a}.

Output:
Question: John found that the average of 15 numbers is 40. If 10 is added to each number then the mean of the numbers is? Answer Choices: (a) 50 (b) 45 (c) 65 (d) 78 (e) 64
Reformatted Question: John found that the <fact1 certainty="medium">average of 15</fact1> numbers is 40. If 10 is added to each <fact2 certainty="low">number</fact2>, then the <fact3 certainty="high">mean of the numbers</fact3> is? Answer Choices: (a) 50 (b) 45 (c) 65 (d) 78 (e) 64
Answer: If <fact2 certainty="low">10 is added to each number</fact2>, then the <fact3 certainty="high">mean of the numbers</fact3> also increases by <fact1 certainty="low">10</fact1>. So the new mean would be <fact1 certainty="low">40</fact1> + 10 = 50. So the answer is {a}.

Please annotate the following question and answer pair: {HoT QA Pair}