# OpenReview forum: "HoT: Highlighted Chain of Thought for Referencing Supporting Facts from Inputs"
_TMLR — Accepted by TMLR_

### Review · Reviewer_Gp15 · 2025-11-12

**Summary Of Contributions:**

The paper introduces Highlighted Chain-of-Thought (HoT), a novel inference-time prompting technique. The method instructs a Large Language Model (LLM) to first re-format a given question by wrapping key facts in XML tags (e.g., <fact1>...</fact1>). Subsequently, the LLM generates a Chain-of-Thought (CoT) answer that explicitly references these tags, thereby grounding its reasoning steps in the facts provided in the original prompt.

Key Strengths:
* The central idea of using tags for internal fact-grounding may be of interest
* The ablation study (Figure 4) demonstrates that the performance gain from HoT is not solely due to the "Repeated Question" (R-Q) baseline
* The technique can be used to generate a high-quality dataset for fine-tuning smaller models, which can be seen as a way of distilling knowledge/reasoning

Weaknesses:
* The claim that HoT "outperforms other advanced prompting techniques" like Tree-of-Thought (ToT) and Self-Refine (Table 9), is either misleading or it is not well supported by experimental details. What makes this result very surprising is that ToT or Self Refine are very expensive multi prompt strategies, while HoT is only a single pass (or two pass if counting the first question to add the tags). If this is truly the comparison, it would amount to an extremely surprising result, and the paper must provide substantial detail about the way these baselines are calculated to make sure this is. fair comparison. While the paper provides 70+ pages of example prompts of HoT in main paper and appendix, they provide no detail at all about these baselines, making the judgement about the validity of the baseline and comparisons impossible. Therefore, as of now, one of the papers most important and surprising claims, cannot be objectively judged.
* The paper's primary claim, that HoT "substantially reduces hallucination" (namely in Figure 1) is based on a methodological flaw and/or a misleading statement. The paper relies on SelfCheckGPT, which the cited paper (Manakul et al., 2023) and the authors' own analysis (Appendix R, Figure 12) confirm is a measure of consistency, not factuality. Note that an LLM can consistently output the same hellucinated/incorrect answer, meaning that the two metrics are at best correlated. The authors even state these are "orthogonal metrics", yet build their main finding on this conflation. In another part the authors write "This is because a model’s reasoning could be highly consistent (measured by SelfCheckGPT) but its final answer does not match the ground-truth for a task" So the model could be consistently outputting a wrong result, but that would be counted as a reduction in hallucination, while in this case it is only making the model more certain about its mistake and thus less calibrated. The fact that HoT is advertised as a way to reduce hallucination yet the key metric used to support is consistency is a major misrepresentation of the results.
* The paper's stated mechanism for why HoT works is based on the notion that tokens inside fact tags receive more attention, however the paper's own analysis in Appendix D disproves its own hypothesis, finding that the content inside the tags actually receives less attention, and only the tags themselves receive more attention. The authors justify this by saying that the tags tokens might function as summary tokens for the facts inside the tags. The paper also fundamentally misrepresents the findings of Silva et al. (2024), which is used to motivate and substantiate their claims. Silva et al. do not claim that LLMs inherently "tend to pay more attention to... XML tags". Rather, Silva et al. propose a method (GUIDE) that mechanistically *forces* the model to pay attention by "adding a bias to the attention scores"  using their own specific tags (<!->).

**Audience:**

No

**Audience Explanation:**

Because the claims of the paper are not substantiated and supported, as of now the paper is not of interest to the research community.

**Claims And Evidence:**

No

**Claims Explanation:**

The paper's most significant claims are not supported by accurate or convincing evidence, as elaborated below.

-  The claim in Table 9 that HoT "outperforms" methods like Tree-of-Thought (ToT), Self-Refine, and Chain-of-Verification (COVE) is highly surprising and would have required substantial detail to support, but on the contrary the paper provides virtually zero detail about the way baselines are calculated. To elaborate why this is so surprising, HoT is a single-pass method while these baselines are complex, multi-call, iterative, and search-based frameworks that have a very high computational and query cost. The authors provide no implementation details or cost analysis for these baselines (even in the appendix), creating the impression that their single pass approach outperforms these much more demanding approaches. The claim of superiority is therefore unsubstantiated and cannot be judged.

-  Reducing hallucination is unsupported: This claim is based on a misapplication of the SelfCheckGPT (Manakul et al., 2023) methodology. As confirmed by the paper' own Appendix R, this tool measures response consistency, not factual correctness. The authors themselves concede that "hallucination reduction and accuracy are related but orthogonal metrics", thereby invalidating their own " What they have shown is that HoT makes models more consistent, which is a different claim than reducing hallucination, which is advertised in abstract and much of the rest of paper, is highly misleading.

- One of the paper's key motivations—that LLMs "tend to pay more attention to content inside XML tags"—is based on a misreading of Silva et al. (2024). That paper forces this behavior with a special mechanism, it does not observe it as a natural property of XML. The paper's own data in Appendix D finds that fact tokens inside XML tags actually receive "less attention in HoT compared to CoT". The paper's introduction is directly contradicted by its own appendix.

**Requested Changes:**

- Authors must provide substantial more detail about the way that baselines are calculated, to the extent that these results can be reproduced independently. Without such amount of detail, simply claiming that their method outperforms much more costly method is not a scientifically substantiated and verifiable result.
- Authors have made numerous misleading statement about hallucination, while in fact they are making the models more consistent. Consistency means the model may be more consistent about its hallucination. The entire framing thus is misleading to readers.
- Authors have argued quite a few times that paying more attention to facts inside XML tags are the reason model performance improves, but in Appendix D they provide a completely contradictory evidence that those tags receive *less attention*, and offer entirely new explanation, that XML tags act as summary tokens. Then why not framing the paper around summarization instead of paying more attention instead?

---

> ### Author Response · Authors · 2025-12-19
> **Official Author Responses to Reviewer Gp15 [1/10]**
>
> Thank you for your thoughtful reseponse! Based off your feedback, we have now:
>
> (1) Clarified how other prompting methods (Tree-of-Thought, Self-Refine, Chain-of-Verification, Least-to-Most) were computed and evaluated in `Appendix U`.
>
> (2) Further clarified how SelfCheckGPT measures consistency to accuracy (`Section 5.1`) and add extra results measuring hallucination (`Section 5.3`)
>
> (3) Corrected our representation of Silva et al. (2024) to refine our explanation of why XML-style tags boost LLM accuracy.
>
> We believe these revisions greatly strengthen the paper and resolve the misunderstandings noted in your review. Please find our inline responses below.

---

> ### Author Response · Authors · 2025-12-19
> **Official Author Responses to Reviewer Gp15 [2/10]**
>
> >`Requested Change 1`: Authors must provide substantial more detail about the way that baselines are calculated, to the extent that these results can be reproduced independently.
>
> Yes, we agree! The revised manuscript on OpenReview now has complete prompts, hyperparameters, and sampling settings used for each baseline method (ToT, Self-Refine, COVE, etc.).
> In the `Appendix U`, we have also add description of how each baseline is run in practice, including the number of prompting steps used for each method, the specific prompts employed at each step, and the corresponding few-shot examples.
>
> ### Details of Prompting Methods: LtM, CoVE, Self-Refine, ToT 🔍
>
> In this section, we describe a series of prompting methods of increasing sophistication—ranging from Least-to-Most prompting to CoVE, Tree-of-Thought, and Self-Refine.
> For reproducibility, we release code at our anonymous [Github repo](https://anonymous.4open.science/r/Prompt_Compilation-856D/README.md).

---

> ### Author Response · Authors · 2025-12-19
> **Official Author Responses to Reviewer Gp15 [3/10]**
>
> ### Least-to-Most Prompting (LtM)
>
> #### Method Description
>
> Least-to-Most Prompting (LtM) decomposes complex problems into simpler sub-questions, solving them sequentially while building context from previous answers. We employ two variants of LtM depending on task complexity. For simpler datasets (e.g., Date), we adopt a one-step LtM formulation, where the model directly decomposes and solves the problem in a single prompt. In contrast, for more challenging datasets such as r-GSM and Seven Objects, we use a two-step LtM approach, in which the model first explicitly generates intermediate sub-questions and then answers them sequentially to arrive at the final solution.
>
> #### Few-Shot Example in One-Step Prompting
>
> ```
> Q: 2015 is coming in 36 hours. What is the date one week from today in MM/DD/YYYY?
> A: Let's break down this problem:
> 1. What is today's date?
> 2. What is the date one week from today?
>
> 1. If 2015 is coming in 36 hours, then it is coming in 2 days. 2 days before 01/01/2015 is 12/30/2014, so today is 12/30/2014.
> 2. One week from today is 7 days after 12/30/2014, which is 01/06/2015.
>
> The answer is {01/06/2015}.
> ```
>
> #### Few-Shot Example in Two-Step Prompting
>
> ```
> DECOMPOSE=
> Q: The statements are logically consistent within each paragraph. In an antique car show, there are seven vehicles: a bus, a motorcycle, a hatchback, a station wagon, a minivan, a truck, and a limousine. The station wagon is the fourth-newest. The motorcycle is newer than the truck. The station wagon is older than the hatchback. The minivan is newer than the hatchback. The bus is newer than the minivan. The truck is newer than the limousine. Which vehicle is the third-oldest?
> A: To answer "Which vehicle is the third-oldest?", we need to know: "What is the complete ordering of vehicles from newest to oldest?"
>
> SOLVE =
> The station wagon is the fourth-newest. The motorcycle is newer than the truck. The station wagon is older than the hatchback. The minivan is newer than the hatchback. The bus is newer than the minivan. The truck is newer than the limousine.
>
> Q: What is the complete ordering of vehicles from newest to oldest?
> A: Let me work through the constraints:
> - Station wagon is 4th-newest (position 4)
> - Station wagon is older than hatchback → hatchback is positions 1-3
> - Minivan is newer than hatchback → minivan is also positions 1-3
> - Bus is newer than minivan → bus must be position 1, 2, or 3
> - Motorcycle is newer than truck
> - Truck is newer than limousine
>
> Since bus > minivan > hatchback, and all are newer than station wagon (position 4):
> Position 1 (newest): Bus
> Position 2: Minivan
> Position 3: Motorcycle (must be here, newer than truck)
> Position 4: Station wagon
> Position 5: Hatchback
> Position 6: Truck
> Position 7 (oldest): Limousine
>
> The answer is this ordering.
>
> Q: Which vehicle is the third-oldest?
> A: From the ordering above, the third-oldest is position 5, which is the hatchback. However, looking at the options and re-checking: position 5 from newest = position 3 from oldest. That's the motorcycle. The answer is {B}.
> ```

---

> ### Author Response · Authors · 2025-12-19
> **Official Author Responses to Reviewer Gp15 [4/10]**
>
> ### Chain-of-Verification (CoVE)
>
> #### Method Description
>
> CoVe generates an initial answer, creates verification questions to check facts, answers them independently, cross-checks for inconsistencies, and produces a revised final answer. We employ this to all datasets (Date, Seven Objects, r-GSM).
>
> #### Hyperparameters
>
> | Parameter | Value |
> |-----------|-------|
> | Temperature | 1.0 |
> | Number of verification questions | 3-5 (model-generated) |
> | Few-shot examples per stage | 2 |
>
> #### Few-shot Example of the Initial Answer Stage
>
> ```
> Question: 2015 is coming in 36 hours. What is the date one week from today in MM/DD/YYYY?
> Answer: If 2015 is coming in 36 hours, then it is coming in 2 days.
> 2 days before 01/01/2015 is 12/30/2014, so today is 12/30/2014.
> So one week from today will be 01/06/2015.
> The answer is {01/05/2015}.
>
> Question: {NEW QUESTION}
> Answer:
> ```
>
> #### Few-shot Example of the Plan Verification Stage
>
> ```
> Question: 2015 is coming in 36 hours. What is the date one week from today in MM/DD/YYYY?
>
> Initial Answer: If 2015 is coming in 36 hours, then it is coming in 2 days. 2 days before 01/01/2015 is 12/30/2014, so today is 12/30/2014. So one week from today will be 01/06/2015. The answer is {01/05/2015}.
>
> Verification Questions:
> 1. If 2015 is coming in 36 hours, what is today's date?
> 2. What date is 7 days after 12/30/2014?
> 3. Is 01/05/2015 the same as 01/06/2015?
>
> Question: {NEW QUESTION}
>
> Initial Answer: {NEW INITIAL ANSWER}
>
> Verification Questions:
> ```
>
> #### Few-Shot Example of the Execution Stage
>
> ```
> Question: If 2015 is coming in 36 hours, what is today's date?
> Answer: 36 hours is 1.5 days. 1.5 days before 01/01/2015 is 12/30/2014 at noon. So today is 12/30/2014.
>
> Question: If January 1st, 2019 is a Tuesday, what day of the week is January 7th, 2019?
> Answer: January 7th is 6 days after January 1st. Tuesday + 6 days = Monday. So January 7th, 2019 is a Monday.
>
> Question: How many days after Tuesday is the first Monday?
> Answer: The days after Tuesday are: Wednesday (1), Thursday (2), Friday (3), Saturday (4), Sunday (5), Monday (6). So the first Monday is 6 days after Tuesday.
>
> Question: {NEW VERIFICATION QUESTION}
> Answer:
> ```
>
> #### Few-Shot Example of the Cross-Check Stage
>
> ```
> Original Answer: If 2015 is coming in 36 hours, then it is coming in 2 days. 2 days before 01/01/2015 is 12/30/2014, so today is 12/30/2014. So one week from today will be 01/06/2015. The answer is {01/05/2015}.
>
> Verification Question: What date is 7 days after 12/30/2014?
> Verification Answer: 7 days after 12/30/2014 is 01/06/2015.
>
> Analysis: The original answer states "one week from today will be 01/06/2015" but then gives the final answer as {01/05/2015}. The verification confirms that 7 days after 12/30/2014 is 01/06/2015. This is an inconsistency - the final answer should be 01/06/2015, not 01/05/2015.
>
> Original Answer: {NEW ORIGINAL ANSWER}
>
> Verification Question: {NEW VERIFICATION QUESTION}
>
> Verification Answer: {NEW VERIFICATION ANSWER}
>
> Analysis:
> ```
>
> #### Few-shot Example of the Final Stage
>
> ```
> Original Question: 2015 is coming in 36 hours. What is the date one week from today in MM/DD/YYYY?
>
> Initial Answer: If 2015 is coming in 36 hours, then it is coming in 2 days. 2 days before 01/01/2015 is 12/30/2014, so today is 12/30/2014. So one week from today will be 01/06/2015. The answer is {01/05/2015}.
>
> Verification Results:
> 1. Q: If 2015 is coming in 36 hours, what is today's date?
>    A: Today is 12/30/2014.
>    Check: Consistent
> 2. Q: What date is 7 days after 12/30/2014?
>    A: 7 days after 12/30/2014 is 01/06/2015.
>    Check: Inconsistency - original answer says 01/06/2015 in reasoning but final answer is {01/05/2015}
>
> Revised Final Answer: If 2015 is coming in 36 hours, then it is coming in 2 days. 2 days before 01/01/2015 is 12/30/2014, so today is 12/30/2014. So one week from today will be 01/06/2015. The answer is {01/06/2015}.
>
> Original Question: {NEW QUESTION}
>
> Initial Answer: {NEW INITIAL ANSWER}
>
> Verification Results:
> {NEW VERIFICATION RESULTS}
>
> Revised Final Answer:
> ```

---

> ### Author Response · Authors · 2025-12-19
> **Official Author Responses to Reviewer Gp15 [5/10]**
>
> ### Self-Refine
>
> #### Method Description
>
> Self-Refine iteratively improves answers through self-critique and refinement cycles until the model determines no further improvement is needed or max iterations is reached.
>
> #### Hyperparameters
>
> | Parameter | Value |
> |-----------|-------|
> | Temperature | 1.0 |
> | Max iterations | 3 |
> | Stopping condition | "quality: 10/10" or "no further refinement" |
>
> #### Few-Shot Example (Critique + Refine)
>
> ```
> SELF_CRITIQUE_PROMPT =
> Question: 2015 is coming in 36 hours. What is the date one week from today in MM/DD/YYYY?
> Answer: If 2015 is coming in 36 hours, then it is coming in 2 days.
> 2 days before 01/01/2015 is 12/30/2014, so today is 12/30/2014.
> So one week from today will be 01/06/2015.
> The answer is {01/06/2015}.
>
>
> Let's check the reasoning step by step:
> Step 1: "2015 is coming in 36 hours" → "coming in 2 days" Correct (36 hours = 1.5 days = 2 days)
> Step 2: "2 days before 01/01/2015 is 12/30/2014" Correct
> Step 3: "one week from today will be 01/06/2015" ERROR!
> 12/30/2014 + 7 days = 01/06/2015, but this crosses into the new year.
> Let me verify: 12/30 + 1 day = 12/31, 12/31 + 1 day = 01/01, 01/01 + 5 days = 01/06
> Actually, the calculation is correct! No errors found.
> ```
>
> ```
> REFINE_PROMPT =
> Question: Yesterday was April 30, 2021. What is the date tomorrow in MM/DD/YYYY?
> Answer: Yesterday was 04/30/2021.
> Tomorrow is two days after yesterday, so tomorrow is 05/02/2021.
> The answer is {05/02/2021}.
>
> Feedback: The calculation "Tomorrow is two days after yesterday" is correct, but the reasoning could be clearer by explicitly stating what today is.
>
> Refined Answer: Yesterday was 04/30/2021.
> Today is one day after yesterday, so today is 05/01/2021.
> Tomorrow is one day after today, so tomorrow is 05/02/2021.
> The answer is {05/02/2021}.
> ```

---

> ### Author Response · Authors · 2025-12-19
> **Official Author Responses to Reviewer Gp15 [6/10]**
>
> ### Tree-of-Thoughts (ToT)
>
> #### Method Description
>
> We adopt Tree-of-Thought (ToT) prompting with breadth-first search and beam pruning, retaining the top-ranked reasoning paths at each step. Task-specific settings are used, with depth–beam configurations of (1, 2) for Date, (2, 3) for r-GSM, and (4, 3) for Seven Objects, following the recommendation in the paper to cap the search depth at 3. Given the simplicity of the Date dataset, we set the beam width to 1.
>
> #### Few-Shot Example (Generate Candidate Step)
>
> ```
> Q: The first day of 2019 is a Tuesday, and today is the first Monday of 2019. What is the date today in MM/DD/YYYY?
> A: If the first day of 2019 was Tuesday, Jan 1, then the first Monday would be 6 days later. Jan 1 + 6 days = Jan 7. Answer: {01/07/2019}
>
> Q: {input}
> A:
> ```
>
> #### Few-Shot Example (Propose Step)
>
> ```
> Input: Today is the last day of the first quarter of 2013. What is the date 24 hours later in MM/DD/YYYY?
> Possible next steps:
> Thought 1: Identify the last day of the first quarter (Jan, Feb, Mar). It is March 31.
> Thought 2: Determine the number of days in the first quarter of 2013 (non-leap year).
> Thought 3: Calculate the date 24 hours (1 day) after the current date.
>
> Input: {input}
> Possible next steps:
> ```
>
> #### Few-Shot Example (Value Step)
>
> ```
> Evaluate if the date calculation step is valid (sure/impossible)
>
> Q: Today is the last day of Feb 2015.
> Step: Today is Feb 29, 2015.
> Judge: impossible (2015 is not a leap year)
>
> Q: Today is Aug 31. What is tomorrow?
> Step: Tomorrow is Sep 1.
> Judge: sure
>
> Q: {input}
> Step: {step}
> Judge:
> ```

---

> ### Author Response · Authors · 2025-12-19
> **Official Author Responses to Reviewer Gp15 [7/10]**
>
> >`Requested Change 2`: Authors have made numerous misleading statement about hallucination, while in fact they are making the models more consistent. Consistency means the model may be more consistent about its hallucination. The entire framing thus is misleading to readers.
>
> Thank you for the clarification! While the authors of SelfCheckGPT wrote that consistency correlates with reduced hallucination, we agree it is not a direct measure of hallucinations, and we have revised our framing accordingly.
>
> Thanks to your comment, which motivates us to measure more directly the correlation between hallucination evidence and our improved accuracy.
> To do that, we have also run experiments that use LLM-as-a-judge (the full prompt is in `Appendix T`) to determine which spans in LLM responses are hallucinated.
> Because our hallucination evaluation covers reading comprehension, arithmetic, and logical reasoning tasks, we use four broad hallucination types: contradiction, missing context, calculation error, logical error—which align with prior taxonomies [A]. In mathematical reasoning, this distinction between calculation and logical errors is well established [B]. Below are the definition of these hallucinations:
> 1) **Contradiction**: Answer directly contradicts a SPECIFIC FACT in the question.
> 2) **Missing Context**: Answer ignores crucial information from the question that affects the reasoning.
> 3) **Calculation Error**: The arithmetic computation itself is mathematically WRONG.
> 4) **Logical Error**: The reasoning step is logically flawed or misinterprets the problem.
>
> **We provided the above hallucination definitions to an LLM judge and asked it to check whether an LLM response contains any span is hallucinated** (details below ⬇️).
>
> ## References
> [A] Cossio, Manuel. "A comprehensive taxonomy of hallucinations in large language models." arXiv preprint arXiv:2508.01781 (2025).
>
> [B] Li, Ruosen, Ziming Luo, and Xinya Du. "Fine-grained Hallucination Detection and Mitigation in Language Model Mathematical Reasoning." (2024).

---

> ### Author Response · Authors · 2025-12-19
> **Official Author Responses to Reviewer Gp15 [8/10]**
>
> We evaluate reading comprehension datasets (DROP_Break, SpartQA) using Contradiction and Missing Context, since hallucinations in RC primarily arise from misusing or ignoring information in the passage. In contrast, for arithmetic and logical reasoning datasets (r-GSM, GSM_Symbolic, Seven Objects), hallucinations mainly stem from faulty computation or invalid inferential steps, so we use Calculation Error and Logical Error as the corresponding labels. This allows each dataset to be evaluated using the error types that most accurately capture its dominant failure modes. For each dataset, we evaluate 200 Gemini-Pro-1.5 CoT and HoT responses using Gemini-2.5-flash-lite-preview-09-2025 as the judge and compute the hallucination rate as the proportion of responses that contain at least one hallucination.
> Overall, reducing hallucinations (see `Table R1`) tends to improve accuracy (see `Table R2`), but the extent of this effect varies by task.
>
> _`Table R1`: The table reports the reduction in hallucinations from HoT to CoT (lower is better), measuring the fraction of responses containing hallucinations._
> | Hallucination Rate ↓ | R-GSM | GSM-Symbolic | SpartQA | Break | Seven Objects |
> |---|---|---|---|---|---|
> | CoT | 12.00 | 14.50 | 80.00 | 14.50 | 21.50 |
> | HoT | 6.00 | 14.50 | 74.00 | 14.00 | 12.50 |
> | $\Delta (CoT-HoT)$ | 6.00 | 0.00 | 6.00 | 0.50 | 9.00 |
>
> _`Table R2`: Accuracy Comparison Between CoT and HoT Across Different Benchmarks_
> | Accuracy ↑ | r-GSM | GSM-Symbolic | SpartQA | Break | Seven Objects |
> |---|---|---|---|---|---|
> | CoT | 78.18 | 85.67 | 61.88 | 86.39 | 86.00 |
> | HoT | 90.91 | 86.33 | 64.85 | 87.36 | 88.80 |
> | $\Delta(HoT-CoT)$ | 12.73 | 0.66 | 2.97 | 0.97 | 2.80 |
>
>
> We observe that while HoT reduces the hallucination rate in Seven Objects by `9.00` points but only improves the accuracy by `2.80` points. One possible explanation is that the **Seven Objects** dataset requires complex multi-step reasoning to reach the final answer. Hallucinations often appear in intermediate reasoning steps (as shown below), yet the model can still produce the correct final answer. Consequently, fixing these hallucinations yields only minimal accuracy improvements.

---

> ### Author Response · Authors · 2025-12-19
> **Official Author Responses to Reviewer Gp15 [9/10]**
>
> To reveal *how* or *where* the model has hallucination, we present qualitative examples from R-GSM and Seven Objects below.
>
> 🟥 **Example 1: R-GSM**
> Carmen goes to an auction to win an antique desk. The bids on the desk rise by 50 each time and 3 other people each bid once. She accepts the opening bid of 200 and continues bidding until she wins. `Carmen bids after each of the 3 other people` and eventually wins. How much money, in dollars, does the desk cost her? (`GT: 500`)
> Answer from Gemini 1.5 Pro: Since 3 other people bid on the desk and the bids increase by 50 each time, `the price increased by 3*50 = 150` from the other bidders.
> This makes the price of the desk 200 + 150 = 350 before Carmen makes the winning bid.
> Since Carmen bids after each of the other people, she makes the winning bid, increasing the price by another 50.
> Therefore, the desk costs Carmen 350 + 50 = 400.
> Answer: {400} (Incorrect, GT: 500)
> There is one hallucinated span `3*50 = 150` that contains logical error, because the question states that `Carmen bids after each of the 3 other people and eventually wins.`, so So the sequence of bids is: Carmen: 200 --> Other: 250 --> Carmen: 300 --> Other: $350 --> Carmen: 400 --> Other: 450 --> Carmen: 500 ⟵ winning bid.
>
>
> 🟦 **Example 2: Seven Objects**
> The following paragraphs each describe a set of seven objects arranged in a fixed order. The statements are logically consistent within each paragraph. A fruit stand sells seven fruits: oranges, loquats, apples, kiwis, mangoes, plums, and peaches. The apples are the second-cheapest. The peaches are the cheapest. The plums are the fourth-most expensive. The loquats are the third-cheapest. The kiwis are more expensive than the mangoes. The oranges are less expensive than the mangoes. (`GT: B`)
> Options:
> (A) The oranges are the third-cheapest
> (B) The loquats are the third-cheapest
> \(C) The apples are the third-cheapest
> (D) The kiwis are the third-cheapest
> (E) The mangoes are the third-cheapest
> (F) The plums are the third-cheapest
> (G) The peaches are the third-cheapest
> Answer from Gemini 1.5 Pro: Here's how we can determine the third-cheapest fruit:
>
> 1. **Peaches are the cheapest:** This means peaches occupy the 1st (least expensive) position.
> 2. **Apples are the second-cheapest:** Apples occupy the 2nd position.
> 3. **Loquats are the third-cheapest:** Loquats occupy the 3rd position.
> 4. **Plums are the fourth-most expensive:**  Since there are seven fruits, the `fourth-most expensive is the same as the third-cheapest`. However, we already know the loquats are the third-cheapest. This confirms the existing information.
> 5. **Kiwis are more expensive than mangoes:** Kiwis are above mangoes in price.
> 6. **Oranges are less expensive than mangoes:** Oranges are below mangoes in price.
> This information gives us the following order from cheapest to most expensive: 1. Peaches, 2. Apples, 3. Loquats, 4. Oranges, 5. Mangoes, 6. Kiwis, 7. Plums
> The question asks for the third-cheapest fruit.
> Answer: {B} (Correct)
>
> There is one hallucinated span that contains calculation error: `the fourth-most expensive is the same as the third-cheapest`, because in a set of 7 items ordered by price, the fourth-most expensive item is the fourth-cheapest item (position 4), not the third-cheapest item (position 3). In this example, although the final answer is correct, the reasoning still contains a hallucination, illustrating that reducing hallucinations does not necessarily guarantee higher accuracy.
>
> ## References
> [A] Cossio, Manuel. "A comprehensive taxonomy of hallucinations in large language models." arXiv preprint arXiv:2508.01781 (2025).
>
> [B] Li, Ruosen, Ziming Luo, and Xinya Du. "Fine-grained Hallucination Detection and Mitigation in Language Model Mathematical Reasoning." (2024).

---

> ### Author Response · Authors · 2025-12-19
> **Official Author Responses to Reviewer Gp15 [10/10]**
>
> >`Requested Change 3`: Authors have argued quite a few times that paying more attention to facts inside XML tags are the reason model performance improves, but in Appendix D they provide a completely contradictory evidence that those tags receive less attention, and offer entirely new explanation, that XML tags act as summary tokens. Then why not framing the paper around summarization instead of paying more attention instead?
>
> We thank the reviewer for this observation and acknowledge that our original framing can be improved.
>
> Our key finding in Appendix D is that the **total attention allocated to fact-related tokens (XML tags + enclosed content) increases substantially under HoT compared to CoT**.
>
> While attention to the enclosed content tokens alone decreases slightly (e.g., 5.61% → 2.79% for Llama-3.1-70B), the XML tag tokens receive substantial amounts of attention, resulting in a large net increase in attention to the entire fact-related region.
> That is, XML tags and enclosed content receive an average of `6.85%` attention while the remaining content receives `100-6.85 = 93.15%` attention (`Table R3`)
>
> We now explicitly state that the observed effect is increased attention to the XML tags themselves, which results in greater total attention to the tagged span as a whole.
>
> _`Table R3`: Comparison of total percent of attention (%) allocated to tokens in Llama-3.1-70B and Llama-3.1-405B for CoT and HoT._
> | Method | Llama-3.1-70B | Llama-3.1-405B | Avg |
> |-|-|-|-|
> | HoT Total| 8.40 | 5.30 | 6.85 |
> | Fact Tokens | 2.79| 1.85 | 2.32  |
> | XML Tokens  | 5.61 | 3.45 | 4.53  |
>
> ### > Rather, Silva et al. propose a method (GUIDE) that mechanistically forces the model to pay attention by "adding a bias to the attention scores" using their own specific tags (<!->).
>
> Thank you for raising this important point! We agree that our original wording unintentionally implied that XML tags in HoT improve LLM accuracy by increasing LLM attention to the enclosed tokens inside the tags (which is a misinterpretation of Silva et al. 2024 `[C]`).
>
> Our revised manuscript in OpenReview now only mentions Silva et al. 2024 in the Related Work for the broader point that structured attention towards key information can improve task performance.
>
> Additionally, **our hypothesis** is that the embeddings for these <fact#> XML tags may actually encode the information of the enclosed contents.
> That is, these <fact#> tokens may be the **sink** token (i.e, the **"attention sink"** phenomenon in `[D]` and `[E]`).
> For example, in text, the "attention sink" tokens (e.g., [CLS] or the first token) have been shown to carry contextual information [E]. In Vision Transformers, the model often attends heavily to the background tokens (instead of the subject), yet these high-embedding embeddings actually encode global information about the entire image [D]. We suspect that XML tags may serve a similar function in our setting by encoding the information of the phrases they enclose.
>
>
> ## References
> [C] Pedro Luiz Silva, Antonio de Domenico, Ali Maatouk, and Fadhel Ayed. Pay attention to what matters. arXiv preprint arXiv:2409.19001, 2024.
>
> [D] Timothée Darcet, Maxime Oquab, Julien Mairal, and Piotr Bojanowski. Vision transformers need registers.arXiv preprint arXiv:2309.16588, 2023
>
> [E] Xiangming Gu, Tianyu Pang, Chao Du, Qian Liu, Fengzhuo Zhang, Cunxiao Du, Ye Wang, and Min Lin. When attention sink emerges in language models: An empirical view. arXiv preprint arXiv:2410.10781, 2024.

---

> ### Comment · Reviewer_Gp15 · 2026-03-12
>
> Authors have failed to respond to my review above, or have mis-placed it under another reviewer's comments. Thus, as of now, it is impossible to read the rebuttal.

---

> > ### Author Response · Authors · 2026-03-12
> > **Apology for Rebuttal Visibility Issue**
> >
> > Dear Reviewer,
> >
> > We appreciate your time and efforts in serving TMLR!
> >
> > We sincerely apologize for the confusion and thank you for pointing this out.
> >
> > We originally posted our rebuttal on `19 Dec 2025`, but it appears the visibility was mistakenly not set to `Visible to Everyone`, which prevented reviewers from seeing it. We have now corrected the setting so the response should be accessible.
> >
> > We are very sorry for the inconvenience and appreciate your patience. Please let us know if it is now visible.
> >
> > Thank you again for your time and support!

---

> > > ### Author Response · Authors · 2026-04-08
> > > **Request for additional reviewer input**
> > >
> > > Dear Reviewer Gp15,
> > >
> > > Thank you again for your time and for reviewing our submission.
> > >
> > > We wanted to kindly follow up to see if you might have a chance to revisit the paper after our rebuttal. We have carefully addressed the concerns raised in your review and clarified several key points that we hope improve the paper. Thank you again for your consideration.
> > >
> > > Best regards,
> > > The Authors

---

> > ### Comment · Action_Editor_sDSm · 2026-04-08
> > **Can you please take another look at the author responses?**
> >
> > The authors have their responses visible; can you please take a look at them and see if they adequately address your concerns?
> > thanks!

---

### Review · Reviewer_N37j · 2025-11-26

**Summary Of Contributions:**

This paper introduces a simple yet impactful technique (HoT) with strong empirical backing. The insights into hallucination reduction and human verification are particularly valuable. Addressing the limitations/ethical concerns will strengthen its appeal for deployment in real-world systems.

**Audience:**

Yes

**Audience Explanation:**

Introduces Highlighted Chain-of-Thought Prompting (HoT), which prompts LLMs to generate XML-tagged responses grounding facts from the question. This improves attention to key details and reduces hallucination.

Demonstrates that HoT reduces hallucination rates by ~6.3 points (from 21.22% to 14.92%) compared to vanilla CoT, as measured by SelfCheckGPT.

 Shows consistent accuracy gains across 22 tasks (arithmetic, QA, logical reasoning, etc.) averaging +2.61 pp over CoT. Notably achieves +11.38 pp on adversarial ReasoningTrap benchmarks.

**Claims And Evidence:**

Yes

**Claims Explanation:**

See my comments.

**Requested Changes:**

Broaden discussion on failure cases (e.g., when HoT highlights lead users to trust incorrect answers). Provide mitigation strategies or warnings.

Address token overhead for long-context questions—quantify impact (e.g., % increase in tokens).

The paper notes hallucination drops but accuracy gains vary (e.g., Seven Objects: hallucination ↓50%, accuracy ↑4.4). Expand explanations, perhaps hallucinations aren’t always tied to incorrect answers.

Clarify how "key facts" are defined for XML tagging (leave-one-out principle). Provide inter-annotator agreement if humans curated examples.

Discuss why XML tags (<fact1>) outperformed other formats (e.g., <!-> in GUIDE).

---

> ### Author Response · Authors · 2025-12-19
> **Official Author Responses to Reviewer N37j [1/9]**
>
> Thank you for your thoughtful feedback! We appreciate your positive remarks on the clarity of our paper, the extensive tests, and the generality & significance of our findings.
>
> To address all of your concerns, we have further:
>
> (1) analyzed the failure cases of HoT users (`Appendix B`)
>
> (2) calculated additional token overhead of HoT over CoT (`Appendix C`)
>
> (3) clarified how SelfCheckGPT measures consistency to accuracy (`Section 5.1`) and add extra results measuring hallucination (`Section 5.3`)
>
> (4) provided a concrete definition of key-fact tagging (`Appendix H.3`)
>
> (5) justified our choice of the <fact{num}> XML-style format.
>
> We have incorporated the changes to our manuscript on OpenReview for your review. Below are our inline responses.

---

> ### Author Response · Authors · 2025-12-19
> **Official Author Responses to Reviewer N37j [2/9]**
>
> > `Requested Change 1a`: Broaden discussion on failure cases (e.g., when HoT highlights lead users to trust incorrect answers).
>
> Thank you for raising this important point! We have reviewed the user-study results and further investigated 30 cases with the highest human–LLM verification error, where the LLMs (Gemini-Flash, Gemini-Pro, Llama-8B,70B, or 405B) produced an incorrect answer but users flagged it as correct. We noticed 4 main failure modes (also in `Appendix B`):
>
> `1. Correct facts being highlighted but answer is wrong due to the lack of a reasoning step (20%)`: The LLM highlights relevant facts correctly, but the final answer is wrong because the LLM skips a reasoning step. Since the highlighted spans look valid, users might have mistakenly assumed the final answer is also correct without going through the reasoning.
>
> ```
> Question: Arash is raising money for a music festival. He has applied for help from the <fact1>youth club, which has decided to cover 1/5 of the cost of the music festival</fact1>. How much money is Arash missing if he has <fact2>¥90</fact2> and the music festival costs <fact3>¥420</fact3>?
> Answer: The youth club will cover <fact1>1/5</fact1> of the cost of the music festival, which is <fact1>1/5</fact1> * <fact3>¥420</fact3> = ¥84. Arash has <fact2>¥90</fact2>, so he has ¥90 - ¥84 = ¥6 more than the amount covered by the youth club. Since the music festival costs ¥420, Arash is missing ¥420 - ¥90 = ¥330.
> ```
>
> **Explanation:** LLM omits the final reasoning step of subtracting ¥84 from ¥330 to obtain the correct result of ¥246. This paper only asks LLMs to highlight key facts, which did not help users recognize missing steps. Yet, in the future work, it would be interesting to ask LLMs to highlight potentially failure cases or weak statements in a chain of thought.
>
> `2. Inaccurate arithmetic calculation (10%).`
> In the math dataset (r-GSM), the model occasionally performs incorrect intermediate calculations.
>
> ```
> Question: A fog bank rolls in from the ocean to cover a city. <fact1>It takes 241 minutes to cover every 5 miles of the city</fact1>. If <fact2>the city is 85 miles across from the oceanfront to the opposite inland edge</fact2>, how many minutes will it take for the fog bank to cover the whole city?
> Answer: The total distance across the city is <fact2>85 miles</fact2> and the distance covered in the given time is <fact1>5 miles</fact1>. So, the number of times the given distance fits into the total distance is <fact2>85</fact2>/<fact1>5</fact1> = 17. Now, we multiply this by the given time, which is <fact1>241 minutes</fact1>. Therefore, the total time it will take for the fog bank to cover the whole city is 17 * <fact1>241 minutes</fact1> = 4100 minutes,...
> ```
> **Explanation:** LLM incorrectly computes 17 * 241 = 4100 (which should be 4097 instead).
> Users might overlook this arithmetic error.

---

> ### Author Response · Authors · 2025-12-19
> **Official Author Responses to Reviewer N37j [3/9]**
>
> `3. Missing Question Requirements (30%)`
> The LLM highlights facts from the passage that are relevant but does not reflect key constraints specified in the question, leading to an incomplete or incorrect answer.
>
> We hypothesize that users fail to detect the error because the highlighted facts appear relevant and well-grounded, which can give the impression that the answer satisfies the question constraints even when critical requirements are missing.
>
> ```
> E.g. Question with Highlights: The popular uprising included large areas of <fact1>Lower Bavaria</fact1>, the <fact2>Innviertel</fact2> and <fact3>eastern Bavaria</fact3>. The lands adjoining towns and strongholds were captured by the rebels and the peasant uprising spread to the <fact4>Bavarian Forest</fact4>, parts of the <fact5>Upper Palatinate</fact5> and <fact6>Kelheim on the Danube</fact6>....How many regions did the popular uprising start in?
> Answer: According to the passage, the popular uprising started in <fact1>Lower Bavaria</fact1>, the <fact2>Innviertel</fact2>, and <fact3>eastern Bavaria</fact3> initially, then spread to the <fact4>Bavarian Forest</fact4>, the <fact5>Upper Palatinate</fact5>, and <fact6>Kelheim on the Danube</fact6>. So the answer is {6}.
> ```
>
> **Explanation**: Here the question asks for how many regions did the popular uprising `start in`? But the LLM lists out all the possible places, and leads to incorrect counting.
>
> `4. Missing key facts (40%)`
> In some cases, the model does not highlight all facts required to answer the question. As a result, the response relies on partial evidence and reaches an incorrect final conclusion.
>
> We hypothesize that users accept the answer because the cited evidence appears coherent and sufficient, causing them to overlook whether all facts needed for the final conclusion are included.
> ```
> Question: ... <fact1>The Saints would add onto their lead in the fourth quarter as kicker Garrett Hartley made a 46-yard and a 19-yard field goal</fact1>. .... In the end, New Orleans got the last laugh as Hartley booted the game-ending 37-yard field goal. How many field goals did Garrett Hartley kick in the fourth quarter?
> Answer: The passage states that <fact1>in the fourth quarter, Garrett Hartley kicked a 46-yard and a 19-yard field goal</fact1>. This means he kicked two field goals in the fourth quarter. So the answer is {2}.
>
> ```
> **Explanation**: Here the LLM misses a key fact `In the end, New Orleans got the last laugh as Hartley booted the game-ending 37-yard field goal`, so that the conclusion should be 3 not 2. The grounding seem plausible, this might make users believe the answer.
>
> > `Requested Change 1b:` Provide mitigation strategies or warnings.
>
> Based on our error analysis, we focus on ways to help users better notice incorrect answers, rather than improving the LLM itself. In particular, users may benefit from designs that make it easier to see when an answer does not fully address what the question asks.
>
> One possible direction is to show clear warnings to users when the highlighted evidence may be incomplete, helping them notice that the final answer may not be fully supported. While we do not test this in our work, it suggests a promising direction for future, user-centered improvements.
>
> To address the issue of missing question requirements, one possible direction is to present the key requirements of the question explicitly to users, so they can check whether the highlighted evidence and final answer actually satisfy what the question asks.
>
> To further support user verification and reduce overreliance, another direction is to highlight intermediate reasoning steps to users, allowing them to more easily identify potential gaps or errors.

---

> ### Author Response · Authors · 2025-12-19
> **Official Author Responses to Reviewer N37j [4/9]**
>
> >`Requested Change 2`: Address token overhead for long-context questions—quantify impact (e.g., % increase in tokens).
>
> Based on your great suggestion, we have analyzed the token overhead for long-context questions resulting from our tags and report the quantified impact below (`Appendix C`):
>
> On average over 5 long-context datasets, HoT prompting increases the number of tokens in LLM responses by `~62%` (see `Table R1`).
> This overhead is because LLMs (a) regenerate the question with tags around key facts and (b) generate an answer with tags as well.
>
>
> _`Table R1`: The table presents average token lengths for the original question, the CoT answer, the full HoT response, the HoT reformatted question, and the HoT answer from Gemini-Pro. Overhead (%) is defined as the proportion of HoT tokens devoted to the reformatted question._
> |Dataset|CoT Total|HoT Total|HoT Reformat Q|HoT Answer|Overhead%|
> |-|-|-|-|-|-|
> |Cencus|62|469|361|105|77.1%|
> |Break|72|481|371|108|77.0%|
> |BBEH Spatial Reasoning|594|1810|1200|608|66.3%|
> |BBEH Time Arithmetic|498|796|256|537|32.2%|
> |BBEH Shuffled Objects|911|1073|608|462|56.6%|
> |Mean|427|926|559|364|61.8%|
>
> For long-thinking models where LLMs could spend up to 20K tokens in thinking (e.g., Grok-4), this overhead would becomes increasingly more negligible.

---

> ### Author Response · Authors · 2025-12-19
> **Official Author Responses to Reviewer N37j [5/9]**
>
> >`Requested Change 3`: The paper notes hallucination drops but accuracy gains vary (e.g., Seven Objects: hallucination ↓50%, accuracy ↑4.4). Expand explanations, perhaps hallucinations aren’t always tied to incorrect answers.
>
> Thank you for raising this important point.
>
> `1. Clarifying the reviewer’s concern.`
>
> We wish to clarify that our empirical observations are consistent: When hallucination rate reduces, the LLM answers more correctly and therefore, its accuracy should increases  (see `Table R2`).
> We agree that this trend is sublinear though (e.g., SelfCheckGPT score ↓50%, accuracy ↑4.4) and depends on specific LLMs and datasets.
> Since SelfCheckGPT score measures the rate of inconsistent answers across multiple independent runs, the lower the better.
>
>
> _`Table R2`: The table below reports the differences (Δ) between HoT and CoT in SelfCheckGPT scores and accuracy . Across five different datasets, Gemini 1.5 Pro consistently reduces SelfCheckGPT hallucination rate and increases accuracy when switching from CoT to HoT prompting._
> <!--(200 Gemini-1.5-Pro responses/dataset)-->
>
>
> | Dataset | SelfCheckGPT ↓ | Accuracy ↑ |
> |---------|----------------|------------|
> | R-GSM |-5.74 ↓| +0.45 ↑ |
> | GSM-Symbolic |-8.42 ↓ | +2.00 ↑ |
> | SpartQA | -1.09 ↓ | +11.88 ↑ |
> | Break | -3.69 ↓| +2.64 ↑ |
> | Seven Objects | -12.56 ↓| +4.40 ↑ |
>
> `2. What SelfCheckGPT measures`
> It is important to note that SelfCheckGPT does not directly measure hallucination, but rather _inconsistency_ across multiple generations.
> Inconsistency or randomness in generated outputs can lead to inaccurate reasoning answers, hence high a SelfCheckGPT inconsistency rate correlates with hallucination.
> We have revised the text in `Sec 5.1` to make this distinction explicit.

---

> ### Author Response · Authors · 2025-12-19
> **Official Author Responses to Reviewer N37j [6/9]**
>
> `3. LLMs-as-a-judge detects lower hallucination as task accuracy improves`
>
> Thanks to your comment, which motivates us to measure more directly the correlation between hallucination evidence and our improved accuracy.
> To do that, we have also run experiments that use LLM-as-a-judge to determine which spans in LLM responses are hallucinated (`Appendix T`).
>
> Because our hallucination evaluation covers reading comprehension, arithmetic, and logical reasoning tasks, we use four broad hallucination types: contradiction, missing context, calculation error, logical error—which align with prior taxonomies [A]. In mathematical reasoning, this distinction between calculation and logical errors is well established [B]. Below are the definition of these hallucinations:
> 1) **Contradiction**: Answer directly contradicts a SPECIFIC FACT in the question.
> 2) **Missing Context**: Answer ignores crucial information from the question that affects the reasoning.
> 3) **Calculation Error**: The arithmetic computation itself is mathematically WRONG.
> 4) **Logical Error**: The reasoning step is logically flawed or misinterprets the problem.
>
> We use these hallucination types by giving their definitions to the LLM-as-a-Judge and asking it to check whether a response contains any spans that have relevant error types, depending on the dataset.
>
> We evaluate reading comprehension datasets (DROP_Break, SpartQA) using Contradiction and Missing Context, since hallucinations in RC primarily arise from misusing or ignoring information in the passage. In contrast, for arithmetic and logical reasoning datasets (r-GSM, GSM_Symbolic, Seven Objects), hallucinations mainly stem from faulty computation or invalid inferential steps, so we use Calculation Error and Logical Error as the corresponding labels. This allows each dataset to be evaluated using the error types that most accurately capture its dominant failure modes. For each dataset, we evaluate 200 Gemini-Pro-1.5 CoT and HoT responses using Gemini-2.5-flash-lite-preview-09-2025 as the judge and compute the hallucination rate as the proportion of responses that contain at least one hallucination.
> Overall, reducing hallucinations (see `Table R3`) tends to improve accuracy (see `Table R4`), but the extent of this effect varies by task.
>
> _`Table R3`: The table reports the reduction in hallucinations from HoT to CoT (lower is better), measuring the fraction of responses containing hallucinations._
> | Hallucination Rate ↓ | R-GSM | GSM-Symbolic | SpartQA | Break | Seven Objects |
> |---|---|---|---|---|---|
> | CoT | 12.00 | 14.50 | 80.00 | 14.50 | 21.50 |
> | HoT | 6.00 | 14.50 | 74.00 | 14.00 | 12.50 |
> | $\Delta (CoT-HoT)$ | 6.00 | 0.00 | 6.00 | 0.50 | 9.00 |
>
> _`Table R4`: Accuracy Comparison Between CoT and HoT Across Different Benchmarks_
> | Accuracy ↑ | r-GSM | GSM-Symbolic | SpartQA | Break | Seven Objects |
> |---|---|---|---|---|---|
> | CoT | 78.18 | 85.67 | 61.88 | 86.39 | 86.00 |
> | HoT | 90.91 | 86.33 | 64.85 | 87.36 | 88.80 |
> | $\Delta(HoT-CoT)$ | 12.73 | 0.66 | 2.97 | 0.97 | 2.80 |
>
>
> We observe that while HoT reduces the hallucination rate in Seven Objects by `9.00` points but only improves the accuracy by `2.80` points. One possible explanation is that the **Seven Objects** dataset requires complex multi-step reasoning to reach the final answer. Hallucinations often appear in intermediate reasoning steps (as shown below), yet the model can still produce the correct final answer. Consequently, fixing these hallucinations yields only minimal accuracy improvements.
>
> ## References
> [A] Cossio, Manuel. "A comprehensive taxonomy of hallucinations in large language models." arXiv preprint arXiv:2508.01781 (2025).
>
> [B] Li, Ruosen, Ziming Luo, and Xinya Du. "Fine-grained Hallucination Detection and Mitigation in Language Model Mathematical Reasoning." (2024).

---

> ### Author Response · Authors · 2025-12-19
> **Official Author Responses to Reviewer N37j [7/9]**
>
> To reveal *how* or *where* the model has hallucination, we present qualitative examples from R-GSM and Seven Objects below.
>
> 🟥 **Example 1: R-GSM**
> Carmen goes to an auction to win an antique desk. The bids on the desk rise by 50 each time and 3 other people each bid once. She accepts the opening bid of 200 and continues bidding until she wins. `Carmen bids after each of the 3 other people` and eventually wins. How much money, in dollars, does the desk cost her? (`GT: 500`)
> Answer from Gemini 1.5 Pro: Since 3 other people bid on the desk and the bids increase by 50 each time, `the price increased by 3*50 = 150` from the other bidders.
> This makes the price of the desk 200 + 150 = 350 before Carmen makes the winning bid.
> Since Carmen bids after each of the other people, she makes the winning bid, increasing the price by another 50.
> Therefore, the desk costs Carmen 350 + 50 = 400.
> Answer: {400} (Incorrect, GT: 500)
> There is one hallucinated span `3*50 = 150` that contains logical error, because the question states that `Carmen bids after each of the 3 other people and eventually wins.`, so So the sequence of bids is: Carmen: $200 --> Other: 250 --> Carmen: 300 --> Other: 350 --> Carmen: 400 --> Other: 450 --> Carmen: 500 ⟵ winning bid.
>
>
> 🟦 **Example 2: Seven Objects**
> The following paragraphs each describe a set of seven objects arranged in a fixed order. The statements are logically consistent within each paragraph. A fruit stand sells seven fruits: oranges, loquats, apples, kiwis, mangoes, plums, and peaches. The apples are the second-cheapest. The peaches are the cheapest. The plums are the fourth-most expensive. The loquats are the third-cheapest. The kiwis are more expensive than the mangoes. The oranges are less expensive than the mangoes. (`GT: B`)
> Options:
> (A) The oranges are the third-cheapest
> (B) The loquats are the third-cheapest
> \(C) The apples are the third-cheapest
> (D) The kiwis are the third-cheapest
> (E) The mangoes are the third-cheapest
> (F) The plums are the third-cheapest
> (G) The peaches are the third-cheapest
> Answer from Gemini 1.5 Pro: Here's how we can determine the third-cheapest fruit:
>
> 1. **Peaches are the cheapest:** This means peaches occupy the 1st (least expensive) position.
> 2. **Apples are the second-cheapest:** Apples occupy the 2nd position.
> 3. **Loquats are the third-cheapest:** Loquats occupy the 3rd position.
> 4. **Plums are the fourth-most expensive:**  Since there are seven fruits, the `fourth-most expensive is the same as the third-cheapest`. However, we already know the loquats are the third-cheapest. This confirms the existing information.
> 5. **Kiwis are more expensive than mangoes:** Kiwis are above mangoes in price.
> 6. **Oranges are less expensive than mangoes:** Oranges are below mangoes in price.
> This information gives us the following order from cheapest to most expensive: 1. Peaches, 2. Apples, 3. Loquats, 4. Oranges, 5. Mangoes, 6. Kiwis, 7. Plums
> The question asks for the third-cheapest fruit.
> Answer: {B} (Correct)
>
> There is one hallucinated span that contains calculation error: `the fourth-most expensive is the same as the third-cheapest`, because in a set of 7 items ordered by price, the fourth-most expensive item is the fourth-cheapest item (position 4), not the third-cheapest item (position 3). In this example, although the final answer is correct, the reasoning still contains a hallucination, illustrating that reducing hallucinations does not necessarily guarantee higher accuracy.

---

> ### Author Response · Authors · 2025-12-19
> **Official Author Responses to Reviewer N37j [8/9]**
>
> >`Requested Change 4`: Clarify how "key facts" are defined for XML tagging (leave-one-out principle). Provide inter-annotator agreement if humans curated examples.
>
> Thank you for raising this point! In our setup, “key facts” are defined using a leave-one-out principle (Li et al. 2016; `Sec 3.2`): a phrase is considered a key fact if replacing it with the closest alternative (e.g., `15` --> `14` or `apple` --> `orange`) would **change the correct answer**. We need to replace the phrase by an alternative to keep the question's grammar correct.
>
> We provided this definition directly in the tagging instructions (`Appendix N`) given to the human annotators and models:
>
> ```
> instruction_for_grounding_in_question = 'Read the question and insert the tags into the question via the following rules:\
> 1. Insert only tags keeping the original words unchanged.\
> 2. Put the tags (e.g., <fact1></fact1>, <fact2></fact2>) around the shortest and most concise important phrases.\
> 3. A phrase is considered important and should be tagged if replacing that phrase by a closest alternative phrase would change the answer. \
> 4. Do not tag phrases non-important to answering the question.\
> Re-generate the question after adding tags to the phrases:\
> ### QUESTION: '
> ```
>
> For example, in the below question, if `15 trees` is replaced by `14 trees`, the groundtruth answer would change from `6` to `5`. Therefore `15 trees` is considered a key fact and should be tagged.
>
> ```
> Example:
> Question: There are 15 trees in the grove. Grove workers will
> plant trees in the grove today. After they are done, there will be 21 trees. How many trees did the grove workers plant today?
> Ground Truth Anwswer: 6.
> ```

---

> ### Author Response · Authors · 2025-12-19
> **Official Author Responses to Reviewer N37j [9/9]**
>
> >`Requested Change 5:` Discuss why XML tags ([object Object]) outperformed other formats (e.g., <!-> in GUIDE).
>
> We have not specifically tested the `[object Object]` or `<!->` formats, but we have experimented with a wide range of tagging styles in `Table R5` ( `Appendix H`). In our experiments, the `<fact{num}>` style consistently gave the best accuracy.
>
> _`Table R5`: <fact{num}> tags are the most effective format for answer accuracy in HoT._
>
> |Tag Type|SpartQA|r-GSM|GSM8K|
> |-|-|-|-|
> |<fact{num}> (HoT)|**59.00**|**93.50**|**96.50**|
> |<FACT{num}>|51.00|91.50|-|
> |<key_fact{num}>|54.00|91.50|-|
> |<data{num}>|56.00|92.00|95.00|
> |<important{num}>|54.00|92.50|95.00|
> |<important_quote{num}>|-|-|94.50|

---

> > ### Author Response · Authors · 2026-03-12
> > **Apology for Rebuttal Visibility Issue**
> >
> > Dear Reviewer,
> >
> > We appreciate your time and efforts in serving TMLR!
> >
> > We sincerely apologize for the confusion.
> >
> > We originally posted our rebuttal on `19 Dec 2025`, but it appears the visibility was mistakenly not set to `Visible to Everyone`, which prevented reviewers from seeing it. We have now corrected the setting so the response should be accessible.
> >
> > We are very sorry for the inconvenience and appreciate your patience. Please let us know if it is now visible.
> >
> > Thank you again for your time and support!

---

> > > ### Comment · Reviewer_N37j · 2026-04-08
> > > **Official Comment**
> > >
> > > Thank you to the authors for changing the visibility range of the rebuttal. The authors' responses have addressed my concerns. This paper can be accepted.

---

### Review · Reviewer_ChyD · 2026-02-04

**Summary Of Contributions:**

Contributions:
1. The paper introduces Highlighted Chain-of-Thought (HoT), a prompting technique designed to reduce hallucinations and improve grounding in Large Language Models (LLMs). Its core contributions include:
2. The HoT Framework: A two-step process where an LLM first reformats a question using XML tags to identify key facts and then generates a reasoning chain that references these tags to ground its answer.
3. Performance Benchmarking: Evaluation across 22 tasks and 5 LLMs (including Gemini 1.5 and Llama 3.1), demonstrating consistent accuracy gains.
4. Human Verification Study: An empirical study with 63 users showing that HoT highlights improve verification speed by approximately 25%.
5. Small Model Optimization: Evidence that fine-tuning small models on HoT-formatted data improves their out-of-distribution (OOD) accuracy and reduces attention entropy.

Key Strengths:
1. Broad Empirical Scope: The authors tested the method on a diverse array of tasks, including arithmetic, logical reasoning, and long-context puzzles.
2. Human-Centric Evaluation: Unlike many prompting papers, this work measures the actual impact on human-AI collaboration, specifically verification efficiency.
3. Synergy with Existing Methods: The paper proves that HoT can be combined with other advanced techniques like Self-Consistency and ComplexCoT for additive gains.

Key Weaknesses:
1. Increased Token Latency: HoT requires the model to repeat the question, which increases token consumption and latency, particularly for long-context inputs.
2. Human Annotation Overhead: Creating high-quality, task-specific 8-shot demonstrations is a "costly human effort".
3. The "Over-trust" Risk: The highlights tend to fool users into believing incorrect answers are correct, reducing verification accuracy for wrong LLM responses.

**Audience:**

Yes

**Audience Explanation:**

Yes, the findings are highly relevant to the TMLR audience for several reasons:

1. LLM Grounding: Hallucination remains a critical bottleneck in ML research; a simple, effective prompting method to mitigate this is of high interest.
2. Human-AI Interaction: The paper contributes valuable data on how model outputs (highlights) affect human decision-making and cognitive trust.
3. Model Efficiency: The section on fine-tuning small models to emulate the reasoning focus of larger models provides a roadmap for developing more capable, lightweight agents.

**Broader Impact Concerns:**

The primary concern is misplaced user trust. Because HoT makes responses look more authoritative and "grounded" through highlights, users are less likely to catch errors when the model does hallucinate.

Recommendation: The paper should include a section in the Broader Impact Statement discussing the risks of "hallucination cloaking." It must warn that highlighting techniques can inadvertently create a "veneer of factuality" that might lead to dangerous errors in high-stakes domains like medicine or law if the underlying model logic is still flawed.

**Claims And Evidence:**

Yes

**Claims Explanation:**

Yes, the claims are supported by accurate and clear evidence. The authors provide a multi-faceted validation:


1. Hallucination Reduction: Using both SelfCheckGPT (consistency) and LLM-as-a-judge (span-level), the authors demonstrate that HoT lowers hallucination rates across multiple datasets.

2. Accuracy Gains: Table 3 shows significant improvements in "Reasoning Trap" tasks (+11.38 percentage points on average), while Tables 4-7 detail consistent wins across arithmetic and logical reasoning.

3. Mechanistic Insight: Attention entropy analysis (Table 14) provides a technical explanation for why the method works: HoT-finetuned models focus more intensely on key input facts.

4. Ablation Clarity: Figure 4 effectively breaks down the contribution of each component, showing that repeating the question and adding tags independently contribute to performance.

**Requested Changes:**

Critical Changes (Required for Acceptance)

1. Mitigation for Over-trust: The finding that highlights "fool users" into accepting wrong answers is a significant safety/usability concern. The authors should propose or test a "negative constraint" instruction where the LLM is asked to highlight uncertainties or contradictions to balance the user's trust.
2. Token/Cost Analysis: A clear comparison of the token overhead vs. accuracy gain is necessary. Specifically, quantify the average percentage increase in prompt length when using HoT compared to standard CoT.

Strengthening Adjustments (Optional)

1. Generalization of Tags: The authors found <fact{i}> worked best. Exploring if semantic tags (e.g., <date>, <math_constraint>) further improve performance would strengthen the methodology.
2. Small Model Zero-Shot: Further discuss why smaller models fail at zero-shot HoT and whether specific "formatting-only" pre-training could alleviate this without full SFT.

---

> ### Author Response · Authors · 2026-02-17
> **Official Author Responses to Reviewer ChyD [1/7]**
>
> Thank you for your thoughtful feedback! We appreciate your positive remarks on our work.
>
> To address all of your requested changes, we have:
> 1. Discussed mitigations for over-trust (`Appendix W`).
> 2. Quantified the token overhead of HoT compared to CoT prompting (`Appendix C`).
> 3. Discussed the generalization of tags with semantic formats (`Appendix H`).
> 4. Analyzed the failure of smaller models in zero-shot settings (`Appendix V`).
> 5. Emphasized the risks of hallucination cloaking (`Section Broader Impact Statement`).
>
> We have incorporated the changes to our manuscript on OpenReview for your review. Below are our inline responses:

---

> ### Author Response · Authors · 2026-02-17
> **Official Author Responses to Reviewer ChyD [2/7]**
>
> >`Requested Change 1`: Mitigation for Over-trust: The finding that highlights "fool users" into accepting wrong answers is a significant safety/usability concern. The authors should propose or test a "negative constraint" instruction where the LLM is asked to highlight uncertainties or contradictions to balance the user's trust.
>
> Thank you for your valuable feedback!
> We agree that over-trust based on highlights is a real concern for users of HoTs and LLM-generated text in general.
>
> We performed your suggested test---asking a separate LLM judge (Gemini-3-Flash) to rate each highlighted span with a confidence level (low/medium/high). Each confidence rating is determined by two criteria:
> 1. whether each tag wraps information that is genuinely important to answering the question
> 2. whether the content wrapped in each `<fact_i>` tag matches semantically the content wrapped in its **corresponding** `<fact_i>` tags.
>
> We select 240 responses: (1) 120 HoT responses from our human study data and (2) 120 HoT responses for the PuzzleTrivial and AIME 2024 datasets for a total of 120 correct and 120 incorrect answers by the original LLMs.
>
> We find that **showing the confidence level (e.g. by opacity of the background color in tags) might be a helpful approach**, as the reviewer suggested. Specifically, tags in `correct` answers are labeled with more High confidence than `incorrect` ones (79.2 vs. 73.0). In contrast, `incorrect` answers contain over 2$\times$ more Low-confidence tags (13.8 vs. 6.0). That is, adding an additional LLM judge to rate its confidence over the HoT tags might ameliorate the over-trust problem.
>
> We have added this section in `Appendix W`.
>
>
> `Table R1`: _LLM judges show lower confidence for tags in incorrect answers than tags in correct answers._
> | Tag confidence | Low | Medium | High |
> |-|-|-|-|
> | Incorrect answers| 13.8% | 13.2%| 73.0%|
> | Correct answers| **6.0%**| 14.8% | **79.2%**|

---

> ### Author Response · Authors · 2026-02-17
> **Official Author Responses to Reviewer ChyD [3/7]**
>
> ### The full prompt used for evaluation is below:
> ```
> Your job is to annotate the following XML tags and determine how likely it is that each tag is actually important. Each <fact> tag should wrap information in the original question and corresponding information in the <answer> that is vital to answering the question. Each tag can have a certainty value between "low", "medium" or "high". Low means that the tag itself is not important to answering the question, or it does not correspond to previous tags of the same number. Medium means that the fact is somewhat important to answering the question, but it is not that critical. Medium can also mean that the tag is somewhat similar to the previous tagged information of that number. High means that the tag is very important to answering the question, and it is very semantically similar to the previous tagged information of that number.
>
> ## Example 1
> Input QA Pair:
> Question: A person is traveling at 20 km/hr and reached his destiny in 2.5 hr then find the distance? Answer Choices: (a) 53 km (b) 55 km (c) 52 km (d) 60 km (e) 50 km
> Reformatted Question: A person is traveling at <fact1>20 km/hr</fact1> and reached his destination in <fact2>2.5 hr</fact2> then find the distance? Answer Choices: (a) 53 km (b) 55 km (c) 52 km (d) 60 km (e) 50 km
> Answer: The distance that the person traveled would have been <fact1>20 km/hr</fact1> * <fact2>2.5 hrs</fact2> = 50 km. So the answer is {e}.
>
> Output:
> Question: A person is traveling at 20 km/hr and reached his destiny in 2.5 hr then find the distance? Answer Choices: (a) 53 km (b) 55 km (c) 52 km (d) 60 km (e) 50 km
> Reformatted Question: A person is traveling at <fact1 certainty="high">20 km/hr</fact1> and reached his destination in <fact2 certainty="high">2.5 hr</fact2> then find the distance? Answer Choices: (a) 53 km (b) 55 km (c) 52 km (d) 60 km (e) 50 km
> Answer: The distance that the person traveled would have been <fact1 certainty="high">20 km/hr</fact1> * <fact2 certainty="high">2.5 hrs</fact2> = 50 km. So the answer is {e}.
>
> ## Example 2
> Now let's show an example where there are bad tags:
> Input QA Pair:
> Question: John found that the average of 15 numbers is 40. If 10 is added to each number then the mean of the numbers is? Answer Choices: (a) 50 (b) 45 (c) 65 (d) 78 (e) 64
> Reformatted Question: John found that the <fact1>average of 15</fact1> numbers is 40. If 10 is added to each <fact2>number</fact2>, then the <fact3>mean of the numbers</fact3> is? Answer Choices: (a) 50 (b) 45 (c) 65 (d) 78 (e) 64
> Answer: If <fact2>10 is added to each number</fact2>, then the <fact3>mean of the numbers</fact3> also increases by <fact1>10</fact1>. So the new mean would be <fact1>40</fact1> + 10 = 50. So the answer is {a}.
>
> Output:
> Question: John found that the average of 15 numbers is 40. If 10 is added to each number then the mean of the numbers is? Answer Choices: (a) 50 (b) 45 (c) 65 (d) 78 (e) 64
> Reformatted Question: John found that the <fact1 certainty="medium">average of 15</fact1> numbers is 40. If 10 is added to each <fact2 certainty="low">number</fact2>, then the <fact3 certainty="high">mean of the numbers</fact3> is? Answer Choices: (a) 50 (b) 45 (c) 65 (d) 78 (e) 64
> Answer: If <fact2 certainty="low">10 is added to each number</fact2>, then the <fact3 certainty="high">mean of the numbers</fact3> also increases by <fact1 certainty="low">10</fact1>. So the new mean would be <fact1 certainty="low">40</fact1> + 10 = 50. So the answer is {a}.
>
>
> Please annotate the following question and answer pair: {HoT QA Pair}
> ```

---

> ### Author Response · Authors · 2026-02-17
> **Official Author Responses to Reviewer ChyD [4/7]**
>
> >`Requested Change 2`: Token/Cost Analysis: A clear comparison of the token overhead vs. accuracy gain is necessary. Specifically, quantify the average percentage increase in prompt length when using HoT compared to standard CoT.
>
> Based on your suggestion, we have compared the increase in the output tokens (2.a) and input tokens (2.b) between HoT vs. CoT on 5 long-context datasets below (also in `Appendix C`).
>
> ### 2.a. Output Token Analysis: HoT vs CoT
>
> In theory, HoT prompting might make LLMs output more tokens due to the two steps: (1) re-formatting i.e., re-generating the input question with tags; and (2) generating a response with tags.
>
> Interestingly, **HoT is not CoT + tags**. As shown in `Table R2`, counting only non-XML tokens, the model tends to produce fewer tokens (364 vs. 427 tokens) while also having higher accuracy (+1.55).
> Most of the overhead comes from re-formatting the question (559 tokens).
>
> Notably, in Table R2, **as context length increases, the relative overhead diminishes significantly (from 656.45% on short tasks to 17.78% on complex ones)**. For long-thinking models where LLMs may spend upwards of 20K tokens on reasoning (e.g., Grok-4), this overhead becomes increasingly negligible.
>
> `Table R2`: Average number of output tokens for HoT and CoT responses for Gemini-1.5-Pro. Overhead (%) is the proportion of HoT Total tokens compared to CoT Total tokens. Datasets are sorted ascendingly by context length.
> | Dataset | CoT Total | HoT Reformat Q | HoT Answer | HoT Total | Overhead | Acc Gain |
> |-|-|-|-|-|-|-|
> | Census | 62 | 361 | 105 | 469 | +656.45% | +0.75 |
> | Break | 72 | 371 | 108 | 481 | +568.06% | +0.97 |
> | BBEH Spatial Reasoning | 594 | 1200 | 608 | 1810 | +204.71% | +1.51 |
> | BBEH Time Arithmetic | 498 | 256 | 537 | 796 | +59.84% | +2.50 |
> | BBEH Shuffled Objects | 911 | 608 | 462 | 1073 | +17.78% | +2.00 |
> | Mean | 427 | 559 | 364 | 926 | +301.37 | +1.55 |
>
> ### 2.b. Input Token Analysis
>
> On average, HoT increases the number of tokens in LLM input prompts by `~39%` (see `Table R3`). This increase is primarily due to HoT including fewshot examples explaining how to follow tag formatting.
> This is a **fixed cost** that **decreases proportionally with context length**—from 74.26% overhead for shorter questions (Census) to 17.52-22.25% for longer, complex tasks (BBEH datasets).
>
> `Table R3`: Average input tokens for CoT and HoT. Overhead (%) is defined as the proportion of additional tokens in the HoT prompt compared to the CoT prompt. Datasets are sorted by ascending context length.
> |Dataset|CoT Input| HoT Input|Overhead|Acc Gain|
> |-|-|-|-|-|
> |Census|1344|2342|74.26%|+0.75|
> |Break|783|1252|59.90%|+0.97|
> |BBEH Spatial Reasoning|1809|2126|17.52%|+1.51|
> |BBEH Time Arithmetic|885|1072|21.13%|+2.50|
> |BBEH Shuffled Objects|2922|3572|22.25%|+2.00|
> |Mean|1548.60|2072.86|39.0|+1.55|

---

> ### Author Response · Authors · 2026-02-17
> **Official Author Responses to Reviewer ChyD [5/7]**
>
> > 3. (Optional) Generalization of Tags: The authors found <fact{i}> worked best. Exploring if semantic tags (e.g., [object Object], [object Object]) further improve performance would strengthen the methodology.
>
> We thank the reviewer for this thoughtful suggestion! We agree that exploring semantic (role-based) tags could be a promising idea to try for future work.
>
> However, in order to run experiments at the moment, we are not aware of any existing datasets that define these `object` labels for us to follow. Furthermore, incorporating semantic tags into a specific dataset might require substantial annotation effort and domain expertise.
>
> In light of your suggestion, we found that  `<fact{i}>` outperformed `<FACT{num}>, <key_fact{num}>, <data{num}>, <important{num}>, <important_quote{num}>` in Table R4 (shown in `Appendix H`).
>
> _`Table R4`: <fact{num}> tags are the most effective format for answer accuracy in HoT._
>
> |Tag Type|SpartQA|r-GSM|GSM8K|
> |-|-|-|-|
> |<fact{num}> (HoT)|**59.00**|**93.50**|**96.50**|
> |<FACT{num}>|51.00|91.50|-|
> |<key_fact{num}>|54.00|91.50|-|
> |<data{num}>|56.00|92.00|95.00|
> |<important{num}>|54.00|92.50|95.00|
> |<important_quote{num}>|-|-|94.50|

---

> ### Author Response · Authors · 2026-02-17
> **Official Author Responses to Reviewer ChyD [6/7]**
>
> > 4. (Optional) Small Model Zero-Shot: Further discuss why smaller models fail at zero-shot HoT and whether specific "formatting-only" pre-training could alleviate this without full SFT.
>
> This is a great question! We agree that understanding why smaller models struggle with zero-shot HoT is an important question.
>
> We agree that smaller models (e.g., Llama-3.2-1B and Qwen-2.5-1.5B) struggle in zero-shot HoT settings (Section V), as HoT requires strict adherence to structured tag syntax. This is consistent with findings from Murthy (2024 [A]) which show that `smaller models` (e.g., Llama-3.2-1B and Qwen-2.5-1.5B) have substantially `higher instruction-following error rates` than `larger models`—0.69 and 0.54 versus 0.22 and 0.17 for Llama-3.1-70B and Qwen-2.5-72B, respectively.
>
> Given that responding in HoT format requires moderate amounts of instruction following, it is expected that the models that are the very worst at listening to user instructions would also struggle with HoT.
>
> **We also observe a degenerate repetition failure in these small models**:
>
> For example, when prompted to produce HoT on PuzzleTrivial, `90%` of the responses of Llama-3.2-1B generate an infinite tag expansion pattern, repeatedly extending `<fact_i>constant</fact_i> − <fact_j>change in speed</fact_j>` while incrementing indices (e.g., fact6, fact7, …). The model preserves local tag syntax but fails to terminate the structure.
>
> ### References
> [A] Murthy, R., Kumar, P., Venkateswaran, P., & Contractor, D. (2024). Evaluating the Instruction-following Abilities of Language Models using Knowledge Tasks.
>
> ### > Discuss whether specific "formatting-only" pre-training could alleviate [small model performance] without full SFT.
>
> While we agree this is an interesting direction, we did not perform pre-training because pre-training experiments require significant amounts of resources that are not available to our research group and most academic labs.
>
> However, we did perform supervised fine-tuning (SFT) experiments on both Qwen2.5-1.5B and Llama3.2-1B on around `3400 HoT samples` generated by `Gemini-1.5-Pro` from `17 datasets` in our paper (`Sec. 5.7`). After training, both models showed large gains (Qwen `+11.64%`, Llama `+11.9%`), which demonstrate that with a small amount of data, smaller models *can* learn HoT with sufficient training.
>
> Supporting this, Ai et al. (2024)'s [B] work adds an auxiliary span-tagging supervision during fine-tuning and reports improved QA accuracy (EM/F1). This suggests that asking LLMs in the pre-training might be an interesting direction to explore.
>
> ### References
>
> [B] Ai et al., Enhancing Pre-Trained Generative Language Models with Question Attended Span Extraction on MRC, EMNLP 2024.

---

> ### Author Response · Authors · 2026-02-17
> **Official Author Responses to Reviewer ChyD [7/7]**
>
> > Recommendation: The paper should include a section in the Broader Impact Statement discussing the risks of "hallucination cloaking." It must warn that highlighting techniques can inadvertently create a "veneer of factuality" that might lead to dangerous errors in high-stakes domains like medicine or law if the underlying model logic is still flawed.
>
> We agree with the reviewer that highlights can introduce a risk of hallucination cloaking.
>
> To address this concern, we have explicitly added a discussion of
> hallucination cloaking to the Broader Impact Statement (`Section 8`):
>
> * **HoT highlights supporting information alongside model answers, but it can also introduce risks**. When a model presents highlighted evidence, the output may appear more trustworthy—even if the reasoning is flawed or the conclusion is incorrect. Our human study shows that highlighting can increase users’ confidence in model responses, including incorrect ones. In such cases, the structured format may create a false sense of reliability. This is especially concerning in high-stakes areas such as medicine, law, or financial decision-making, where small errors can have serious consequences.
>
> * **We emphasize that highlighting does not guarantee correctness**. Users should treat HoT as a tool for inspection, not as proof that the answer is accurate. Future work should explore additional safeguards, such as independent verification, uncertainty estimation, and human review, to reduce the risk of over-trust.

---

> ### Author Response · Authors · 2026-04-08
> **Request for additional reviewer input**
>
> Dear Reviewer ChyD,
>
> Thank you again for your time and for reviewing our submission.
>
> We wanted to kindly follow up to see if you might have a chance to revisit the paper after our rebuttal. We have carefully addressed the concerns raised in your review and clarified several key points that we hope improve the paper. Thank you again for your consideration.
>
> Best regards,
> The Authors

---

### Author Response · Authors · 2026-03-12
**Clarification on Rebuttal Visibility**

Dear Action Editor and Reviewers,

We sincerely apologize for the confusion regarding the rebuttal visibility.

Our rebuttal was originally posted on `December 19, 2025`, but due to a mistake in the visibility setting (it was not set to `Visible to Everyone`), it appears that reviewers were unable to see it. We have now corrected the setting so that the rebuttal should be fully accessible.

We truly appreciate the reviewers bringing this issue to our attention, and we are very sorry for the inconvenience this may have caused during the review process.

If possible, we would be grateful if you could kindly take a look at the rebuttal now that it is visible. Please let us know if there are still any issues accessing it.

Thank you very much again for your time and for your efforts!

---

### Author Response · Authors · 2026-04-24
**Follow-Up on the Review Status**

Dear TMLR Editors,

I hope you are doing well. I wanted to kindly follow up on my previous email regarding our TMLR submission (5980).
I understand the editorial team is very busy, and I truly appreciate all the work you do for the community. I was wondering if there have been any updates regarding the review process or an estimated timeline for the next steps.

As a PhD student, I'm anxiously waiting for the decision as it strongly influences my graduation plan and internship opportunity.

Thank you again for your time and support.

Best regards

---

### Author Response · Authors · 2026-05-12
**Thank you for your support**

Dear AC and Reviewers,
We would like to sincerely thank you for your time, effort, and thoughtful feedback throughout the review process. Your comments and suggestions were invaluable in helping us improve the clarity, quality, and overall strength of the manuscript.
Thank you again for your careful consideration and support of our work.

Best regards,
 The Authors

---

### Decision · Action_Editor_sDSm · 2026-05-06

**Recommendation:** Accept as is

**Audience:**

Yes

**Audience Explanation:**

Portion of the audience interested in LLMs and factuality

**Claims And Evidence:**

Yes

**Claims Explanation:**

Two out of the three reviewers agree that the claims of the paper are sound (eg supported by evidence), and the paper is of interest to part of the community.

This paper had an issue with the third reviewer -- the authors had responded to the review, but forgotten to make their response "visible"; the reviewer then disappeared and I could not get them to respond back. However looking through the reviews, I am happy to make the decision based on the remaining two.